# EFFICIENT INFERENCE FOR LARGE LANGUAGE MODEL-BASED GENERATIVE RECOMMENDATION

**Xinyu Lin**[1*] **Chaoqun Yang**[2*] **Wenjie Wang**[3†] **Yongqi Li**[4] **Cunxiao Du**
**Fuli Feng**[3†] **See-Kiong Ng**[1] **Tat-Seng Chua**[1]
[1]National University of Singapore  [2]Tsinghua University
[3]University of Science and Technology of China
[4]The Hong Kong Polytechnic University
xylin1028@gmail.com, chaoqun@yang.email.cn,
{wenjiewang96,liyongqi0,cnsdunm,fulifeng93}@gmail.com,
seekiong@nus.edu.sg, dcscts@nus.edu.sg

## ABSTRACT

Large Language Model (LLM)-based generative recommendation has achieved notable success, yet its practical deployment is costly particularly due to excessive inference latency caused by autoregressive decoding. For lossless LLM decoding acceleration, Speculative Decoding (SD) has emerged as a promising solution. However, applying SD to generative recommendation presents unique challenges due to the requirement of generating top-$K$ items (*i.e.,* $K$ distinct token sequences) as a recommendation list by beam search. This leads to more stringent verification in SD, where all the top-$K$ sequences from the target LLM must be successfully drafted by the draft model at each decoding step. To alleviate this, we consider 1) boosting top-$K$ sequence alignment between the draft model and the target LLM, and 2) relaxing the verification strategy to reduce trivial LLM calls. To this end, we propose an alignment framework named *AtSpeed*, which presents the *AtSpeed-S* optimization objective for top-$K$ alignment under the strict top-$K$ verification. Moreover, we introduce a relaxed sampling verification strategy that allows high-probability non-top-$K$ drafted sequences to be accepted, significantly reducing LLM calls. Correspondingly, we propose *AtSpeed-R* for top-$K$ alignment under this relaxed sampling verification. Empirical results on two real-world datasets demonstrate that AtSpeed significantly accelerates LLM-based generative recommendation, *e.g.,* near 2× speedup under strict top-$K$ verification and up to 2.5× speedup under relaxed sampling verification. The codes and datasets are available at https://github.com/Linxyhaha/AtSpeed.

## 1 INTRODUCTION

Large Language Model (LLM)-based generative recommendation has achieved remarkable performance, emerging as a promising avenue and attracting widespread attention (Bao et al., 2023; Rajput et al., 2023b). Technically speaking, LLMs encode the user's historical interactions, and then perform multiple LLM calls (*i.e.,* forward processes of LLMs) autoregressively to decode top-$K$ ranked items as recommendations (Zheng et al., 2024). Despite the effectiveness, the inference of LLM-based recommender models is unaffordably time-consuming, hindering real-world deployments (Cui et al., 2024). Such intolerable time consumption primarily arises from the decoding process as shown in Figure 1(a), where multiple serial LLM calls are required for step-by-step autoregressive generation (Leviathan et al., 2023; Cai et al., 2024). In light of this, it is essential to achieve lossless LLM decoding acceleration for LLM-based generative recommendation.

To accelerate the LLM decoding losslessly, Speculative Decoding (SD) (Leviathan et al., 2023; Xia et al., 2023) has been proposed as a promising approach in Natural Language Processing (NLP). SD utilizes a draft model (*e.g.,* a compatible small-sized language model) to reduce the number of target

---

*Equal contributions.
†Corresponding authors.

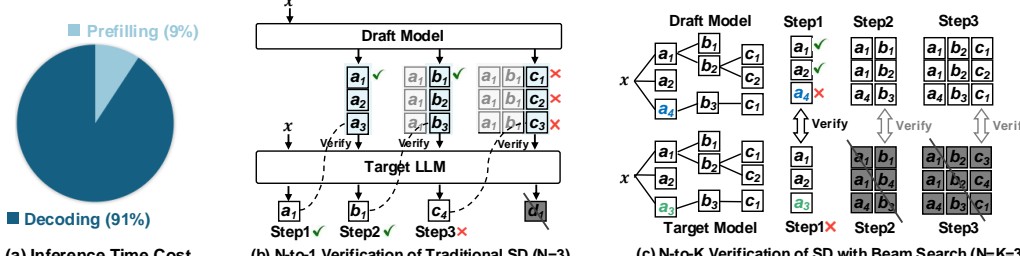

Figure 1: (a) The inference time costs of LC-Rec (Zheng et al., 2024) with LLaMA-7B on a single A5000 GPU. (b) The illustration of the $N$-to-1 verification of SD with greedy decoding in NLP tasks. (c) $N$-to-$K$ verification of SD with beam search in recommendation tasks, where the drafting length, candidate number $N$, and beam size $K$ are set at 3 for illustration.

LLM calls in the decoding process (Miao et al., 2023). Technically, SD follows a draft-then-verify paradigm (Xia et al., 2024), which first efficiently drafts multiple subsequent tokens by a draft model, and then verifies the drafted tokens in parallel by the target LLM in a single call. As depicted in Figure 1(b), tokens are either accepted or rejected, with the accepted tokens up to the first rejection step being utilized for subsequent decoding. As such, the number of LLM calls can be reduced by leveraging the accepted tokens from the draft model, thereby improving the decoding efficiency.

However, it is non-trivial to apply SD for LLM-based generative recommendation due to the challenge of more stringent $N$-to-$K$ verification. Specifically, traditional SD for NLP tasks typically follows an $N$-to-1 verification to generate only one response, which requires accepting a single token out of $N$ drafted tokens at each step (Figure 1(b)). In contrast, recommendation tasks necessitate generating top-$K$ items (*i.e.,* $K$ distinct token sequences) through beam search, resulting in an $N$-to-$K$ verification problem (Figure 1(c)). For each verification step, one LLM call can be skipped if and only if all the top-$K$ sequences are successfully drafted from the $N$ candidates. To elaborate, SD fails to reduce target LLM calls in Figure 1(c) since $a_3$ is not drafted in the first step. As such, the $N$-to-$K$ verification poses greater challenges than $N$-to-1 verification as each step requires drafting all the top-$K$ sequences.

To achieve effective SD for LLM-based generative recommendation, we formulate the SD task under the $N$-to-$K$ verification. To reduce the target LLM calls under $N$-to-$K$ verification, we consider two objectives in the drafting and verification steps of SD: 1) **top-$K$ alignment**, which aims to align the drafted sequences with the top-$K$ sequences generated by the target LLM, thereby maximizing the recall of all top-$K$ sequences; and 2) **verification relaxation**, which seeks to ease the strict matching with the top-$K$ sequences from the target LLM, enhancing the acceptance rate of drafted sequences while maintaining the accuracy of top-$K$ recommendations.

To this end, we propose an **A**lignmen**t** framework for **Spe**culativ**e d**ecoding (AtSpeed) tailored for LLM-based generative recommendation. First, under the strict top-$K$ verification, we propose an optimization objective named *AtSpeed-S* to train the draft model. AtSpeed-S improves the top-$K$ alignment theoretically by minimizing the Reverse Kullback-Leibler Divergence (RKLD) (Huszár, 2015) and a probability density regularization term (see Section 3.1). Moreover, for verification relaxation, we introduce a relaxed sampling verification strategy that allows the non-top-$K$ drafted sequences with high generation probabilities to be accepted. To maintain recommendation accuracy, this verification strategy ensures that the generation distribution of SD is approximately equivalent to that of the target LLM with sampling-based beam search (see Section 3.2). Under the relaxed sampling verification, we design *AtSpeed-R* for top-$K$ alignment, which minimizes the Total Variance Distance (TVD) on the generation probabilities of top-$K$ sequences (see Section 3.2).

We conduct extensive experiments using both verification strategies on two real-world recommendation datasets, demonstrating that AtSpeed significantly accelerates the decoding for LLM-based recommendation (around $2\times$ speedup). Besides, the results confirm that the relaxed sampling verification strategy substantially improves decoding efficiency without sacrificing much recommendation accuracy. The contributions of this work are summarized as follows:

- We are the first to propose the speculative decoding task for LLM-based recommender acceleration, highlighting the significant challenge of shifting from $N$-to-1 verification to $N$-to-$K$ verification.

---

**Algorithm 1** SD step with Top-$K$ Strict Verification

---

**Input:** Draft model $\mathcal{M}_q$, target LLM $\mathcal{M}_p$, prefix, target beam size $K$, draft beam size $N$
1: $\mathcal{Y}_0^q \leftarrow$ prefix, $\mathcal{Y}_{\gamma+1}^q \leftarrow \emptyset$, $\mathcal{Y}_{\text{out}} \leftarrow \emptyset$
2: **for** $j = 1$ to $\gamma$ **do**
3: $\quad q_j \leftarrow \mathcal{M}_q(\mathcal{Y}_{j-1}^q)$, $\mathcal{Y}_j \leftarrow \text{Top}N(q_j)$ $\qquad\qquad$ ▷ Drafting sequences for every beam search step
4: $p_1, p_2, \ldots, p_{\gamma+1} \leftarrow \mathcal{M}_p(\mathcal{Y}_0^q), \mathcal{M}_p(\mathcal{Y}_1^q), \ldots, \mathcal{M}_p(\mathcal{Y}_\gamma^q)$ $\qquad\qquad$ ▷ Run $\mathcal{M}_p$ in parallel
5: **for** $j = 1$ to $\gamma + 1$ **do**
6: $\quad \mathcal{Y}_j^p \leftarrow \text{Top}K(p_j)$, $\mathcal{Y}_{\text{out}} \leftarrow \mathcal{Y}_j^p$
7: $\quad$ **if** $\mathcal{Y}_j^p \in \mathcal{Y}_j^q$ **then**
8: $\qquad$ continue $\qquad\qquad\qquad\qquad\qquad$ ▷ Accept if ideal top-$K$ sequences are fully drafted
9: $\quad$ **else**
10: $\qquad$ break $\qquad\qquad\qquad\qquad\qquad\qquad\qquad\qquad\qquad\qquad\qquad$ ▷ Reject
**Output:** $\mathcal{Y}_{\text{out}}$

---

- We propose a novel alignment framework named AtSpeed for speculative decoding under $N$-to-$K$ verification, with a relaxed sampling verification strategy for verification relaxation and two alignment objectives for superior top-$K$ alignment of draft models.

- We conduct extensive experiments on two datasets, which 1) demonstrate the verification efficiency and accuracy of the relaxed sampling verification strategy; and 2) validate that AtSpeed achieves almost $2\times$ speedup for LLM-based recommender decoding.

## 2 TASK FORMULATION

**LLM-based Generative Recommendation.** In LLM-based generative recommendation, each item is represented by an item identifier, *i.e.,* a token sequence such as item title (Bao et al., 2023) or learnable token sequence (Zheng et al., 2024), linking the recommendation items to the language space for LLMs to understand user behaviors and recommend items (Lin et al., 2024a). Formally, given the user's historical interactions $\boldsymbol{x}$, the well-trained target LLM-based recommender model $\mathcal{M}_p$ generates top-$K$ ranked items via beam search, *i.e.,* $\{\boldsymbol{y}_{L,i}\}_{i=1}^K \leftarrow \mathcal{M}_p(\boldsymbol{x})$, where $\boldsymbol{y}_{L,i} = (y_1, y_2, \ldots, y_L)_i$ is the item identifier of the $i$-th recommended item of length $L$[1]. However, the LLM inference is time-consuming due to the need to perform multiple LLM calls (*i.e.,* forward process) during autoregressive generation. To accelerate the LLM inference, we are motivated to leverage SD for LLM-based generative recommendation for its decoding acceleration without losing accuracy.

**SD for LLM-based Recommendation.** The challenge of applying SD to LLM-based generative recommendation with beam search lies in the shift from $N$-to-1 to the harder $N$-to-$K$ verification. In the following, we detail the draft-then-verify paradigm under the strict top-$K$ verification and formulate the task of SD for LLM-based generative recommendation under $N$-to-$K$ verification.

- **Drafting**. Given the user's historical interactions $\boldsymbol{x}$ and the generated sequences in the previous SD step $\mathcal{Y}_t = \{\boldsymbol{y}_{t,i}\}_{i=1}^K$, a compatible small-sized draft model $\mathcal{M}_q$ is used to generate the drafted beam sequences for $\gamma$ steps via beam search with the beam size of $N$:

$$
\begin{aligned}
q_{t+1}, q_{t+2}, \ldots, q_{t+\gamma} &\leftarrow \text{BeamSearch}(\boldsymbol{x}, \mathcal{Y}_t, \mathcal{M}_q), \\
\mathcal{Y}_{t+1}^q, \mathcal{Y}_{t+2}^q, \ldots, \mathcal{Y}_{t+\gamma}^q &\leftarrow \text{Top}N(q_{t+1}), \text{Top}N(q_{t+2}), \cdots, \text{Top}N(q_{t+\gamma}),
\end{aligned}
\tag{1}
$$

  where $q_{t+j}$ with $j \in \{1, \ldots, \gamma\}$ is the sequence probability distribution at step $j$ obtained by the beam search of draft model; and $\mathcal{Y}_{t+j}^q$ collects the top-$N$ drafted sequences with highest probabilities. The drafted sequences are then fed into the target LLM $\mathcal{M}_p$ to obtain the target sequence probability $p_{t+1}, p_{t+2}, \ldots, p_{t+\gamma+1}$, where $p_{t+j} = \mathcal{M}_p(\mathcal{Y}_{t+j}^q)$ with $j \in \{0, \ldots, \gamma\}$. Target LLM $\mathcal{M}_p$ then select the sequences with top-$K$ probabilities as ideal sequences, *i.e.,* $\mathcal{Y}_{t+j}^p \leftarrow \text{Top}K(p_{t+j})$. For simplicity, we omit the subscript $t$ and use $q_j, p_j, \mathcal{Y}_j^q$, and $\mathcal{Y}_j^p$ as shorthand for $q_{t+j}, p_{t+j}, \mathcal{Y}_{t+j}^q$, and $\mathcal{Y}_{t+j}^p$, respectively, in contexts where the exact value of $t$ is not essential.

---

[1]We follow the widely used codebook-based item identifier, due to its promising results and generalization ability on cold-start items (Wang et al., 2024a; Rajput et al., 2023a). The codebook-based identifier will ensure each item has an identifier of the same length.

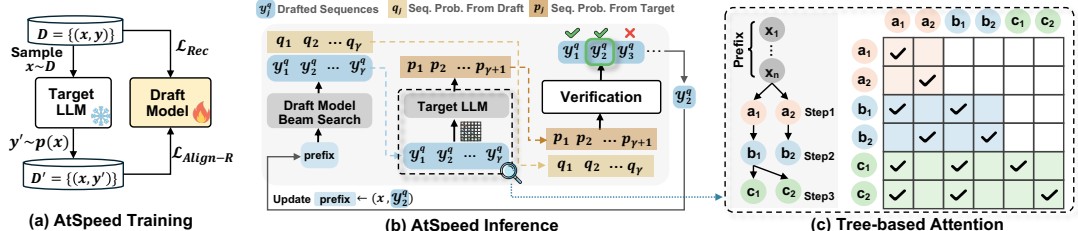

Figure 2: Overview of AtSpeed. (a) shows the alignment training of the draft model with an additional alignment loss tailored for different verification strategies, *e.g.*, $\mathcal{L}_{\text{Align-R}}$. (b) depicts the AtSpeed inference, where the well-trained draft model produces beam search sequences from each step, *i.e.*, $\mathcal{Y}_j^q$, for the target LLM to verify. The beam sequences from the last accepted step before encountering the first rejection are utilized in the following SD step, *i.e.*, $\mathcal{Y}_2^q$. (c) demonstrates the tree-based attention for the drafted beam sequences ($N=2$ and $\gamma=3$).

- **Strict Top-$K$ Verification.** For every beam search step $j$, we have $N$ drafted sequences $\mathcal{Y}_q$[2] and $K$ ideal sequences $\mathcal{Y}_p$. We define the *step acceptance* as successfully drafting all the top-$K$ ideal sequences for that step. Formally, the beam search step is accepted if $\mathcal{Y}_p \in \mathcal{Y}_q$. Otherwise, we reject the step, and discard the whole drafted sequences $\mathcal{Y}_q$ and correct it with $\mathcal{Y}_p$. From beam search step $j = 1$ to $j = \gamma$, we sequentially verify $\mathcal{Y}_q$ until the first rejection step occurs. The corrected sequences $\mathcal{Y}_p$ at the first rejection step will be the output of the current SD step. If every step is successfully accepted, we select top-$K$ sequences from $p_{\gamma+1}$ as the output of the current SD step. The target LLM calls for all previously accepted steps could be reduced as they are successfully skipped through a single LLM call via parallel verification. The process of strict top-$K$ verification is presented in Algorithm 1.

It is highlighted that SD for beam search under $N$-to-$K$ verification is more difficult than $N$-to-1 verification because an LLM call can be skipped if and only if the top-$K$ sequences are entirely drafted. To improve the acceptance rate of the drafted sequences to reduce the LLM calls, we have two key considerations: 1) top-$K$ alignment, which encourages the draft model to generate strongly aligned top-$K$ sequences; and 2) verification relaxation, which seeks to relax the strict matching for greater acceleration without much accuracy sacrifice.

## 3 ATSPEED

To pursue the two objectives, we propose AtSpeed, an alignment framework for SD under $N$-to-$K$ verification. AtSpeed designs two effective alignment objectives for the draft model to get a higher acceptance rate under the strict top-$K$ verification (Section 3.1) and the proposed novel relaxed verification (Section 3.2). The overview of AtSpeed is presented in Figure 2.

### 3.1 ALIGNMENT FOR STRICT TOP-K VERIFICATION (ATSPEED-S)

Under strict top-$K$ verification, the draft model is expected to generate top-$K$ sequences that strictly align with the top-ranked sequences from the target LLM. To achieve this, we design an optimization objective named AtSpeed-S that directly optimizes the acceptance rate for the strict top-$K$ verification.

**Acceptance Rate.** We define the acceptance rate $\beta$ as the probability of the step acceptance, *i.e.*, accepting all the top-$K$ ideal sequences from the draft sequences $\mathcal{Y}_q$. With strict top-$K$ verification strategy, for each step, we have $\beta = 1$ if $\exists \mathcal{Y}'_q \subseteq \mathcal{Y}_q$ such that $p(\boldsymbol{y}) \geq p(\boldsymbol{y}_K)$ for $\forall \boldsymbol{y} \in \mathcal{Y}'_q$, where $|\mathcal{Y}'_q| = K$, and $\boldsymbol{y}_K$ is the sequence that has the $K$-th highest probability in $p$.

**Alignment Objective.** Since the acceptance rate directly affects the acceleration performance, we aim to optimize the acceptance rate of the draft model to achieve superior top-$K$ alignment with the target LLM. To elaborate, we consider the following alignment objective for the draft model $\mathcal{M}_q$:

$$\theta^* := \arg\max_{\theta \in \Theta} \quad \mathbb{E}_{\boldsymbol{y} \sim \mathcal{Y}_{q\theta}} \frac{p(\boldsymbol{y})}{p(\boldsymbol{y}_K)} = \arg\max_{\theta \in \Theta} - \sum_{\boldsymbol{y} \in \mathcal{Y}_{q\theta}} q_\theta(\boldsymbol{y}) log \frac{q_\theta(\boldsymbol{y})}{p(\boldsymbol{y})} + \sum_{\boldsymbol{y} \in \mathcal{Y}_{q\theta}} q_\theta(\boldsymbol{y}) \log \frac{q_\theta(\boldsymbol{y})}{p(\boldsymbol{y}_K)}, \quad (2)$$

---

[2]We denote $\mathcal{Y}_j^p$ and $\mathcal{Y}_j^q$ with $\mathcal{Y}_p$ and $\mathcal{Y}_q$, respectively, whenever step $j$ is clear.

where $\mathcal{Y}_{q_\theta} = \mathcal{Y}_1^q \cup \mathcal{Y}_2^q \cdots \cup \mathcal{Y}_L^q$ aims at maximizing acceptance rate for every beam search step (see detailed derivatives in Appendix A.2.2). This alignment objective can be further expanded as:

$$\theta^* := \arg\min_{\theta \in \Theta} \mathbb{E}_{(\boldsymbol{x},\mathcal{Y}) \sim \mathcal{D}'} \sum_{\boldsymbol{y} \in \mathcal{Y}} \Big[ \frac{1}{|\boldsymbol{y}|} \sum_{t=1}^{|\boldsymbol{y}|} \Big[ \sum_{y_t \in \mathcal{V}} q_\theta(y_t|c_{<t}) \log \frac{q_\theta(y_t|c_{<t})}{p(y_t|c_{<t})} - \sum_{y_t \in \mathcal{V}} q_\theta(y_t|c_{<t}) \log \frac{q_\theta(y_t|c_{<t})}{p(y_K)} \Big] \Big],$$
(3)

where $c_t = (\boldsymbol{x}, \boldsymbol{y}_{<t})$ is the context, $p(y_K) = p(\boldsymbol{y}_{K,t}|\boldsymbol{y}_{K,<t})$, and $\mathcal{D}' = \{(\boldsymbol{x}, \mathcal{Y} = \text{Top}K((1 - \lambda)\mathcal{M}_q(\boldsymbol{x}) + \lambda\mathcal{M}_p))|\boldsymbol{x} \sim \mathcal{D}\}$[3], and $\mathcal{V}$ is the LLM vocabulary.

However, aligning every token over the entire vocabulary with $y_t \in \mathcal{V}$ might lead to suboptimal results due to the noises introduced by invalid tokens with high probabilities. LLM-based recommender models usually employ constrained generation (Hua et al., 2023) to generate valid item identifiers only. Blindly aligning across all tokens in $\mathcal{V}$ may cause unnecessary alignment to these high-probability but invalid tokens that will never be generated by the target LLM. To mitigate this issue, we define $\mathcal{V}_c = \text{ConstrainedTop}K(q)$ to block out the potential alignment noises from invalid tokens. We then define the alignment loss as:

$$\mathcal{L}_{\text{Align-S}} = \frac{1}{|\mathcal{D}'|} \sum_{(\boldsymbol{x},\mathcal{Y}) \sim \mathcal{D}'} \sum_{\boldsymbol{y} \in \mathcal{Y}} \Big[ \frac{1}{|\boldsymbol{y}|} \sum_{t=1}^{|\boldsymbol{y}|} \Big[ \underbrace{\sum_{y_t \in \mathcal{V}_c} q_\theta(y_t|c_{<t}) \log \frac{q_\theta(y_t|c_{<t})}{p(y_t|c_{<t})}}_{(\textbf{RKLD})} - \underbrace{\sum_{y_t \in \mathcal{V}_c} q_\theta(y_t|c_{<t}) \log \frac{q_\theta(y_t|c_{<t})}{p(y_K)}}_{(\textbf{Density Regularization})} \Big] \Big],$$
(4)

which is essentially minimizing the RKLD over the top-$K$ sequence probabilities and a density regularization term. Intuitively, minimizing RKLD emphasizes aligning the draft model $\mathcal{M}_q$ to the target LLM $\mathcal{M}_p$ within the draft model's generation capability (Gu et al., 2024), particularly for the top-$K$ sequence probability distribution. Moreover, the density regularization term encourages the top-$K$ sequence probabilities of the draft model to dominate the whole probability distribution.

**Overall Loss.** Based on Eq.(4), AtSpeed-S defines the overall training loss for the draft model as:

$$\mathcal{L}_{\text{AtSpeed-S}} = \alpha \mathcal{L}_{\text{Align-S}} + (1 - \alpha)\mathcal{L}_{\text{Rec}},$$
(5)

where $\mathcal{L}_{\text{Rec}}$ is the original loss to train the model for the recommendation tasks (see Appendix A.3.1) and $\alpha$ is the hyper-parameter to control the alignment strength.

### 3.2 ALIGNMENT FOR RELAXED SAMPLING VERIFICATION (ATSPEED-R)

In addition to the strengthened alignment for the strict top-$K$ verification, we further consider verification relaxation to reduce the trivial LLM calls. From the angle of verification, we introduce a relaxed sampling verification strategy based on the sequence generation probability to improve the acceptance rate of draft sequences. This verification strategy ensures that sequences falling outside the target's top-$K$ predictions can still be accepted with a certain probability, while preserving the output distribution closely aligned with that of the target LLM to maintain the recommendation accuracy to some extent. Under this relaxed sampling verification strategy, we propose AtSpeed-R for top-$K$ alignment accordingly.

**Relaxed Sampling Verification**. We design the verification strategy for each beam search step $j \in \{1, \ldots, \gamma\}$ as follows. Given the drafted results $\mathcal{Y}_q \sim q$[4], for each $\boldsymbol{y} \in \mathcal{Y}_q$, we accept $\boldsymbol{y}$ if $p(\boldsymbol{y}) \geq q(\boldsymbol{y})$. Intuitively, if the target LLM is even more confident in generating a high-probability sequence, the candidate sequence is likely to be generated by the LLMs and should be accepted. Otherwise, we reject $\boldsymbol{y}$ with probability of $1 - \frac{p(\boldsymbol{y})}{q(\boldsymbol{y})}$ and resample $\boldsymbol{y} \sim p' = \text{norm}(\max(0, p(\boldsymbol{y}) - q(\boldsymbol{y})))$. Notably, for the non-top-$K$ drafted sequence $\boldsymbol{y}$ with a high draft probability (*i.e.*, $q(\boldsymbol{y}) \geq p(\boldsymbol{y})$), this verification strategy ensures they have a certain probability of $\frac{p(\boldsymbol{y})}{q(\boldsymbol{y})}$ to be accepted. We present more details on the relaxed sampling verification strategy in Algorithm 2 in Appendix. Drawing upon the with-replacement sampling approximation[5], the joint distribution of $K$ output sequences under such

---

[3]We follow (Gu et al., 2024) to adopt the target LLM-mixed sampling for data construction to improve training efficiency and generated data quality.

[4]The drafted sequences are sampled from the sequence distribution $q_j$, and superscript $j$ is omitted for clarity.

[5]When the sample size (*i.e.,* the number of recommended items $K$) is much smaller than the total population size (*i.e.,* all sequences for sampling), the sampling without replacement can be effectively approximated to the sampling with replacement. See Appendix A.2.1 for proof.

verification is approximately equivalent to that of the target LLM as proven in Appendix A.2.1. Such equivalence ensures that the relaxed sampling verification does not sacrifice much recommendation accuracy (see Section 4.2 for empirical evidence).

**Acceptance Rate.** For the relaxed sampling verification strategy, we obtain the probability of step acceptance $\beta = \prod_K b$ under the with-replacement sampling approximation, where

$$b = \mathbb{E}_{\boldsymbol{y} \sim q(\boldsymbol{y})} \begin{cases} 1, & q(\boldsymbol{y}) \leq p(\boldsymbol{y}) \\ \frac{p(\boldsymbol{y})}{q(\boldsymbol{y})}, & q(\boldsymbol{y}) > p(\boldsymbol{y}) \end{cases} = \mathbb{E}_{y \sim q(\boldsymbol{y})} \min(1, \frac{p(\boldsymbol{y})}{q(\boldsymbol{y})}) = 1 - \text{TVD}(p(\boldsymbol{y}), q(\boldsymbol{y})). \tag{6}$$

The derivation of Eq.(6) can be easily extended from (Leviathan et al., 2023) as in Appendix A.2.3.

**Alignment Objective.** Under the relaxed sampling verification, we consider maximizing $\log \beta = -\sum_K \log \text{TVD}(q, p)$, which is equivalent to minimizing $\sum_K \text{TVD}(q, p)$ to improve the acceptance rate. Thereafter, we present the alignment objective for the draft model:

$$\theta^* := \arg\min_{\theta \in \Theta} \mathbb{E}_{(\boldsymbol{x}, \boldsymbol{y}) \sim \mathcal{D}'} K \cdot \Big[ \frac{1}{|\boldsymbol{y}|} \sum_{t=1}^{|\boldsymbol{y}|} \text{TVD}(p'(\cdot|\boldsymbol{x}, \boldsymbol{y}_{<t}), q'(\cdot|\boldsymbol{x}, \boldsymbol{y}_{<t})) \Big]. \tag{7}$$

Similar to AtSpeed-S, we perform constrained alignment over the top-$K$ valid tokens, and therefore $p'(\cdot|\boldsymbol{x}, \boldsymbol{y}_{<t})$ and $p'(\cdot|\boldsymbol{x}, \boldsymbol{y}_{<t})$ is the normalized truncated sequence probability over top-$K$ valid tokens. Based on the with-replacement sampling approximation, Eq.(7) minimizes TVD with the strength of $K$ for the same sequence. Nevertheless, considering the beam search with sampling would obtain $K$ different sequences, we alternatively leverage the top-$K$ sequences from the target LLM, *i.e.,* $\mathcal{D}' = \{(\boldsymbol{x}, \boldsymbol{y} \sim \text{Top}K(\mathcal{M}_p(\boldsymbol{x})))|\boldsymbol{x} \sim \mathcal{D}\}$ to minimize the TVD. Finally, we define the alignment loss of AtSpeed-R as,

$$\mathcal{L}_{\text{Align-R}} = \frac{1}{|\mathcal{D}'|} \sum_{(\boldsymbol{x}, \boldsymbol{y}) \sim \mathcal{D}'} \Big[ \frac{1}{|\boldsymbol{y}|} \sum_{t=1}^{|\boldsymbol{y}|} TVD(p'(\cdot|\boldsymbol{x}, \boldsymbol{y}_{<t}), q'(\cdot|\boldsymbol{x}, \boldsymbol{y}_{<t})) \Big]. \tag{8}$$

**Overall Loss.** The overall loss of AtSpeed-R is defined as:

$$\mathcal{L}_{\text{AtSpeed-R}} = \alpha \mathcal{L}_{\text{Align-R}} + (1 - \alpha) \mathcal{L}_{\text{Rec}}, \tag{9}$$

where $\alpha$ controls the strength of the alignment loss.

## 3.3 INFERENCE OF ATSPEED

The inference of AtSpeed follows the draft-then-verify paradigm. To efficiently recommend items during inference, AtSpeed first leverages a well-aligned draft model for drafting and then chooses a specific strategy for verification. Specifically, 1) to obtain identical recommendation results, AtSpeed can utilize the draft model trained by AtSpeed-S and adopt the strict top-$K$ verification strategy for SD. Alternatively, 2) to further improve the inference speedup, AtSpeed can use the draft model trained by AtSpeed-R along with the corresponding relaxed sampling verification strategy. Note that any draft model can be interchangeably applied to the strict top-$K$ and relaxed sampling verification strategies, although they may not be optimized for enhancing the acceptance rate under the corresponding verification strategy.

**Tree-based Attention.** A challenge of AtSpeed inference is the verification inefficiency issue. Using beam search results from every step would largely increase the number of drafted sequences ($\gamma N$ sequences), where some prefix is shared by different beam sequences. As such, self-attention is repeatedly calculated for the same prefix through a single LLM call, thus leading to verification inefficiency, especially with a larger $N$ (*e.g.,* $N = 20$). In this work, we leverage the tree-based attention (Miao et al., 2024) to eliminate the repeated calculations via a single flattened sequence, thereby boosting the efficiency of verification. The tree-based attention for the drafted sequences from all steps is illustrated in Figure 2(c) and implementation details are presented in Appendix A.3.1.

## 4 EXPERIMENTS

In this section, we conduct extensive experiments to answer the following research questions:

1) **RQ1**: How does our proposed AtSpeed perform on LLM-based recommendation under strict top-$K$ and relaxed sampling verification strategies? 2) **RQ2**: How do different components of AtSpeed affect the decoding acceleration? 3) **RQ3**: How do different hyper-parameters affect the performance?

## 4.1 Experimental Settings

**Datasets and Baselines.** To evaluate our proposed framework, we instantiate AtSpeed on a SOTA LLM-based generative recommender model LC-Rec (Zheng et al., 2024) and test on two real-world recommendation datasets[6] from the popular benchmark Amazon review datasets[7]. 1) **Beauty** contains user interactions with the beauty products and 2) **Games** collects the user interactions with the video games. For both datasets, each item has rich textual information such as title and description. Based on the item textual information, we follow LC-Rec to assign each item with an item identifier generated by RQ-VAE (Lee et al., 2022). More details of the datasets can be found in Appendix A.3.1.

We compare our proposed alignment methods with Supervised Fine-tuning (SFT) and several representative Knowledge Distillation (KD) methods as follows: 1) **SFT** fine-tunes the draft model with the recommendation loss $\mathcal{L}_{Rec}$ on the recommendation dataset; 2) **WordKD** (Sanh, 2019; Hinton, 2015) fine-tunes the draft model to align with the token probability of the target LLM by KLD on the recommendation dataset; 3) **SeqKD** (Kim & Rush, 2016) additionally utilizes the top-$K$ target LLM-generated data to fine-tune the draft model with the recommendation loss $\mathcal{L}_{Rec}$; 4) **TVDKD** (Wen et al., 2023) aligns the draft model and the target LLM with the symmetric divergence TVD. We also extend a retrieval-based drafting method 5) **DARE** (Xi et al., 2024) for a comprehensive comparison.

**Evaluation Metrics.** To measure the decoding efficiency, we use walltime speedup@$K$ (WS@$K$) compared to the standard target LLM inference for evaluation. In addition, we report accept step@$K$ (AS@$K$), which is defined as the average number of accepted steps during the SD for each user. As for the recommendation performance, we adopt the widely used metrics Recall@$K$ and NDCG@$K$ for evaluation.

**Implementation Details.** We instantiate AtSpeed on LC-Rec, where the backend LLM is LLaMA-7B (Touvron et al., 2023). For each dataset, we fine-tune the target LLM using the recommendation loss $\mathcal{L}_{Rec}$ with the parameter-efficient fine-tuning technique LoRA (Hu et al., 2021). As for the draft model, we fully fine-tune a compatible small-sized model LLaMA-68M (Miao et al., 2024) for alignment training, and then utilize the fine-tuned model for SD inference. We set draft length $\gamma = 4$, number of recommended items $K = \{1, 3, 5, 10, 20\}$, and draft beam size $N = 40$. More implementation details are provided in Appendix A.3.1.

## 4.2 Overall Performance (RQ1)

We evaluate our method under both standard beam search (beam size=$K$) with strict top-$K$ verification and sampling-based beam search (temperature=1) with relaxed sampling verification. The acceleration results of the baselines and AtSpeed on two datasets are presented in Table 1. We can observe that:

- The walltime speedup under the relaxed sampling verification is generally better than the strict top-$K$ verification, validating the effectiveness of our proposed relaxed strategy. It boosts the speedup by allowing non-top-$K$ sequences to be accepted with a certain probability.

- Compared to SFT, the logit-level KD methods (WordKD and TVDKD) often yield comparable or even inferior performance. However, this is not surprising because the alignment task for the logit-level KD is more difficult due to two possible reasons. 1) While SFT solely focuses on fitting the top-1 probability token, logit-level KD requires alignment across the entire vocabulary, making it more difficult to achieve due to the limited expressiveness of the smaller model (Agarwal et al., 2024). 2) The alignment on the top-$K$ ranked items might be negatively affected by the invalid tokens with high probabilities. During item generation, certain tokens with high probabilities will never be generated as they are deemed invalid tokens, and thereby will never be accepted by the target LLM (*cf.* Section 3.1).

- Among the baselines, SeqKD yields competitive results in most cases. This is reasonable since SeqKD 1) aligns the draft model with the target LLM at the sequence level, which fosters the generation of sequences more aligned with the target LLM's distribution (superior performance than WordKD); and 2) fine-tunes the draft model with target LLM-generated data, which implicitly incorporates the information of valid tokens in item identifiers, thereby alleviating the negative

---

[6]We also evaluate our methods on MovieLens-1M and Goodreads datasets (refer to Table 6 in Appendix A.3.2).

[7]https://cseweb.ucsd.edu/~jmcauley/datasets/amazon_v2/.

Table 1: Overall comparison of walltime speedup (WS@$K$) and accept steps (AS@$K$) between the baselines and AtSpeed instantiated on LC-Rec (LLaMA-7B) across two datasets. "Avg" is calculated across $K$ from 1 to 20 (full results are presented in Table 4 in Appendix A.3.2). For each verification strategy, the best results are highlighted in bold and the second-best results are underlined.

| Beauty | | | | | | | | | | |
|---|---|---|---|---|---|---|---|---|---|---|
| Verification | Method | WS@5 | WS@10 | WS@20 | Avg WS | AS@5 | AS@10 | AS@20 | Avg AS | Recall@5 | NDCG@5 |
| **Strict Top-$K$** | DARE | 1.06 | 1.15 | 1.48 | 1.18 | 0.26 | 0.05 | 0.00 | 0.31 | 0.0056 | 0.0051 |
| | SFT | 1.43 | 1.37 | 1.55 | 1.56 | 1.32 | 0.66 | 0.09 | 1.18 | 0.0056 | 0.0051 |
| | WordKD | 1.58 | 1.52 | 1.58 | 1.68 | 1.60 | 1.03 | 0.16 | 1.40 | 0.0056 | 0.0051 |
| | TVDKD | 1.44 | 1.37 | 1.57 | 1.55 | 1.31 | 0.65 | 0.09 | 1.17 | 0.0056 | 0.0051 |
| | SeqKD | 1.75 | 1.67 | 1.68 | 1.83 | 1.85 | 1.27 | 0.30 | 1.60 | 0.0056 | 0.0051 |
| | AtSpeed-S | **1.84** | **1.87** | **1.84** | **1.97** | **2.00** | **1.64** | **0.57** | **1.80** | 0.0056 | 0.0051 |
| | AtSpeed-R | 1.70 | 1.71 | 1.74 | 1.76 | 1.82 | 1.33 | 0.43 | 1.56 | 0.0056 | 0.0051 |
| **Relaxed Sampling** | DARE | 1.70 | 1.53 | 1.69 | 1.73 | 1.97 | 1.14 | 1.00 | 1.62 | 0.0059 [↑0.0003] | 0.0044 [↓0.0007] |
| | SFT | 1.80 | 2.06 | 2.36 | 1.95 | 2.03 | 1.99 | 1.48 | 1.94 | 0.0057 [↑0.0001] | 0.0041 [↓0.0010] |
| | WordKD | 1.81 | 1.99 | 2.05 | 1.87 | 2.01 | 1.87 | 1.07 | 1.82 | 0.0066 [↑0.0010] | 0.0043 [↓0.0008] |
| | TVDKD | 1.81 | 2.06 | 2.35 | 1.96 | 2.03 | 1.99 | 1.45 | 1.94 | 0.0057 [↑0.0001] | 0.0045 [↓0.0006] |
| | SeqKD | 1.90 | 2.11 | 2.31 | 2.01 | **2.10** | 2.01 | 1.40 | 1.97 | 0.0055 [↓0.0001] | 0.0045 [↓0.0006] |
| | AtSpeed-S | 1.89 | 2.12 | **2.51** | 2.07 | 2.09 | **2.03** | 1.71 | 2.05 | 0.0060 [↑0.0004] | 0.0046 [↓0.0005] |
| | AtSpeed-R | **1.94** | **2.16** | 2.47 | **2.11** | 2.13 | 2.01 | **1.77** | **2.10** | 0.0058 [↑0.0002] | 0.0049 [↓0.0002] |

| Games | | | | | | | | | | |
|---|---|---|---|---|---|---|---|---|---|---|
| Verification | Method | WS@5 | WS@10 | WS@20 | Avg WS | AS@5 | AS@10 | AS@20 | Avg AS | Recall@5 | NDCG@5 |
| **Strict Top-$K$** | DARE | 0.99 | 1.13 | 1.44 | 1.09 | 0.00 | 0.00 | 0.00 | 0.03 | 0.0074 | 0.0065 |
| | SFT | 1.43 | 1.40 | 1.58 | 1.53 | 1.32 | 0.91 | 0.15 | 1.20 | 0.0074 | 0.0065 |
| | WordKD | 1.31 | 1.35 | 1.47 | 1.48 | 1.10 | 0.80 | 0.14 | 1.13 | 0.0074 | 0.0065 |
| | TVDKD | 1.24 | 1.32 | 1.50 | 1.42 | 0.95 | 0.66 | 0.09 | 0.99 | 0.0074 | 0.0065 |
| | SeqKD | 1.60 | 1.46 | **1.77** | 1.71 | 1.67 | 1.05 | **0.76** | 1.55 | 0.0074 | 0.0065 |
| | AtSpeed-S | **1.78** | **1.85** | 1.76 | **1.83** | **1.96** | **1.69** | 0.68 | **1.72** | 0.0074 | 0.0065 |
| | AtSpeed-R | 1.76 | 1.76 | 1.60 | 1.74 | 1.95 | 1.53 | 0.32 | 1.59 | 0.0074 | 0.0065 |
| **Relaxed Sampling** | DARE | 1.68 | 1.19 | 1.42 | 1.51 | 1.96 | 0.37 | 0.00 | 1.27 | 0.0076 [↑0.0002] | 0.0065 [↑0.0000] |
| | SFT | 1.84 | 1.97 | 1.69 | 1.86 | 2.05 | 1.89 | 0.58 | 1.78 | 0.0073 [↓0.0001] | 0.0060 [↓0.0005] |
| | WordKD | 1.78 | 1.84 | 1.56 | 1.76 | 1.99 | 1.68 | 0.25 | 1.63 | 0.0072 [↓0.0002] | 0.0058 [↓0.0007] |
| | TVDKD | 1.81 | 1.90 | 1.55 | 1.80 | 2.02 | 1.80 | 0.29 | 1.69 | 0.0069 [↓0.0005] | 0.0061 [↓0.0004] |
| | SeqKD | 1.90 | 2.03 | 2.05 | 1.95 | **2.10** | 1.98 | 1.22 | 1.93 | 0.0071 [↓0.0003] | 0.0059 [↓0.0006] |
| | AtSpeed-S | 1.91 | 2.04 | 2.13 | 2.04 | 2.10 | 1.98 | 1.28 | 2.00 | **0.0080** [↑0.0006] | **0.0068** [↑0.0003] |
| | AtSpeed-R | **2.00** | **2.05** | **2.20** | **2.05** | **2.17** | **1.98** | **1.35** | **2.02** | 0.0076 [↑0.0002] | 0.0063 [↓0.0002] |

effect from the noisy invalid tokens (superior performance than TVDKD). The competitiveness of SeqKD is also consistent with the observations in NLP tasks (Agarwal et al., 2024). The less satisfying performance of DARE is reasonable since the uniformly retrieved valid sequences might not be aligned with the target LLM specifically on top-$K$ sequences.

- AtSpeed significantly accelerates the LLM-based recommender inference under both strict top-$K$ and relaxed sampling verifications, showing the superiority of our proposed method for strong top-$K$ alignment. In particular, AtSpeed-S under strict top-$K$ verification achieves an average speedup of $1.97\times$ on Beauty and $1.83\times$ on Games while obtaining the identical results of the top-$K$ recommended items. Moreover, AtSpeed-R further enhances the efficiency with an average speedup of $2.11\times$ and $2.05\times$ on Beauty and Games, respectively. Furthermore, as $K$ increases, the number of accept steps gradually decreases due to the increased difficulty of the $N$-to-$K$ verification. Nevertheless, it is highlighted that the speedup for $K = 20$ is still comparable to that for $K = 5$ and outperforms the speedup for $K = 10$. This is attributed to tree-based attention, which efficiently verifies the drafted sequences (more detailed analysis in Appendix A.3.2).

- The ranking performance under strict top-$K$ verification is lossless because we only accept the perfectly matched top-$K$ beam sequences to obtain identical recommendation results. For relaxed sampling verification, the ranking performance across different alignment methods only shows limited performance drops compared to the target LLM's top-$K$ results, *i.e.,* performance under strict top-$K$ verification. This meets our expectations since the sampling-based verification ensures the approximately equivalent distribution between the SD output and target LLM output (refer to Appendix A.3.2 for additional accuracy results and analysis).

### 4.3 IN-DEPTH ANALYSIS

**Ablation Study (RQ2).** To analyze the effect of AtSpeed-S, we remove the density regularization term ("w/o DR"), constrained alignment ("w/o CA"), and tree-based attention ("w/o TA") for AtSpeed-S. These variants of AtSpeed-S are evaluated under the strict top-$K$ verification. For AtSpeed-R, we replace the top-$K$ sequences with the top-1 sequence from target LLM (denoted as "w/o top-$K$"), and separately remove the constrained alignment ("w/o CA") and the tree-based attention ("w/o TA"). These variants are evaluated under the relaxed sampling verification.

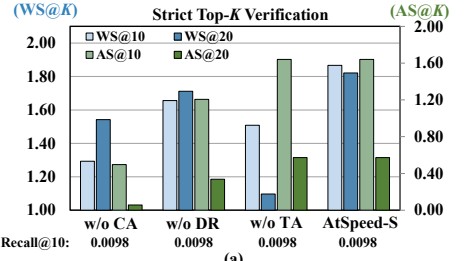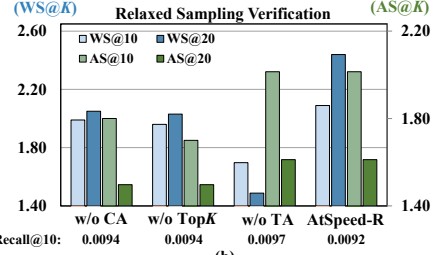

Figure 3: Ablation study of AtSpeed-S and AtSpeed-R on Beauty dataset.

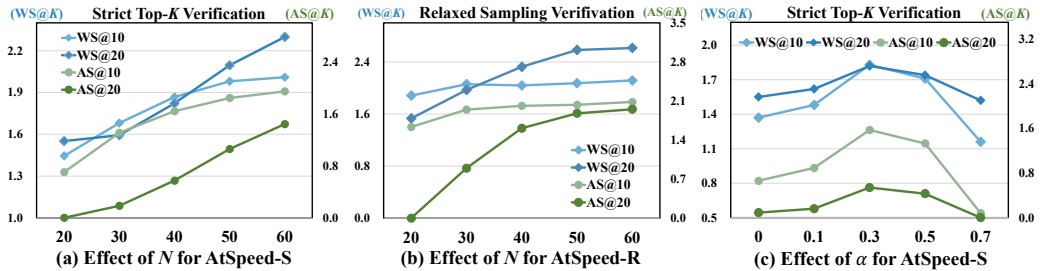

Figure 4: Effect of draft beam size $N$ and the alignment strength $\alpha$ on Beauty dataset.

From the results as shown in Figure 3, we have the following observations. 1) the performance decline of "w/o DR" indicates that aggregating probability around the top-$K$ tokens can effectively strengthen the alignment. 2) The use of top-1 sequence from the target LLM to execute AtSpeed-R results in a lower acceptance rate. This makes sense since the top-1 sequence lacks diversity, which might overemphasize the alignment on the top-1 sequence and hurt the speedup. 3) By discarding either the constrained alignment or tree attention would lead to performance decline for both AtSpeed-S and AtSpeed-R. This is expected since removing constrained alignment would introduce potential noisy tokens, which have high probability yet will be rejected to ensure in-corpus item recommendation. Furthermore, 4) the removal of tree-based attention would hurt the speedup although it does not affect the acceptance rate, which validates the efficiency of utilizing tree-based attention for verification.

**Effect of Draft Beam Size $N$ (RQ3).** To study how the draft beam size $N$ affects the acceleration of AtSpeed, we vary $N$ from 20 to 60 and test AtSpeed-S under strict top-$K$ verification (Figure 4) and AtSpeed-R under relaxed sampling verification, respectively. From the figures, we can observe that 1) from $N = 20$ to $N = 60$, the acceptance rate constantly increases for both AtSpeed-S and AtSpeed-R. This is reasonable as a larger beam size can produce more drafted sequences for verification, thus enhancing the acceptance rate. Moreover, 2) the walltime speedup continues to improve steadily as $N$ gets larger. We attribute this phenomenon to the utilization of tree-based attention, which efficiently verifies drafted sequences with a single flattened sequence (refer to Appendix A.3 for detailed speedup analysis of tree-based attention). Nevertheless, 3) the improvement of acceptance rate and speedup gradually slows down (*i.e.,* decreasing slope), indicating continuously increasing $N$ is approaching a balance between the enhanced acceptance rate and the increased overhead.

**Effect of Alignment Strength $\alpha$ (RQ3).** We train the draft model with different alignment strength $\alpha$ and present the results of AtSpeed-S in Figure 4(c)[8]. From the figure, it is observed that the integration of alignment loss (incrementally increased from 0) improves performance, which validates the necessity of alignment to improve acceptance rate. However, 2) blindly emphasizing the alignment loss may be ineffective, as the model struggles to capture the output of the target LLM due to the limited expressiveness of the small-sized model without the knowledge of recommendation task. Therefore, it is essential to strike a balance between the task-specific objective and the alignment objective. Empirically, we recommend setting $\alpha = 0.5$ for alignment training.

## 5 RELATED WORK

**Speculative Decoding**. As an efficient LLM decoding technique, SD (Leviathan et al., 2023; Fu et al., 2024; Du et al., 2024; Liu et al., 2024c; Chen et al., 2024; Sun et al., 2024) has been proposed

---

[8]More analysis of AtSpeed is presented in Appendix A.3.2 to save space.

to reduce the LLM calls by efficiently drafting multiple future tokens and then verifying them in parallel. Early studies focus on SD in greedy decoding, where the draft model generates one token at each generation step for LLM to verify if they match the LLM's top-1 token. Later on, some studies extend SD to nucleus sampling to accelerate LLM decoding without changing its output distribution (Leviathan et al., 2023). However, the early attempts simply draft one token at each step for verification, which significantly limits the acceptance rate (Xia et al., 2023). To improve the acceptance rate, subsequent work designs various relaxed verification strategies for greedy decoding such as enlarging LLM candidates with a tolerable score away from the top-1 token (Xia et al., 2023) and measuring the distance of the probability between the draft model and the target LLM (Kim et al., 2024). Concurrently, some studies propose different methods to draft multiple candidates for different sampling, as seen in Medusa (Cai et al., 2024), EAGLE (Li et al., 2024a) and so on. While previous work primarily applies SD for NLP tasks and follows the $N$-to-1 verification, our work extends SD to LLM-based generative recommendation tasks, addressing the unique challenge of $N$-to-$K$ verification, *i.e.,* perfect matching over the top-$K$ sequences. Closely related to our work, DistillSpec (Zhou et al., 2024) investigates the use of various KD techniques to enhance the alignment between the draft model and the target LLM, to achieve high acceptance rate. However, they mainly focus on how KD helps for the $N$-to-1 verification. In contrast, this work studies the SD under $N$-to-$K$ verification for recommendation tasks, and proposes a framework to align the draft model with the target LLM on the top-$K$ sequences specifically for recommendation tasks. While another conceptually similar work SpecGR (Ding et al., 2024) drafts unseen items for cold-start recommendations, this work aims at drafting beam sequences at every step for inference acceleration.

**Inference Acceleration for LLM-based Recommendation.** While LLM-based generative recommendations have shown remarkable performance (Liu et al., 2024a; Xu et al., 2024a; Zhang et al., 2024a; Lv et al., 2025; Lin et al., 2024b; Zhao et al., 2025) compared to traditional recommender models (Liu et al., 2024b; Gao et al., 2023; Li et al., 2024b; Zhang et al., 2021), their practical implementation is impeded due to the high time latency of LLM decoding. Consequently, accelerating LLM decoding is imperative to facilitate real-world deployment of the powerful LLM-based recommender models. To tackle this issue, several approaches leverage KD to transfer knowledge from a teacher LLM to either a smaller language model (Xu et al., 2024b; Wang et al., 2024b) or a traditional recommender model (Cui et al., 2024). Additionally, some methods focus on designing more lightweight model architectures, such as narrow and deep transformer structures (Mei & Zhang, 2023). However, the application of SD is underexplored in LLM-based recommendation. DARE (Xi et al., 2024) is a relevant study that applies SD in user feature augmentation for LLM-based recommendation. However, they focus on the LLM decoding for user feature generation that requires only one response from the target LLM, which is essentially SD under the $N$-to-1 verification. In contrast, our work explores decoding acceleration for top-$K$ ranked item recommendation, tackling the more difficult $N$-to-$K$ verification problem. In addition to these approaches, various other techniques, such as model quantization (Xiao et al., 2023), model pruning (Hassibi & Stork, 1992), and Key-Value (KV) cache optimization (Zhang et al., 2024b), can also enhance inference acceleration. Nonetheless, these methods are orthogonal to SD and can be effectively integrated with SD to further improve decoding efficiency.

## 6    Conclusion and Future Work

In this work, we formulated the speculative decoding task for LLM-based generative recommendation, highlighting the significant challenge of shifting from $N$-to-1 to $N$-to-$K$. To effectively accelerate decoding under $N$-to-$K$ verification, the key lies in drafting all top-$K$ sequences from the target LLM, which drives two objectives for SD: top-$K$ alignment and verification relaxation. To achieve this, we proposed an alignment framework called AtSpeed, along with a relaxed sampling verification strategy. AtSpeed designs two alignment objectives for the strict top-$K$ verification and relaxed sampling verification strategies. Extensive experiments show that AtSpeed achieves around $2\times$ speedup on two real-world recommendation datasets. As an initial attempt work, it points out many promising directions for further explorations, such as 1) more efficient drafting methods to reduce the overhead of drafting, such as non-LLM recommender models for efficient high-quality drafting that might lead to better recommendation accuracy under the relaxed sampling verification strategy, and 2) integration with other acceleration techniques such as memory access optimization.

## ETHICS STATEMENT

Our research focuses on effectively boosting the inference efficiency of LLM-based generative recommendation with the AtSpeed framework, unlocking the significant practical potential of LLM-based recommender models. This work is technical and practical, with applications to LLM-based generative recommendation. We have fully considered the potential societal impacts and do not foresee any direct, immediate, or negative consequences. We are committed to the ethical dissemination of our findings and encourage their responsible use.

## REPRODUCIBILITY STATEMENT

All the results in this work are reproducible. We provide all the necessary code to replicate our results in an GitHub repository. The repository includes environment configurations, run scripts, and other relevant materials. We discuss the experimental settings in Section 4.1, including implementation details such as the utilization of parameter-efficient training. Additionally, detailed experimental settings are provided in Appendix A.3.1.

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

# A   APPENDIX

## A.1   ALGORITHM OF RELAXED SAMPLING VERIFICATION

---

**Algorithm 2** SD step with Relaxed Sampling Verification

---

**Input:** Draft model $\mathcal{M}_q$, target LLM $\mathcal{M}_p$, prefix, target beam size $K$, draft beam size $N$
 1: $\mathcal{Y}_0^q \leftarrow$ prefix, $\mathcal{Y}_{\gamma+1}^q \leftarrow \emptyset$,    $\mathcal{Y}_{\text{out}} \leftarrow \emptyset$
 2: **for** $j = 1$ to $\gamma$ **do**
 3:     $q_j \leftarrow \mathcal{M}_q(\mathcal{Y}_{j-1}^q)$, $\mathcal{Y}_j^q \leftarrow$ Sample $K$ sequences from $q_j$
 4: $p_1, p_2, \ldots, p_{\gamma+1} \leftarrow \mathcal{M}_p(\mathcal{Y}_0^q), \mathcal{M}_p(\mathcal{Y}_1^q), \ldots, \mathcal{M}_p(\mathcal{Y}_\gamma^q)$                    ▷ Run $\mathcal{M}_p$ in parallel
 5: **for** $j = 1$ to $\gamma + 1$ **do**
 6:     $\mathcal{Y}_{\text{rej}} \leftarrow \emptyset$
 7:     **for** $i = 1$ to $K$ **do**                    ▷ Verify each sequence based on independent modeling
 8:         $r_i \sim U(0,1)$, $\boldsymbol{y}_i \leftarrow$ The $i$-th sequence from $\mathcal{Y}_j^q$
 9:         **if** $r_i \leq \frac{p(\boldsymbol{y}_i)}{q(\boldsymbol{y}_i)}$ **then**                    ▷ Accept the sequence
10:             $\mathcal{Y}_{\text{out}} \leftarrow \mathcal{Y}_{\text{out}} \bigcup \{\boldsymbol{y}_i\}$
11:         **else**                    ▷ Reject the sequence
12:             $\mathcal{Y}_{\text{rej}} \leftarrow \mathcal{Y}_{\text{rej}} \bigcup \{\boldsymbol{y}_i\}$
13:     **if** $|\mathcal{Y}_{\text{out}}| > K$ **then**                    ▷ Accept the step
14:         $\mathcal{Y}_{\text{out}} \leftarrow$ Randomly sample K sequences from $\mathcal{Y}_{\text{out}}$
15:     **else**                    ▷ Reject the step and resample the sequences
16:         $p_j' \leftarrow \text{norm}(\max(0, p_j - q_j))$
17:         $p_j'(\boldsymbol{y}_i) \leftarrow 0$ for $\boldsymbol{y} \in \mathcal{Y}_{\text{out}}$
18:         $\mathcal{Y}_{\text{resample}} \leftarrow$ Sample $B$ sequences from $p_j'$,    $B = K - |\mathcal{Y}_{\text{out}}|$
19:         $\mathcal{Y}_{\text{out}} \leftarrow \mathcal{Y}_{\text{out}} \bigcup \mathcal{Y}_{\text{resample}}$
**Output:** $\mathcal{Y}_{\text{out}}$

---

In Algorithm 2, $\text{norm}(\max(0, p_j - q_j)) = \frac{\max(0, p_j - q_j)}{\sum \max(0, p_j - q_j)}$.

## A.2   PROOF AND DETAILED DERIVATION

### A.2.1   WITH-REPLACEMENT SAMPLING APPROXIMATION

We aim to prove that the distribution of sampling without replacement is approximately equivalent to that of sampling with replacement. Our proof is mainly based on Stirling's approximation (Dutka, 1991). In the following, we will first clarify notations, and introduce multivariate hypergeometric distribution and the multinomial distribution, which are used to model the sampling without replacement and sampling with replacement, respectively. We then present Stirling's approximation, and show the step-by-step proof.

**Notations.** To model the sampling, we have the total population size $N$, sample size $n$, the category size $r$, the number of items in category $i$ in the population $K_i$, and the number of items in category $i$ in the samples $k_i$. In the case of sequence sampling in LLM decoding, the population includes every possible sequence. Every possible sequence is a unique category, and the sample size $n$ is the beam size, the population size is the total number of all possible sequences at each beam search step. Now we assume the population sizes go to infinity in such a way that $p_i = \frac{K_i}{N}$. And we have $\sum_{i=1}^r k_i = n$ and $\sum_{i=1}^r K_i = N$.

**Multivariate Hypergeometric Distribution.** Formally, when sampling without replacement, the probability of drawing $k_1, k_2, \ldots, k_r$ items from each category is given by the multivariate hypergeometric distribution

$$
\begin{aligned}
P_{\text{hyper}}(k_1, k_2, \ldots, k_r) &= \frac{\prod_{i=1}^r \binom{K_i}{k_i}}{\binom{N}{n}} \\
&= \frac{\prod_{i=1}^r \frac{K_i!}{k_i!(K_i-k_i)!}}{\frac{N!}{n!(N-n)!}}.
\end{aligned}
\tag{10}
$$

**Multinomial Distribution.** Formally, when sampling with replacement, the probability follows the multinomial distribution as

$$P_{\text{multi}}(k_1, k_2, \ldots, k_r) = \frac{n!}{\prod_{i=1}^{r} k_i!} \prod_{i=1}^{r} p_i^{k_i}. \tag{11}$$

**Stirling's Approximation.** Stirling's approximation (Dutka, 1991) gives us the approximation of the logarithm of factorials as:

$$\ln n! \approx n \ln n - n + \frac{1}{2} \ln(2\pi n). \tag{12}$$

**Theorem 1.** *When population size $N$ is large and sample size $n$ is small compared to $N$ (i.e., $n \ll N$), the multivariate hypergeometric distribution approximates the multinomial distribution:*

$$P_{\text{hyper}}(k_1, k_2, \ldots, k_r) \approx P_{\text{multi}}(k_1, k_2, \ldots, k_r). \tag{13}$$

*Proof.* We can expand the logarithm of multivariate hypergeometric probability in factorials as follows:

$$
\begin{aligned}
\ln P_{\text{hyper}}(k_1, k_2, \ldots, k_r) &= \ln \frac{\prod_{i=1}^{r} \frac{K_i!}{k_i!(K_i - k_i)!}}{\frac{N!}{n!(N-n)!}} \\
&= \ln \prod_{i=1}^{r} \frac{K_i!}{k_i!(K_i - k_i)!} - \ln \frac{N!}{n!(N-n)!} \\
&= \sum_{i=1}^{r} \ln \frac{K_i!}{k_i!(K_i - k_i)!} - \ln \frac{N!}{n!(N-n)!} \\
&= \sum_{i=1}^{r} [\ln K_i! - \ln k_i! - \ln(K_i - k_i)!] - [\ln N! - \ln n! - \ln(N - n)!].
\end{aligned}
\tag{14}
$$

Using the Stirling's approximation, we have:

$$
\begin{cases}
\ln K_i! \approx K_i \ln K_i - K_i + \frac{1}{2} \ln 2\pi K_i, \\[2mm]
\ln k_i! \approx k_i \ln k_i - k_i + \frac{1}{2} \ln 2\pi k_i, \\[2mm]
\ln(K_i - k_i)! \approx (K_i - k_i) \ln(K_i - k_i) - (K_i - k_i) + \frac{1}{2} \ln 2\pi(K_i - k_i), \\[2mm]
\ln N! \approx N \ln N - N + \frac{1}{2} \ln 2\pi N, \\[2mm]
\ln n! \approx n \ln n - n + \frac{1}{2} \ln 2\pi n, \\[2mm]
\ln(N - n)! \approx (N - n) \ln(N - n) - (N - n) + \frac{1}{2} \ln 2\pi(N - n).
\end{cases}
\tag{15}
$$

Then, we can substitute the logarithm of factorials with the approximation in Eq.(14) as:

$$\ln P_{\text{hyper}}(k_1, k_2, \ldots, k_r)$$

$$= \sum_{i=1}^{r}[\ln K_i! - \ln k_i! - \ln(K_i - k_i)!] - [\ln N! - \ln n! - \ln(N - n)!]$$

$$= \sum_{i=1}^{r}[K_i \ln K_i - K_i + \frac{1}{2}\ln 2\pi K_i - (k_i \ln k_i - k_i + \frac{1}{2}\ln 2\pi k_i)$$

$$- ((K_i - k_i)\ln(K_i - k_i) - (K_i - k_i) + \frac{1}{2}\ln 2\pi(K_i - k_i))]$$

$$- [N \ln N - N + \frac{1}{2}\ln 2\pi N - (n \ln n - n + \frac{1}{2}\ln 2\pi n)$$

$$- ((N - n)\ln(N - n) - (N - n) + \frac{1}{2}\ln 2\pi(N - n))]$$

$$= \sum_{i=1}^{r}[K_i \ln K_i - k_i \ln k_i - (K_i - k_i)\ln(K_i - k_i) + \frac{1}{2}\ln 2\pi K_i - \frac{1}{2}\ln 2\pi k_i - \frac{1}{2}\ln 2\pi(K_i - k_i)]$$

$$- [N \ln N - n \ln n - (N - n)\ln(N - n) + \frac{1}{2}\ln 2\pi N - \frac{1}{2}\ln 2\pi n - \frac{1}{2}\ln 2\pi(N - n)]. \tag{16}$$

Since $N$ is a very large number and $n \ll N$, $k_i \ll K_i$, we have $\frac{1}{2}\ln 2\pi K_i - \frac{1}{2}\ln 2\pi k_i - \frac{1}{2}\ln 2\pi(K_i - k_i) \approx 0$ and $\frac{1}{2}\ln 2\pi N - \frac{1}{2}\ln 2\pi n - \frac{1}{2}\ln 2\pi(N - n) \approx 0$. Then, the logarithm of multivariate hypergeometric distribution is approximated as:

$$\ln P_{\text{hyper}} \approx \sum_{i=1}^{r}[K_i \ln K_i - k_i \ln k_i - (K_i - k_i)\ln(K_i - k_i)] - [N \ln N - n \ln n - (N - n)\ln(N - n)]. \tag{17}$$

Since $k_i$ is small compared to $K_i$, we can expand $\ln(K_i - k_i)$ using Taylor expansion

$$\ln(K_i - k_i) = \ln K_i - \frac{k_i}{K_i} - \frac{1}{2}(\frac{k_i}{K_i})^2 + \ldots, \tag{18}$$

where we can neglect the high-order terms and obtain

$$\ln(K_i - k_i) \approx \ln K_i - \frac{k_i}{K_i}. \tag{19}$$

Similarly, for $\ln(N - n)$, we have

$$\ln(N - n) \approx \ln N - \frac{n}{N}. \tag{20}$$

We can then substitute $\ln(K_i - k_i)$ and $\ln(N - n)$ in logarithm of multivariate hypergeometric distribution (Eq.(17)) and obtain

$$\ln P_{\text{hyper}} \approx \sum_{i=1}^{r}[K_i \ln K_i - k_i \ln k_i - (K_i - k_i)\ln(K_i - k_i)] - [N \ln N - n \ln n - (N - n)\ln(N - n)]$$

$$= \sum_{i=1}^{r}[K_i \ln K_i - k_i \ln k_i - (K_i - k_i)(\ln K_i + \frac{k_i}{K_i})] - [N \ln N - n \ln n - (N - n)(\ln N + \frac{n}{N})]$$

$$= \sum_{i=1}^{r}[K_i \ln K_i - k_i \ln k_i - (K_i \ln K_i - k_i \ln K_i + k_i - \frac{k_i^2}{K_i})] - [N \ln N - n \ln n - (N \ln N - n \ln N + n - \frac{n^2}{N})]$$

$$= \sum_{i=1}^{r}[k_i \ln \frac{K_i}{k_i} + \frac{k_i^2}{K_i} - k_i] - [n \ln \frac{N}{n} + \frac{n^2}{N} - n]$$

$$= \sum_{i=1}^{r}[k_i \ln \frac{K_i}{k_i} + \frac{k_i^2}{K_i}] - [n \ln \frac{N}{n} + \frac{n^2}{N}]. \qquad (\text{we have } -\sum_{i=1}^{r}k_i + n = 0 \text{ in last expression}) \tag{21}$$

We then relate $K_i$ to $N$ and $p_i$. Since $K_i = Np_i$, we have $\ln \frac{K_i}{k_i} = \ln \frac{Np_i}{k_i}$. We also have $n = \sum_{i=1}^{r} k_i$. Then, we can express the logarithm of hypergeometric distribution in terms of $p_i$ and $k_i$ as

$$
\begin{aligned}
\ln P_{\text{hyper}} &\approx \sum_{i=1}^{r} [k_i \ln \frac{K_i}{k_i} + \frac{k_i^2}{K_i}] - [n \ln \frac{N}{n} + \frac{n^2}{N}] \\
&= \sum_{i=1}^{r} [k_i \ln \frac{Np_i}{k_i} + \frac{k_i^2}{Np_i}] - [n \ln \frac{N}{n} + \frac{n^2}{N}] \\
&= \sum_{i=1}^{r} [k_i (\ln N + \ln p_i - \ln k_i) + \frac{k_i^2}{Np_i}] - [n \ln \frac{N}{n} + \frac{n^2}{N}] \\
&= \sum_{i=1}^{r} k_i \ln N + \sum_{i=1}^{r} [k_i (\ln p_i - \ln k_i) + \frac{k_i^2}{Np_i}] - n \ln \frac{N}{n} - \frac{n^2}{N} \\
&= n \ln N - n \ln N + n \ln n - \frac{n^2}{N} + \sum_{i=1}^{r} [k_i (\ln p_i - \ln k_i) + \frac{k_i^2}{Np_i}] \quad \text{(we have } \sum_{i=1}^{r} k_i = n \text{ in last expression)} \\
&= \sum_{i=1}^{r} [k_i (\ln p_i - \ln k_i) + \frac{k_i^2}{Np_i}] + n \ln n - \frac{n^2}{N} \\
&= n \ln n - \sum_{i=1}^{r} k_i \ln k_i + \sum_{i=1}^{r} k_i \ln p_i - \frac{n^2}{N} + \sum_{i=1}^{r} \frac{k_i^2}{Np_i}.
\end{aligned}
\tag{22}
$$

Now the approximation of the logarithm of multivariate hypergeometric distribution has been finished. Similarly, we approximate the multinomial distribution with Stirling's approximation. We expand the logarithm of multinomial distribution as

$$
\begin{aligned}
\ln P_{\text{multi}}(k_1, k_2, \ldots, k_r) &= \ln \frac{n!}{\prod_{i=1}^{r} k_i!} \prod_{i=1}^{r} p_i^{k_i} \\
&= \ln n! - \sum_{i=1}^{r} \ln k_i! + \sum_{i=1}^{r} k_i \ln p_i.
\end{aligned}
\tag{23}
$$

Using Stirling's approximation, we have $\ln n! \approx n \ln n - n$ and $\ln k_i! \approx k_i \ln k_i - k_i$. We then substitute $\ln n!$ and $\ln k_i!$ with the approximation and obtain

$$
\begin{aligned}
\ln P_{\text{multi}} &\approx n \ln n - n - \sum_{i=1}^{r} (k_i \ln k_i - k_i) + \sum_{i=1}^{r} k_i \ln p_i \\
&= n \ln n - n - \sum_{i=1}^{r} k_i \ln k_i + \sum_{i=1}^{r} k_i + \sum_{i=1}^{r} k_i \ln p_i.
\end{aligned}
\tag{24}
$$

Since we have $\sum_{i=1}^{r} k_i = n$, we have

$$
\begin{aligned}
\ln P_{\text{multi}} &\approx n \ln n - n - \sum_{i=1}^{r} k_i \ln k_i + n + \sum_{i=1}^{r} k_i \ln p_i \\
&= n \ln n - \sum_{i=1}^{r} k_i \ln k_i + \sum_{i=1}^{r} k_i \ln p_i.
\end{aligned}
\tag{25}
$$

Now, comparing the approximated logarithm of multinomial distribution with the approximated logarithm of multivariate hypergeometric distribution, we have

$$
\ln P_{\text{multi}} \approx \ln P_{\text{hyper}} + \frac{n^2}{N} - \sum_{i=1}^{r} \frac{k_i^2}{Np_i}.
\tag{26}
$$

Note that the term $\frac{n^2}{N} - \sum_{i=1}^{r} \frac{k_i^2}{N p_i}$ is negalectable when $N$ is large and $k_i$ and $n$ are small compared to N. Therefore, we show that when the population size $N$ is large (*e.g.,* all sequences for sampling) and $k_i, n$ are small (*e.g.,* $k_i = 1$ or 0 since each sequence represents a category and $n$ is usually less than 20 in LLM-based recommendation), the multivariate hypergeometric distribution can be approximated to the multinomial distribution. That is, sampling without replacement is approximately equivalent to sampling with replacement. □

### A.2.2 DERIVATION OF ALIGNMENT OBJECTIVES OF ATSPEED-S

To maximize the acceptance rate under the strict top-$K$ verification, we consider:

$$
\begin{aligned}
&\underset{\theta \in \Theta}{\arg\max} \quad \mathbb{E}_{\boldsymbol{y} \sim \mathcal{Y}_{q_\theta}} \frac{p(\boldsymbol{y})}{p(\boldsymbol{y}_K)} \\
&\underset{\theta \in \Theta}{\arg\max} \quad \mathbb{E}_{\boldsymbol{y} \sim \mathcal{Y}_{q_\theta}} \log \frac{p(\boldsymbol{y})}{p(\boldsymbol{y}_K)} \\
&\qquad = \sum_{\boldsymbol{y}} q(\boldsymbol{y}) \log p(\boldsymbol{y}) - \sum_{\boldsymbol{y}} q(\boldsymbol{y}) \log p(\boldsymbol{y}_K) \\
&\qquad = -\sum_{\boldsymbol{y}} q(\boldsymbol{y}) \log \frac{1}{p(\boldsymbol{y})} - \sum_{\boldsymbol{y}} q(\boldsymbol{y}) \log p(\boldsymbol{y}_K).
\end{aligned}
\tag{27}
$$

We can add $(\sum_{\boldsymbol{y}} q(\boldsymbol{y}) \log q(\boldsymbol{y}) - \sum_{\boldsymbol{y}} q(\boldsymbol{y}) \log q(\boldsymbol{y}))$ to the expression since $\sum_{\boldsymbol{y}} q(\boldsymbol{y}) \log q(\boldsymbol{y}) - \sum_{\boldsymbol{y}} q(\boldsymbol{y}) \log q(\boldsymbol{y}) = 0$. After adding the term with some rearrangement, we obtain

$$
\begin{aligned}
&-\sum_{\boldsymbol{y}} q(\boldsymbol{y}) \log \frac{1}{p(\boldsymbol{y})} - \sum_{\boldsymbol{y}} q(\boldsymbol{y}) \log p(\boldsymbol{y}_K) + \left( \sum_{\boldsymbol{y}} q(\boldsymbol{y}) \log q(\boldsymbol{y}) - \sum_{\boldsymbol{y}} q(\boldsymbol{y}) \log q(\boldsymbol{y}) \right) \\
&= -\sum_{\boldsymbol{y}} q(\boldsymbol{y}) \log \frac{1}{p(\boldsymbol{y})} - \sum_{\boldsymbol{y}} q(\boldsymbol{y}) \log q(\boldsymbol{y}) + \sum_{\boldsymbol{y}} q(\boldsymbol{y}) \log q(\boldsymbol{y}) - \sum_{\boldsymbol{y}} q(\boldsymbol{y}) \log p(\boldsymbol{y}_K) \\
&= -\sum_{\boldsymbol{y}} q(\boldsymbol{y}) \log \frac{q(\boldsymbol{y})}{p(\boldsymbol{y})} + \sum_{\boldsymbol{y}} q(\boldsymbol{y}) \log q(\boldsymbol{y}) - \sum_{\boldsymbol{y}} q(\boldsymbol{y}) \log p(\boldsymbol{y}_K) \\
&= -\sum_{\boldsymbol{y}} q(\boldsymbol{y}) \log \frac{q(\boldsymbol{y})}{p(\boldsymbol{y})} + \sum_{\boldsymbol{y}} q(\boldsymbol{y}) \log \frac{q(\boldsymbol{y})}{p(\boldsymbol{y}_K)},
\end{aligned}
\tag{28}
$$

where $p(y_K) = p(\boldsymbol{y}_{K,t}|\boldsymbol{y}_{K,<t})$. According to the derivation, we obtain the alignment objective as in Eq.(2). We now consider context in LLM-based recommendation. We omit input $\boldsymbol{x}$ and denote $c_{<t} = \boldsymbol{y}_{<t}$, referred to as context, then we have $q(\boldsymbol{y}) = \prod_t q(y_t|c_{<t})$ and $p(\boldsymbol{y}) = \prod_t p(y_t|c_{<t})$. We then can substitute $q(\boldsymbol{y})$ with $\prod_t q(y_t|c_{<t})$ and $p(\boldsymbol{y})$ with $\prod_t p(y_t|c_{<t})$ and rewrite the objective in Eq.(28) as:

$$
\begin{aligned}
&-\sum_{\boldsymbol{y}} q(\boldsymbol{y}) \log \frac{q(\boldsymbol{y})}{p(\boldsymbol{y})} + \sum_{\boldsymbol{y}} q(\boldsymbol{y}) \log \frac{q(\boldsymbol{y})}{p(\boldsymbol{y}_K)} \\
&= -\sum_{\boldsymbol{y}} q(\boldsymbol{y}) \log \frac{\prod_t q(y_t|c_{<t})}{\prod_t p(y_t|c_{<t})} + \sum_{\boldsymbol{y}} q(\boldsymbol{y}) \log \frac{\prod_t q(y_t|c_{<t})}{\prod_t p(\boldsymbol{y}_{K,t}|\boldsymbol{y}_{K,<t})} \\
&= -\sum_{\boldsymbol{y}} q(\boldsymbol{y}) [\log \prod_t q(y_t|c_{<t}) - \log \prod_t p(y_t|c_{<t})] + \sum_{\boldsymbol{y}} q(\boldsymbol{y}) [\log \prod_t q(y_t|c_{<t}) - \log \prod_t p(\boldsymbol{y}_{K,t}|\boldsymbol{y}_{K,<t})] \\
&= -\sum_{\boldsymbol{y}} q(\boldsymbol{y}) [\sum_t \log q(y_t|c_{<t}) - \sum_t \log p(y_t|c_{<t})] + \sum_{\boldsymbol{y}} q(\boldsymbol{y}) [\sum_t \log q(y_t|c_{<t}) - \sum_t \log p(\boldsymbol{y}_{K,t}|\boldsymbol{y}_{K,<t})] \\
&= -\sum_{\boldsymbol{y}} q(\boldsymbol{y}) \sum_t \log \frac{q(y_t|c_{<t})}{p(y_t|c_{<t})} + \sum_{\boldsymbol{y}} q(\boldsymbol{y}) \sum_t \log \frac{q(y_t|c_{<t})}{p(\boldsymbol{y}_{K,t}|\boldsymbol{y}_{K,<t})} \\
&= -\sum_t \sum_{\boldsymbol{y}} q(\boldsymbol{y}) \log \frac{q(y_t|c_{<t})}{p(y_t|c_{<t})} + \sum_t \sum_{\boldsymbol{y}} q(\boldsymbol{y}) \log \frac{q(y_t|c_{<t})}{p(\boldsymbol{y}_{K,t}|\boldsymbol{y}_{K,<t})}.
\end{aligned}
\tag{29}
$$

Since $\boldsymbol{y} = (y_1, y_2, \ldots, y_n) = (\boldsymbol{y}_{<t}, y_t, \boldsymbol{y}_{>t})$ is a sequence, we can decompose $\sum_{\boldsymbol{y}} q(\boldsymbol{y})$ into nested sum $\sum_{\boldsymbol{y}_{<t}} \sum_{y_t} \sum_{\boldsymbol{y}_{>t}} q(\boldsymbol{y}) = \sum_{\boldsymbol{y}_{<t}} \sum_{y_t} \sum_{\boldsymbol{y}_{>t}} q(\boldsymbol{y}_{<t}, y_t, \boldsymbol{y}_{>t})$, where the nested sum over

three parts of $\boldsymbol{y}$, *i.e.,* $\boldsymbol{y}_{<t}, y_t$, and $\boldsymbol{y}_{>t}$ can cover all possible sequence $\boldsymbol{y}$. Since $q(\boldsymbol{y}_{<t}, y_t, \boldsymbol{y}_{>t}) = q(\boldsymbol{y}_{<t})q(y_t|c_{<t})q(\boldsymbol{y}_{>t}|c_{\leq t})$, we can rewrite the nested sum of $q(\boldsymbol{y}_{<t}, y_t, \boldsymbol{y}_{>t})$ over $\boldsymbol{y}$:

$$
\begin{aligned}
&\sum_{\boldsymbol{y}} q(\boldsymbol{y}) \\
&= \sum_{\boldsymbol{y}_{<t}} \sum_{y_t} \sum_{\boldsymbol{y}_{>t}} q(\boldsymbol{y}_{<t})q(y_t|c_{<t})q(\boldsymbol{y}_{>t}|c_{\leq t}) \\
&= \sum_{\boldsymbol{y}_{<t}} \sum_{y_t} q(\boldsymbol{y}_{<t})q(y_t|c_{<t}) \sum_{\boldsymbol{y}_{>t}} q(\boldsymbol{y}_{>t}|c_{\leq t}) \\
&= \sum_{\boldsymbol{y}_{<t}} q(\boldsymbol{y}_{<t}) \sum_{y_t} q(y_t|c_{<t}) \sum_{\boldsymbol{y}_{>t}} q(\boldsymbol{y}_{>t}|c_{\leq t}).
\end{aligned}
\tag{30}
$$

We then can substitute $\sum_{\boldsymbol{y}} q(\boldsymbol{y})$ with $\sum_{\boldsymbol{y}_{<t}} q(\boldsymbol{y}_{<t}) \sum_{y_t} q(y_t|\boldsymbol{y}_{<t}) \sum_{\boldsymbol{y}_{>t}} q(\boldsymbol{y}_{>t}|c_{\leq t})$ into Eq.(29) and obtain

$$
\begin{aligned}
&-\sum_t \sum_{\boldsymbol{y}} q(\boldsymbol{y}) \log \frac{q(y_t|c_{<t})}{p(y_t|c_{<t})} + \sum_t \sum_{\boldsymbol{y}} q(\boldsymbol{y}) \log \frac{q(y_t|c_{<t})}{p(\boldsymbol{y}_{K,t}|\boldsymbol{y}_{K,<t})} \\
&= -\sum_t [\sum_{\boldsymbol{y}_{<t}} q(\boldsymbol{y}_{<t}) \sum_{y_t} q(y_t|c_{<t}) \sum_{\boldsymbol{y}_{>t}} q(\boldsymbol{y}_{>t}|c_{\leq t})] \log \frac{q(y_t|c_{<t})}{p(y_t|c_{<t})} \\
&\qquad\qquad\qquad + \sum_t [\sum_{\boldsymbol{y}_{<t}} q(c_{<t}) \sum_{y_t} q(y_t|c_{<t}) \sum_{\boldsymbol{y}_{>t}} q(\boldsymbol{y}_{>t}|c_{\leq t})] \log \frac{q(y_t|c_{<t})}{p(\boldsymbol{y}_{K,t}|\boldsymbol{y}_{K,<t})} \\
&= -\sum_t \sum_{\boldsymbol{y}_{<t}} q(\boldsymbol{y}_{<t}) \sum_{y_t} q(y_t|c_{<t}) \log \frac{q(y_t|c_{<t})}{p(y_t|c_{<t})} \sum_{\boldsymbol{y}_{>t}} q(\boldsymbol{y}_{>t}|c_{\leq t}) \\
&\qquad\qquad\qquad + \sum_t \sum_{\boldsymbol{y}_{<t}} q(\boldsymbol{y}_{<t}) \sum_{y_t} q(y_t|c_{<t}) \log \frac{q(y_t|c_{<t})}{p(\boldsymbol{y}_{K,t}|\boldsymbol{y}_{K,<t})} \sum_{\boldsymbol{y}_{>t}} q(\boldsymbol{y}_{>t}|c_{\leq t}).
\end{aligned}
\tag{31}
$$

Since $\sum_{\boldsymbol{y}_{>t}} q(\boldsymbol{y}_{>t}|c_{\leq t}) = 1$, we can remove $\sum_{\boldsymbol{y}_{>t}} q(\boldsymbol{y}_{>t}|c_{\leq t})$ in Eq.(31):

$$
\begin{aligned}
&-\sum_t \sum_{\boldsymbol{y}_{<t}} q(\boldsymbol{y}_{<t}) \sum_{y_t} q(y_t|c_{<t}) \log \frac{q(y_t|c_{<t})}{p(y_t|c_{<t})} \sum_{\boldsymbol{y}_{>t}} q(\boldsymbol{y}_{>t}|c_{\leq t}) \\
&\qquad\qquad\qquad + \sum_t \sum_{\boldsymbol{y}_{<t}} q(\boldsymbol{y}_{<t}) \sum_{y_t} q(y_t|c_{<t}) \log \frac{q(y_t|c_{<t})}{p(\boldsymbol{y}_{K,t}|\boldsymbol{y}_{K,<t})} \sum_{\boldsymbol{y}_{>t}} q(\boldsymbol{y}_{>t}|c_{\leq t}) \\
&= -\sum_t \sum_{\boldsymbol{y}_{<t}} q(\boldsymbol{y}_{<t}) \sum_{y_t} q(y_t|c_{<t}) \log \frac{q(y_t|c_{<t})}{p(y_t|c_{<t})} + \sum_t \sum_{\boldsymbol{y}_{<t}} q(\boldsymbol{y}_{<t}) \sum_{y_t} q(y_t|c_{<t}) \log \frac{q(y_t|c_{<t})}{p(\boldsymbol{y}_{K,t}|\boldsymbol{y}_{K,<t})} \\
&= -\sum_t \sum_{\boldsymbol{y}_{<t}} q(\boldsymbol{y}_{<t}) [\sum_{y_t} q(y_t|c_{<t}) \log \frac{q(y_t|c_{<t})}{p(y_t|c_{<t})} - \sum_{y_t} q(y_t|c_{<t}) \log \frac{q(y_t|c_{<t})}{p(\boldsymbol{y}_{K,t}|\boldsymbol{y}_{K,<t})}] \\
&= -\sum_t \mathbb{E}_{\boldsymbol{y}_{<t} \sim q} [\sum_{y_t} q(y_t|c_{<t}) \log \frac{q(y_t|c_{<t})}{p(y_t|c_{<t})} - \sum_{y_t} q(y_t|c_{<t}) \log \frac{q(y_t|c_{<t})}{p(\boldsymbol{y}_{K,t}|\boldsymbol{y}_{K,<t})}].
\end{aligned}
\tag{32}
$$

Now we have a step-wise alignment over all sequence from $q$, *i.e.,* $\mathbb{E}_{\boldsymbol{y}_{<t} \sim q}$. We aim to align the draft model with the target LLM at every beam search step. Notably, at each beam search step $T$, the sequence lengths are fixed and is independent with $t$. Therefore, we can rewrite Eq.(32) as:

$$
\begin{aligned}
&-\sum_t \mathbb{E}_{\boldsymbol{y}_{<t} \sim q} [\sum_{y_t} q(y_t|c_{<t}) \log \frac{q(y_t|c_{<t})}{p(y_t|c_{<t})} - \sum_{y_t} q(y_t|c_{<t}) \log \frac{q(y_t|c_{<t})}{p(\boldsymbol{y}_{K,t}|\boldsymbol{y}_{K,<t})}] \\
&= -\mathbb{E}_{\boldsymbol{y}_{\leq T} \sim q} \sum_{t=1}^T [\sum_{y_t} q(y_t|c_{<t}) \log \frac{q(y_t|c_{<t})}{p(y_t|c_{<t})} - \sum_{y_t} q(y_t|c_{<t}) \log \frac{q(y_t|c_{<t})}{p(\boldsymbol{y}_{K,t}|\boldsymbol{y}_{K,<t})}].
\end{aligned}
\tag{33}
$$

Since we aim to align at every step $T \in \{1, \ldots, L\}$, where $L$ is the length of item identifier in LLM-based recommendation, we further normalize the expression by sequence length to prevent

different scales on alignment loss across different steps. As such, for every step $T \in \{1, \ldots, L\}$, the objective can be rewritten as

$$-\mathbb{E}_{\boldsymbol{y}_{\leq T} \sim q} \frac{1}{|y_T|} \sum_{t=1}^{|y_T|} [\sum_{y_t} q(y_t) \log \frac{q(y_t|c_{<t})}{p(y_t|c_{<t})} - \sum_{y_t} q(y_t) \log \frac{q(y_t|c_{<t})}{p(\boldsymbol{y}_{K,t}|\boldsymbol{y}_{K,<t})}]. \quad (34)$$

However, the expectation over the sequence space, (*i.e.*, $\mathbb{E}_{\boldsymbol{y}_{\leq T} \sim q}$) is intractable, so we follow previous work (Wen et al., 2023; Kim & Rush, 2016) to approximate it by sampling top-$K$ sequences generated by draft model $\mathcal{M}_q$. Now we can rewrite our alignment objective for strict top-$K$ verification and obtain Eq.(3) in our paper:

$$\arg\max_{\theta \in \Theta} -\mathbb{E}_{(\boldsymbol{x}, \mathcal{Y}) \sim D'} \sum_{\boldsymbol{y} \in \mathcal{Y}} \frac{1}{|\boldsymbol{y}|} \sum_{t=1}^{|\boldsymbol{y}|} [\sum_{y_t} q(y_t|c_{<t}) \log \frac{q(y_t|c_{<t})}{p(y_t|c_{<t})} - \sum_{y_t} q(y_t|c_{<t}) \log \frac{q(y_t|c_{<t})}{p(\boldsymbol{y}_{K,t}|\boldsymbol{y}_{K,<t})}]$$

$$= \arg\min_{\theta \in \Theta} \mathbb{E}_{(\boldsymbol{x}, \mathcal{Y}) \sim D'} \sum_{\boldsymbol{y} \in \mathcal{Y}} \frac{1}{|\boldsymbol{y}|} \sum_{t=1}^{|\boldsymbol{y}|} [\sum_{y_t} q(y_t|c_{<t}) \log \frac{q(y_t|c_{<t})}{p(y_t|c_{<t})} - \sum_{y_t} q(y_t|c_{<t}) \log \frac{q(y_t|c_{<t})}{p(\boldsymbol{y}_{K,t}|\boldsymbol{y}_{K,<t})}], \quad (35)$$

where $\mathcal{Y}$ denotes the top-$K$ beam sequences generated from the mixture distribution of draft model and target LLM (see "Alignment Objective" paragraph in Section 3.1).

### A.2.3 DERIVATIVES OF ACCEPTANCE RATE FOR RELAXED SAMPLING VERIFICATION.

Based on the with-replacement sampling approximation, for every independent sampling, we have acceptance rate of a sequence

$$b = \mathbb{E}_{\boldsymbol{y} \sim q(\boldsymbol{y})} \begin{cases} 1, & q(\boldsymbol{y}) \leq p(\boldsymbol{y}) \\ \frac{p(\boldsymbol{y})}{q(\boldsymbol{y})}, & q(\boldsymbol{y}) > p(\boldsymbol{y}) \end{cases} = \mathbb{E}_{y \sim q(\boldsymbol{y})} \min(1, \frac{p(\boldsymbol{y})}{q(\boldsymbol{y})}) = 1 - \text{TVD}(p(\boldsymbol{y}), q(\boldsymbol{y})), \quad (36)$$

which can be easily extended from (Leviathan et al., 2023) as follows.

**Definition 1.** $TVD(p, q) = \sum_{\boldsymbol{y}} |p(\boldsymbol{y}) - M(\boldsymbol{y})| = \sum_{\boldsymbol{y}} |q(\boldsymbol{y}) - M(\boldsymbol{y})|$, *where* $M(q) = \frac{p(\boldsymbol{y}) + q(\boldsymbol{y})}{2}$.

**Lemma 1.** $TVD = 1 - \sum_{\boldsymbol{y}} min(p(\boldsymbol{y}), q(\boldsymbol{y}))$.

*Proof.* $TVD(p, q) = \sum_y |p(\boldsymbol{y}) - M(\boldsymbol{y})| = \sum_{\boldsymbol{y}} \frac{|p(\boldsymbol{y}) - q(\boldsymbol{y})|}{2} = 1 - \sum_{\boldsymbol{y}} \frac{p(\boldsymbol{y}) + q(\boldsymbol{y}) - |p(\boldsymbol{y}) - q(\boldsymbol{y})|}{2} = 1 - \sum_{\boldsymbol{y}} \min(p(\boldsymbol{y}) - q(\boldsymbol{y}))$. $\square$

**Theorem 2.** $b = 1 - TVD(p, q)$.

*Proof.* $b = \mathbb{E}_{\boldsymbol{y} \sim q(\boldsymbol{y})} \begin{cases} 1, & q(\boldsymbol{y}) \leq p(\boldsymbol{y}) \\ \frac{p(\boldsymbol{y})}{q(\boldsymbol{y})}, & q(\boldsymbol{y}) > p(\boldsymbol{y}) \end{cases} = \mathbb{E}_{y \sim q(\boldsymbol{y})} \min(1, \frac{p(\boldsymbol{y})}{q(\boldsymbol{y})}) = \sum_{\boldsymbol{y}} \min(1, \frac{p(\boldsymbol{y})}{q(\boldsymbol{y})}) = 1 - \text{TVD}(p(\boldsymbol{y}), q(\boldsymbol{y}))$. $\square$

Based on the with-replacement sampling approximation (see Appendix A.2.1 for proof), we can obtain the acceptance rate of a step with $K$ accepted sequences as $\beta = \prod_K b$. Similarly, we could also extend the distribution equivalence for sampling a sequence between the draft model and the target LLM from Appendix A.1 in (Leviathan et al., 2023). As such, when the sampling-based beam search is performed with a beam size $K$, the joint distribution of the output $K$ sequences from the SD under the relaxed sampling verification is approximately equivalent to the counterparts from the target LLM, according to the approximation of with-replacement sampling, *i.e.,* independent sampling.

### A.3 EXPERIMENTAL DETAILS AND ANALYSIS

#### A.3.1 DETAILED EXPERIMENTAL SETTINGS

**Datasets Details.** For both Beauty and Games, all interactions are sorted according to the global timestamps, and then split into training, validation, and testing sets with the ratio of 8:1:1. The dataset statistics is presented in Table 2. For the item identifier, we follow LC-Rec (Zheng et al., 2024) to set the length $L = 4$, *i.e.*, the token sequence length of a generated item would be 4.

Table 2: Statistics of the datasets.

| Datasets | # Users | # Items | # Interactions | Density |
|---|---|---|---|---|
| Beauty | 19,383 | 12,035 | 138,870 | 0.06% |
| Games | 49,156 | 17,332 | 342,329 | 0.04% |
| MovieLens-1M | 6,038 | 3,017 | 60,162 | 0.33% |
| Goodreads | 2,437 | 4,667 | 80,744 | 0.71% |

**Implementation Details.** For draft model training, we use AdamW optimizer with batch size$= 64$, learning rate$=0.001$, and a cosine scheduler with warmup step of 200 to adjust the learning rate. We train the draft model for 20 epochs on 4 NVIDIA RTX A5000 GPUs. Meanwhile, we search the alignment strength $\alpha$ in $\{0.1, 0.3, 0.5, 0.7\}$ and weight decay in $\{0.01, 0.1\}$. For efficient training, we utilize top-$K$ sequences from the last step of beam search to construct the alignment data $\mathcal{D}'$. Despite the utilization of beam sequences from different steps of beam search could potentially enhance the acceleration effects further, it exacerbates the training burden regarding time cost and storage cost (see Table 7).

**Recommendation Loss.** The recommendation loss used for the backbone LLM-based recommender model in our work is defined as

$$\mathcal{L}_{\text{Rec}} = -\frac{1}{N} \sum_{(\boldsymbol{x}, \boldsymbol{y}) \sim \mathcal{D}} \sum_{t=1}^{|\boldsymbol{y}|} \log \mathcal{M}_p(y_t | \boldsymbol{y}_{<t}, \boldsymbol{x}), \tag{37}$$

where $\boldsymbol{x}$ is user's historical interactions, $\boldsymbol{y}$ is the user's next interacted item identifier, and $\mathcal{D} = \{(\boldsymbol{x}, \boldsymbol{y})\}$ denotes the original recommendation dataset. Nevertheless, our method is model-agnostic, thus the recommendation loss $\mathcal{L}_{\text{Rec}}$ can be can be substituted by any form of the loss function from the backend LLM-based recommender models.

**Implementation of Tree-based Attention.** We first compress the $N$ beam sequences into a single flattened sequence, and then construct the sparse tree-based attention mask for efficient target LLM verification. More precisely, given $\gamma N$ drafted beam sequences with different lengths, where $N$ is the draft beam size and $\gamma$ is the number of drafted beam steps, we first 1) flattened the beam sequences by sequentially adding the newly generated tokens from the beam sequences at each step. We denote the length of the flattened sequence as $L_f$. Then, based on the flattened sequence, 2) we construct an attention mask with the shape of $L_f \times L_f$. Specifically, each row in attention mask represents a specific beam sequence. For each row, we set the corresponding column of the last token as well as the preceding tokens in the beam sequence as 1, otherwise 0. For example, we set the beam size of 2 and collect the beam sequences of 3 steps as shown in Table 3.

|  |  |  |  |
|---|---|---|---|
| step 1 beam 1: | a1 | | |
| step 1 beam 2: | a2 | | |
| step 2 beam 1: | a1 | b1 | |
| step 2 beam 2: | a2 | b2 | |
| step 3 beam 1: | a1 | b1 | c1 |
| step 3 beam 2: | a1 | b1 | c2 |

Table 3: Beam sequences from 3 beam search steps with beam size 2. The flattened sequence is "a1 a2 b1 b2 c1 c2".

The flattened sequence will be "a1 a2 b1 b2 c1 c2". The constructed tree-based attention is shown in Figure 2(c) of our manuscript, where each row represents a specific beam sequence. For each row, the tickled cell represents the preceding tokens and the last tokens of each beam. And the different colors represent the different steps of beam search. This flattened sequence and sparse tree-based attention enable efficient verification since it saves the repeated calculation of the same prefix across different beam sequences, *e.g.,* "a1 b1".

### A.3.2 ADDITIONAL EXPERIMENT RESULTS.

**Comprehensive Results on Beauty and Games Datasets.** We report the full acceleration performance comparison on Beauty and Games datasets in Table 4 and the full recommendation performance comparison on the two datasets in Table 5, where we can have similar observations as in Table 1.

Table 4: Full results of walltime speedup (WS@$K$), accept steps (AS@$K$) between the baselines and AtSpeed instantiated on LC-Rec (LLaMA-7B) across two datasets. For each verification strategy, the best results are highlighted in bold and the second-best results are underlined.

| Beauty | | | | | | | | | | | |
|---|---|---|---|---|---|---|---|---|---|---|---|
| Verification | Method | WS@1 | WS@3 | WS@5 | WS@10 | WS@20 | AS@1 | AS@3 | AS@5 | AS@10 | AS@20 |
| Strict Top-$K$ | DARE | 1.13 | 1.07 | 1.06 | 1.15 | 1.48 | 0.81 | 0.44 | 0.26 | 0.05 | 0.00 |
| | SFT | 1.88 | 1.55 | 1.43 | 1.37 | 1.55 | 2.19 | 1.66 | 1.32 | 0.66 | 0.09 |
| | WordKD | 2.02 | 1.70 | 1.58 | 1.52 | 1.58 | 2.33 | 1.89 | 1.60 | 1.03 | 0.16 |
| | TVDKD | 1.84 | 1.55 | 1.44 | 1.37 | 1.57 | 2.15 | 1.67 | 1.31 | 0.65 | 0.09 |
| | SeqKD | 2.18 | 1.88 | 1.75 | 1.67 | 1.68 | 2.46 | 2.10 | 1.85 | 1.27 | 0.30 |
| | AtSpeed-S | **2.33** | **1.97** | **1.84** | **1.87** | **1.84** | **2.58** | **2.20** | **2.00** | 1.64 | **0.57** |
| | AtSpeed-R | 1.95 | 1.71 | 1.70 | 1.71 | 1.74 | 2.26 | 1.96 | 1.82 | 1.33 | 0.43 |
| Relaxed Sampling | DARE | 1.61 | 1.65 | 1.70 | 1.53 | 1.95 | 2.00 | 2.00 | 1.97 | 1.14 | 1.00 |
| | SFT | 1.77 | 1.76 | 1.80 | 2.06 | 2.36 | 2.15 | 2.06 | 2.03 | 1.99 | 1.48 |
| | WordKD | 1.74 | 1.75 | 1.81 | 1.99 | 2.05 | 2.11 | 2.04 | 2.01 | 1.87 | 1.07 |
| | TVDKD | 1.80 | 1.77 | 1.81 | 2.06 | 2.35 | 2.17 | 2.07 | 2.03 | 1.99 | 1.45 |
| | SeqKD | 1.85 | 1.86 | 1.90 | 2.11 | 2.31 | 2.21 | 2.13 | 2.10 | 2.01 | 1.40 |
| | AtSpeed-S | 1.94 | 1.87 | 1.89 | 2.12 | **2.51** | 2.26 | 2.14 | 2.09 | **2.03** | 1.71 |
| | AtSpeed-R | **2.05** | **1.94** | **1.94** | **2.16** | 2.47 | **2.37** | **2.19** | **2.13** | 2.01 | **1.77** |
| Games | | | | | | | | | | | |
| Verification | Method | WS@1 | WS@3 | WS@5 | WS@10 | WS@20 | AS@1 | AS@3 | AS@5 | AS@10 | AS@20 |
| Strict Top-$K$ | DARE | 0.94 | 0.95 | 0.99 | 1.13 | 1.44 | 0.15 | 0.00 | 0.00 | 0.00 | 0.00 |
| | SFT | 1.82 | 1.45 | 1.43 | 1.40 | 1.58 | 2.13 | 1.49 | 1.32 | 0.91 | 0.15 |
| | WordKD | 1.82 | 1.45 | 1.31 | 1.35 | 1.47 | 2.13 | 1.49 | 1.10 | 0.80 | 0.14 |
| | TVDKD | 1.70 | 1.33 | 1.24 | 1.32 | 1.50 | 1.99 | 1.26 | 0.95 | 0.66 | 0.09 |
| | SeqKD | 2.01 | 1.72 | 1.60 | 1.46 | **1.77** | **2.31** | 1.95 | 1.67 | 1.05 | **0.76** |
| | AtSpeed-S | **2.01** | **1.77** | **1.78** | **1.85** | 1.76 | 2.27 | **2.02** | **1.96** | **1.69** | 0.68 |
| | AtSpeed-R | 1.85 | 1.71 | 1.76 | 1.76 | 1.60 | 2.15 | 1.98 | 1.95 | 1.53 | 0.32 |
| Relaxed Sampling | DARE | 1.59 | 1.64 | 1.68 | 1.19 | 1.42 | 2.00 | 2.00 | 1.96 | 0.37 | 0.00 |
| | SFT | 1.96 | 1.85 | 1.84 | 1.97 | 1.69 | 2.28 | 2.12 | 2.05 | 1.89 | 0.58 |
| | WordKD | 1.84 | 1.77 | 1.78 | 1.84 | 1.56 | 2.17 | 2.05 | 1.99 | 1.68 | 0.25 |
| | TVDKD | 1.96 | 1.81 | 1.81 | 1.90 | 1.55 | 2.26 | 2.08 | 2.02 | 1.80 | 0.29 |
| | SeqKD | 1.91 | 1.87 | 1.90 | 2.03 | 2.05 | 2.22 | 2.13 | 2.10 | 1.98 | 1.22 |
| | AtSpeed-S | **2.16** | **1.94** | 1.91 | 2.04 | 2.13 | **2.44** | **2.19** | 2.10 | 1.98 | 1.28 |
| | AtSpeed-R | 2.10 | 1.92 | **2.00** | **2.05** | **2.20** | 2.40 | 2.18 | **2.17** | 1.98 | **1.35** |

Table 5: Full results of ranking performance in terms of Recall and NDCG between the baselines and AtSpeed instantiated on LC-Rec (LLaMA-7B) on Beauty and Games datasets.

| Beauty | | | | | |
|---|---|---|---|---|---|
| | Method | Recall@5 | Recall@10 | NDCG@5 | NDCG@10 |
| Without SD | Target LLM (Top$K$) | 0.0056 | 0.0098 | 0.0051 | 0.0066 |
| | Target LLM (Sampling) | 0.0056 | 0.0082 | 0.0043 | 0.0066 |
| Relaxed Sampling Verification | DARE | 0.0059 | 0.0102 | 0.0044 | 0.0060 |
| | SFT | 0.0057 | 0.0091 | 0.0041 | 0.0063 |
| | WordKD | 0.0066 | 0.0105 | 0.0043 | 0.0058 |
| | TVDKD | 0.0057 | 0.0083 | 0.0045 | 0.0054 |
| | SeqKD | 0.0055 | 0.0116 | 0.0045 | 0.0067 |
| | **AtSpeed-S** | **0.0060** | **0.0096** | **0.0046** | **0.0060** |
| | **AtSpeed-R** | **0.0058** | **0.0092** | **0.0049** | **0.0063** |
| | Average | 0.0059 | 0.0098 | 0.0045 | 0.0061 |
| Games | | | | | |
| | Method | Recall@5 | Recall@10 | NDCG@5 | NDCG@10 |
| Without SD | Target LLM (Top$K$) | 0.0074 | 0.0125 | 0.0065 | 0.0083 |
| | Target LLM (Sampling) | 0.0075 | 0.0115 | 0.0066 | 0.0079 |
| Relaxed Sampling Verification | DARE | 0.0076 | 0.0119 | 0.0065 | 0.0080 |
| | SFT | 0.0073 | 0.0112 | 0.0060 | 0.0074 |
| | WordKD | 0.0072 | 0.0113 | 0.0058 | 0.0073 |
| | TVDKD | 0.0069 | 0.0108 | 0.0061 | 0.0074 |
| | SeqKD | 0.0071 | 0.0110 | 0.0059 | 0.0073 |
| | **AtSpeed-S** | **0.0080** | **0.0131** | **0.0068** | **0.0085** |
| | **AtSpeed-R** | **0.0076** | **0.0123** | **0.0063** | **0.0080** |
| | Average | 0.0074 | 0.0117 | 0.0062 | 0.0077 |

**Detailed Analysis on Recommendation Accuracy.** Based on the results in Table 5, we have following observations: 1) ***The ranking performance under strict top-$K$ verification is lossless.*** This is expected since strict verification only accepts the drafts that perfectly match the top-$K$ sequence from the target LLM. Therefore, we obtain identical generation results with and without speculative decoding under strict verification. Based on lossless results, our proposed method AtSpeed-S achieves up to an average of $1.85\times$ speedup. 2) ***The ranking performance under relaxed sampling verification across different alignment methods only shows limited performance drops***

Table 6: Performance comparison between AtSpeed and baselines on MovieLens-1M dataset. For each verification strategy, the best results are highlighted in bold and the second-best results are underlined.

| MovieLens-1M | | | | | | | | | |
|---|---|---|---|---|---|---|---|---|---|
| Verification | Method | WS@3 | WS@5 | WS@10 | WS@20 | AS@3 | AS@5 | AS@10 | AS@20 |
| **Strict Top-$K$** | DARE | 1.26 | 1.28 | 1.39 | 1.72 | 1.00 | 1.00 | 1.00 | 1.00 |
| | SFT | 1.65 | 1.35 | 1.48 | 1.76 | 1.99 | 1.26 | 1.14 | 1.03 |
| | WordKD | 1.29 | 1.29 | 1.39 | 1.69 | 1.07 | 1.02 | 1.00 | 1.00 |
| | TVDKD | 1.73 | 1.72 | 1.25 | 1.24 | 1.98 | 1.90 | 1.04 | 1.00 |
| | SeqKD | 1.77 | 1.78 | 1.42 | 1.50 | 2.03 | 2.00 | 1.24 | 1.05 |
| | AtSpeed-S | **1.86** | **1.79** | **1.80** | **1.75** | **2.08** | **2.03** | **1.98** | **1.09** |
| | AtSpeed-R | 1.76 | 1.78 | 1.54 | 1.74 | 2.01 | 1.99 | 1.29 | 1.08 |
| **Relaxed Sampling** | DARE | 2.01 | 1.84 | 1.35 | 1.44 | 2.16 | 2.02 | 1.00 | 0.35 |
| | SFT | 2.08 | 2.03 | 2.02 | 1.61 | 2.28 | 2.20 | 2.06 | 0.93 |
| | WordKD | 1.97 | 1.87 | 1.48 | 1.28 | 2.15 | 2.08 | 1.23 | 0.00 |
| | TVDKD | 2.00 | 1.98 | 1.68 | 1.29 | 2.30 | 2.21 | 1.59 | 0.00 |
| | SeqKD | 2.13 | 2.08 | 1.93 | 1.56 | 2.19 | 2.14 | 2.01 | 0.78 |
| | AtSpeed-S | 2.23 | 2.16 | 2.11 | 1.65 | 2.41 | 2.33 | **2.16** | **1.02** |
| | AtSpeed-R | **2.24** | **2.22** | **2.14** | **1.64** | **2.44** | **2.38** | 2.15 | 0.95 |
| Goodreads | | | | | | | | | |
| Verification | Method | WS@3 | WS@5 | WS@10 | WS@20 | AS@3 | AS@5 | AS@10 | AS@20 |
| **Strict Top-$K$** | DARE | 1.30 | 1.32 | 1.44 | 1.75 | 1.00 | 1.00 | 1.00 | 1.00 |
| | SFT | 1.83 | 1.81 | 2.17 | 2.46 | 2.04 | 1.98 | 1.72 | 1.07 |
| | WordKD | 1.83 | 1.92 | 2.07 | 2.38 | 2.00 | 1.96 | 1.58 | 1.00 |
| | TVDKD | 1.89 | 1.93 | 2.17 | 2.46 | 2.07 | 1.97 | 1.70 | 1.07 |
| | SeqKD | 1.82 | 1.89 | 2.19 | 2.48 | 2.00 | 1.96 | 1.73 | 1.08 |
| | AtSpeed-S | **2.25** | **2.26** | **2.20** | 2.48 | **2.32** | **2.18** | **1.81** | 1.08 |
| | AtSpeed-R | 2.11 | 2.07 | 2.20 | **2.49** | 2.24 | 2.09 | 1.80 | **1.12** |
| **Relaxed Sampling** | DARE | 1.84 | 1.83 | 1.35 | 1.43 | 2.06 | 2.02 | 1.00 | 0.35 |
| | SFT | 2.15 | 2.09 | 1.70 | 1.91 | 2.27 | 2.08 | 1.01 | 0.10 |
| | WordKD | 2.01 | 2.04 | 1.68 | 1.92 | 2.15 | 2.05 | 1.00 | 0.15 |
| | TVDKD | 2.27 | 2.22 | 1.71 | 2.02 | 2.36 | 2.18 | 1.03 | 0.25 |
| | SeqKD | 1.90 | 1.96 | 1.66 | 1.85 | 2.08 | 2.01 | 1.00 | 0.02 |
| | AtSpeed-S | 2.18 | 2.13 | 1.71 | 1.93 | 2.28 | 2.12 | 1.02 | 0.17 |
| | AtSpeed-R | **2.45** | **2.39** | **1.77** | **2.36** | **2.50** | **2.32** | **1.10** | **0.87** |

*compared to the target LLM's top-$K$ results* (comparable performance on AtSpeed-S, AtSpeed-R and "Average" line), which is consistent with the results in Table 1. This also meets our expectations since the sampling-based verification ensures the approximately equivalent distribution between the SD output and target LLM output under sampling-based generation. We calculate the average over all methods for comparison because we care about how relaxed sampling verification affects the recommendation accuracy. In other words, baseline draft models are also expected to show limited ranking performance drop even if they are less aligned with the target LLM and have a relatively low speedup (*e.g.,* SFT on Beauty). 3) *Compared to NDCG, the Recall under relaxed sampling verification usually achieves comparable or even better values than that of the target LLM.* This is reasonable since this work aims to align the top-$K$ sequence distribution between the draft model and the target LLM. We emphasize the top-$K$ drafted sequence to be accepted with a higher acceptance rate (*i.e.,* a high recall of top$K$ sequences), which does not explicitly require the draft model to distinguish the ranking between top-$K$ sequences (potentially lead to relatively limited performances in terms of NDCG). Nonetheless, it is worth pursuing the non-trivial explicit probability ordering during alignment, which we consider leaving for further exploration in future work.

**Additional Results on MovieLens-1M[9] and Goodreads Datasets[10].** To evaluate the generalization ability of AtSpeed across different domains, we compare our proposed method with baselines on the MovieLens-1M and Goodreads datasets. The results are presented in Table 6. From the results, we can observe that 1) AtSpeed-S and AtSpeed-R outperforms baselines in most cases under strict top-$K$ verification and relaxed sampling verification, respectively. This validate the effectiveness of our proposed method on diverse datasets and is consistent with the observations on Amazon Beauty and Games (see Table 1). 2) The relaxed sampling verification generally shows superior speedup compared to strict top-$K$ verification when $K = 3, 5$, while yields inferior speedup when $K$ is large (*e.g.,* $K = 20$ on MovieLens-1M and Goodreads). One possible reason is that the item size is relatively small on the two datasets ($3,017$ movies and $4,667$ books) compared to Beauty ($12,035$ products) and Games ($17,332$ products), which might results in long-tailed draft distribution, where

---

[9] https://grouplens.org/datasets/movielens/1m/.
[10] https://www.kaggle.com/datasets/bahramjannesarr/goodreads-book-datasets-10m.

top$K$ valid sequences have overwhelmingly high probabilities (*i.e.*, $q \geq p$), thus leading to a high rejection probabilities.

**Speedup of Tree-based Attention.** To analyze the speedup effect of tree-based attention, we instantiate it on the standard LLM decoding with beam search, where each decoding step will utilize the tree-based attention in the LLM forward process. We compare the beam search with and without tree-based attention and report the results in Figure 5 (a). As shown in the figure, we can find that the utilization of tree-based attention indeed exhibits an acceleration effect, corroborating the efficacy of our method. Furthermore, as the draft beam size $N$ grows larger, the speedup of tree-based attention becomes higher. This enhancement is attributed to the substantial reduction in repeated calculations of the same prefix shared across different beam sequences.

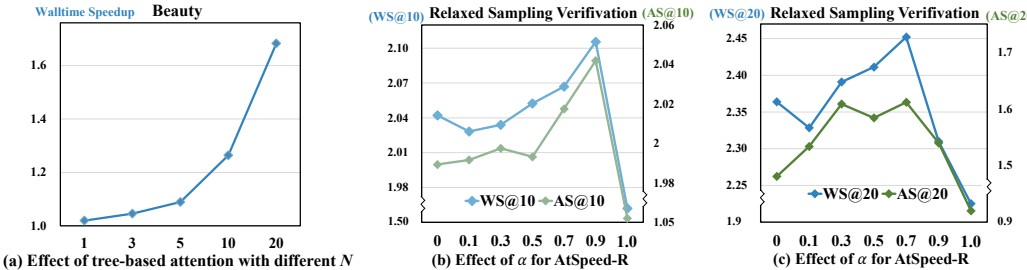

Figure 5: (a) Speedup effect of tree-based attention on Beauty. (b) Performance of AtSpeed on varying hyper-parameters (alignment strength $\alpha$) on Beauty.

**Utilization of Beam Sequences from All Steps.** Table 7 shows inference and training efficiency comparison between AtSpeed training with and without additional data generated from beam search results on diverse steps. From the table, it is observed that incorporating data from diverse generating steps usually enhances the speedup and accept steps, which is attributed to the additional alignment of top-$K$ sequences across different steps. Nevertheless, using additional data could incur increased training time and storage cost. Specifically, utilizing a larger dataset necessitates greater storage capacity and extends the training duration. Despite these additional costs, the marginal improvement in performance is not substantial, leading to a low cost-effectiveness ratio. Therefore, we should carefully consider the balance between the benefits of enhanced alignment against the increased training resource burden.

Table 7: Comparison between AtSpeed with and without additional training data regarding decoding efficiency, training time efficiency, and storage efficiency. "w/ AD" denotes AtSpeed trains the draft model with additional data generated from different steps.

| Verification | Method | WS@10↑ | WS@20↑ | AS@10↑ | AS@20↑ | Time Cost (h)↓ | Storage Cost (GB)↓ |
|---|---|---|---|---|---|---|---|
| **Strict Top-$K$** | AtSpeed-S | 1.87 | 1.84 | 1.64 | 0.57 | 2.56 | 14.68 |
| | AtSpeed-S w/ MD | $1.87^{\uparrow 0.00}$ | $1.66^{\downarrow 0.18}$ | $1.60^{\downarrow 0.04}$ | $0.65^{\uparrow 0.08}$ | $4.10^{\uparrow 1.54}$ | $28.09^{\uparrow 13.41}$ |
| **Relaxed Sampling** | AtSpeed-R | 2.09 | 2.44 | 2.01 | 1.61 | 2.63 | 14.68 |
| | AtSpeed-R w/ MD | $2.16^{\uparrow 0.07}$ | $2.42^{\downarrow 0.02}$ | $2.08^{\uparrow 0.07}$ | $1.65^{\uparrow 0.04}$ | $3.00^{\uparrow 0.37}$ | $28.09^{\uparrow 13.41}$ |

**Hyper-parameter Analysis. 1) Effect of alignment strength $\alpha$.** Figure 5(b-c) presents the results indicating the impact of alignment strength on AtSpeed-R. As the value of $\alpha$ increases, the overall trend of accept steps exhibits an upward trajectory from $0.1$ to $0.9$, which validates the necessity of alignment to improve the performance. Nevertheless, compared to AtSpeed-S (Figure 4(b-c)), we can find that only extremely large $\alpha$ (*e.g.,* 1) would hurt the performance of AtSpeed-R while setting $\alpha$ to $0.7$ for AtSpeed-S already cause a performance drop. We suspect the different behaviors are due to the different scales between $L_{\text{Align-S}}$ and $L_{\text{Align-R}}$. Specifically, given the sequence distribution from draft model $q$ and that from target LLM $p$, we have RKLD $q \log \frac{q}{p}$ in AtSpeed-S and TVD $\frac{|q-p|}{2}$ in AtSpeed-R to align the two models. When sequence probability $q$ becomes small as the sequence length increases, $\frac{q}{p}$ becomes very large. It is possible for AtSpeed-S to give a larger loss value compared to AtSpeed-R (*i.e.,* $\frac{|p-q|}{2}$). Therefore, for the same $\alpha < 1$, the strength that is too large for AtSpeed-S might still benefit alignment for AtSpeed-R.

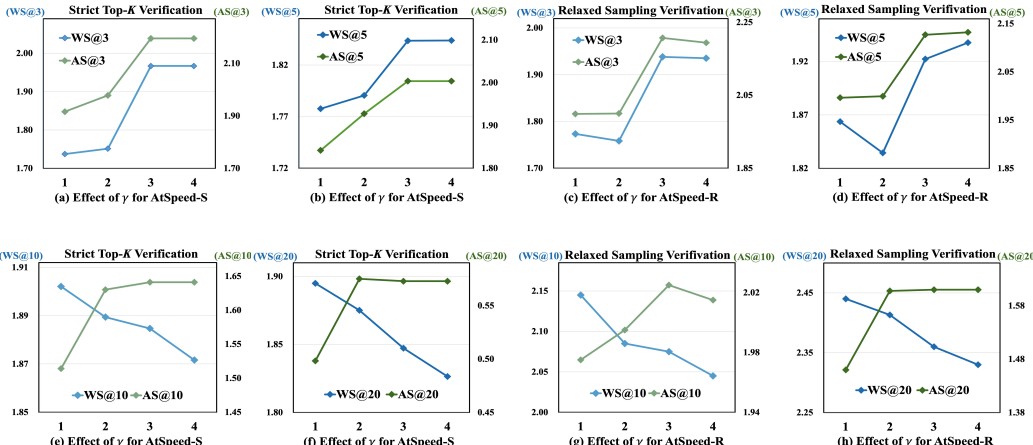

Figure 6: Performance of AtSpeed with different draft length $\gamma$ on Beauty.

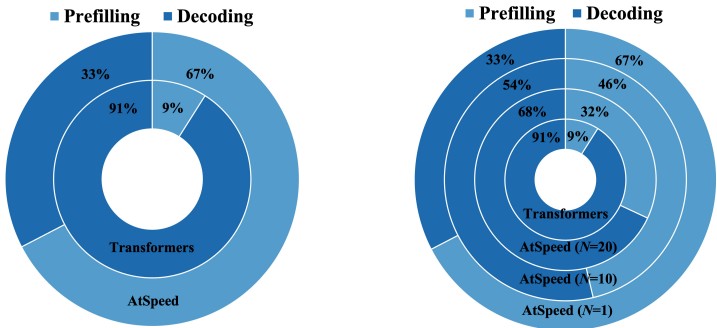

Figure 7: Comparison of the time cost ratio between the prefilling and decoding. The time cost of prefilling is the same across different methods.

**2) Effect of draft length** $\gamma$**.** Figure 6 presents the impact of varying $\gamma$ on the performance of AtSpeed. Despite acceptance generally increases with $\gamma$, speedup does not consistently improve with $\gamma$. The trend of speedup in relation to $\gamma$ is influenced by draft beam size parameter $N$. Specifically, when $N$ is small, speedup tends to increase with $\gamma$; conversely, when $N$ is large, speedup tends to decrease with $\gamma$. This variation is primarily attributed to the computational overhead during the draft phase, which escalates with larger beam size, thereby diminishing the acceleration benefits. The key determinant of the draft phase's computational cost is the constraint search process.

**Case Study.** As shown in Figure 7, our proposed methodology achieves a significant reduction in decoding time. Since the prefilling time is the same between target LLM with and without SD, AtSpeed boosts the LLM decoding by reducing the percentage of decoding by 58% ($N$=1), 37% ($N$=10), and 23% ($N$=20).

