# OpenReview forum: "Efficient Inference for Large Language Model-based Generative Recommendation"
_ICLR.cc/2025/Conference — ICLR 2025 Poster_

### Official Review · Reviewer_Up12 · 2024-10-22

**Soundness:** 3
**Presentation:** 2
**Contribution:** 3
**Rating:** 6
**Confidence:** 4

**Summary:**

This paper introduces a speculative decoding approach for generative recommendation. In traditional speculative decoding, a cheap draft LLM is given a prefix and at each decoding step it suggests $N$ tokens among which the token predicted by the target LLM should be. This enables the target model to verify all the drafted predictions in parallel through a single call, conditioning on the draft model's previous predictions. The sequence of drafted predictions is then accepted up to the decoding step where the draft model and the target model disagree in their predictions.

In top-$K$ generative recommendation, a beam search is performed to generate the $K$ item identifiers. Applying speculative decoding to this problem then requires the $N$ tokens suggested by the draft model to contain the $K$ tokens predicted by the target model to accept the current decoding step. This condition is hard to fulfill in practice. To address this, the paper defines a framework named AtSpeed, which trains the draft model to be more aligned with the target model and optionally relaxes the acceptance condition to allow the $N$ drafted tokens to contain tokens that are similar (rather than identical) to the $K$ target tokens. AtSpeed was validated through experiments on the Amazon Beauty and Games datasets.

**Strengths:**

- The approach proposed for applying speculative decoding to generative recommendation is both novel and intuitive.
- The experiments are comprehensive and provide convincing evidence of the benefits of AtSpeed from an empirical standpoint.
- The source code was made available.

**Weaknesses:**

- The paper suffers from an overall lack of mathematical rigor, both in the proofs and notations, which raises concerns about the theoretical soundness of the approach. In particular, some key formulas, such as the proposed alignment objective, appear to be either incorrect or lack sufficient derivation to validate their correctness (see specific points in the 'Questions' section).
- The paper is overall lacking polish and contains multiple typos in mathematical formulas, which makes it hard for the reader to follow the details of the approach. Thorough proofreading would be needed to improve the paper's readability. Certain parts of the paper could also benefit from being further clarified, as pointed out in the 'Questions' section.

**Questions:**

- In Section 3.1, paragraph "Acceptance Rate", the condition given for having $\beta = 1$, namely that $p(\mathbf{y}) \geq p(\mathbf{y}_K)$ for all $\mathbf{y}$ in $\mathcal{Y}_q$, doesn't seem to be fulfillable. By definition of $\mathbf{y}_K$, there exists only $K-1$ sequences with greater probability according to $p$, yet the condition requires that all $N$ ($\geq K$) sequences in $\mathcal{Y}_q$ have a probability greater than $\mathbf{y}_K$. The only possibility for this condition to be realized is if $\mathcal{Y}_q$ contains duplicates, which should not happen in the case of beam search.
- Derivation of Eq 15 in App A.2 (which leads to the definition of the alignment objective in Eq 3) seems incorrect or at least misses important steps to be understandable for the reader. More steps or explanations are needed.
- Eq 3 introduces $\mathcal{D'}$ in which $\mathcal{Y}$ is the top-$K$ of a mixture of $q$ and $p$. Following Gu et al (2024) is mentioned as the motivation for this, but it would be appreciated to have more intuitions on this choice. Moreover, the $\mathcal{D'}$ used later in the relaxed alignment objective (Eq 8) is different. What is the rationale to have different $\mathcal{D'}$ in the strict and relaxed objectives?
- What loss is used in practice for $\mathcal{L}_{Rec}$? It is only mentioned to be a recommendation loss but additional details would be appreciated (at least in the appendix).
- The relaxation sampling verification strategy misses some intuition: why is $p(\mathbf{y}) \geq q(\mathbf{y})$ the right criterion for accepting a sequence $\mathbf{y}$? Why are such $\mathbf{y}$'s good candidates? Moreover, in the case of rejection, $\mathbf{y}$ is drawn from $p' = norm(max(0, p(\mathbf{y}) - q(\mathbf{y})))$ but it is unclear whether this is an actual distribution and what the norm operator exactly consists of. Is $p'$ denoting the uniform distribution over the $\mathbf{y}$'s such that $p(\mathbf{y}) \geq q(\mathbf{y})$? If so, how does one sample from this in practice? The definition of this distribution would deserve more clarifications and details.
- The "with-replacement sampling approximation" paragraph in App A.2 is difficult to follow so it would benefit from being reworked.
- The proof of Lemma 1 in App A.2, which corresponds to Lemma 3.3 from Leviathan et al (2023), is missing the step with the term $1 - \sum_{\mathbf{y}} \frac{p(\mathbf{y})+q(\mathbf{y})-|p(\mathbf{y})-q(\mathbf{y})|}{2}$.
- In Eq 7, what does $\sum_K$ denote? It seems like there is an index variable missing there.
- The strategy relying on tree-based attention to speed up inference would benefit from being described in more details (at least in Appendix). Section 3.3 only refers to Figure 2(c) to describe the strategy, but this figure is not self-explanatory.
-  The WS metric represents the walltime speedup, but with respect to which baseline? I assume this is in comparison to directly running the target model without speculative decoding, but it would be helpful for the reader to mention this when defining the metric.
- Table 1 reports the results for AtSpeed-S and AtSpeed-R on both the strict and the relaxed settings. How can AtSpeed-S be applied to the relaxed setting and AtSpeed-R to the strict setting?
- In Figure 5 of the Appendix, it seems that a larger value for $\alpha$ is always beneficial for AtSpeed-R, whereas Figure 3 (c) showed that $\alpha$ should neither be too small nor too large for AtSpeed-S. Are there any intuitions on these different behaviors between AtSpeed-R and AtSpeed-S?

---

> ### Author Response · Authors · 2024-11-19
> **Reply to Question 1-2**
>
> Dear Reviewer Up12,
>
> Thanks for your comments. Your review is very detailed and thorough. we greatly appreciate it! We have provided detailed clarification, explanations, intuitions behind method design, and step-by-step derivation to address each of your concerns. We have also updated our manuscript accordingly (marked in blue). If we have any misunderstanding, please feel free to leave further comments. We eagerly anticipate our discussion with you!
>
> > **Q1: In Section 3.1, paragraph "Acceptance Rate", the condition given for having $\beta = 1$, namely that $p(\mathbf{y}) \geq p(\mathbf{y}_K)$ for all $\mathbf{y}$ in $\mathcal{Y}_q$, doesn't seem to be fulfillable. By definition of $\mathbf{y}_K$, there exists only $K-1$ sequences with greater probability according to $p$, yet the condition requires that all $N$ ($\geq K$) sequences in $\mathcal{Y}_q$ have a probability greater than $\mathbf{y}_K$. The only possibility for this condition to be realized is if $\mathcal{Y}_q$ contains duplicates, which should not happen in the case of beam search.**
>
>
> **Reply**: Thank you for pointing out this problem. The acceptance rate in Section 3.1 indeed would be unfulfillable if $N>K$. The only possibility to achieve this condition is when $N=K$ and the top-$K$ ($N=K$) token sequences from the draft model are also the top-$K$ token sequences from the target model. To correct this, we revise the condition as:
>
> $\beta=1$ if $\exists \mathcal{Y}'\_{q}\in\mathcal{Y}\_{q}$ such that $p(\mathbf{y})\ge p({\mathbf{y}}\_k)$ for $\forall \mathbf{y}\in\mathcal{Y}'\_{q}$, where $|\mathcal{Y}'_{q}| = K$, $K$ is the target LLM beam size and $\mathbf{y}_k$ is the sequence that has the $K$-th highest probability in $p$.
>
> Based on this condition, the alignment objective as in Eq.(2) is unaffected, which aims to encourage the top-$N$ generated sequences from the target model to have high probabilities in the target LLM distribution (a high $\frac{p(\mathbf{y})}{p(\mathbf{y}_K)}$). Precisely, if the drafted sequence fails to be in the top-$K$ sequences according to $p$ (i.e., $p(\mathbf{y})<p(\mathbf{y}_K)$), we still encourage it to have a high probability closer to $\mathbf{y}_K$ (i.e., a high $\frac{p(\mathbf{y})}{p(\mathbf{y}_K)}$), aiming to push the top-$N$ distribution of draft model to cover the top-$K$ distribution of the target LLM.
>
>
> > **Q2: Eq 3 introduces $\mathcal{D'}$ in which $\mathcal{Y}$ is the top-$K$ of a mixture of $q$ and $p$. Following Gu et al (2024) is mentioned as the motivation for this, but it would be appreciated to have more intuitions on this choice.**
>
> **Reply**: Thanks for your valuable suggestions.
> For the choice of the mixture of $q$ and $p$ in Eq.(3), we have two main considerations:
>
> - ***1) Higher training efficiency***. The original $\mathcal{D}$ should include the sequences sampled from the draft model (i.e., $\mathbf{y}\in\mathcal{Y}_q$ in Eq.(2)). However, training over draft model-generated sequences requires the online learning process, which will lead to high computational costs and time costs for training. Because we need to sample the sequences continuously from the draft models for every epoch or even every batch during the alignment training process. Therefore, to alleviate the reliance on the frequent sampling from the draft model $q$, we consider using the mixture of draft model $q$ and target LLM $p$. In practice, the mixture of $p$ and $q$ is achieved by alternating the sequences sampled from $p$ and $q$, where the alternating sessions are controlled by the mixture coefficient $\lambda$. Since the target LLM-generated sequences can be pre-stored, we can improve the training efficiency by significantly reducing the number of sampling sequences from draft model.
>
> - ***2) Mitigation of low-quality training data issue.*** During alignment training, the draft model might generate low-quality sequence (e.g., repeated phrases). However, such low-quality sequences will be rejected by the target LLM during inference since they are invalid identifiers. As such, pushing $q(\mathbf{y})$ closer to $p(\mathbf{y})$ over low-quality sequences will lead to unnecessary and suboptimal alignment. Therefore, we consider utilizing LLM-generated data to ensure high-quality training data for alignment training.

---

> ### Author Response · Authors · 2024-11-19
> **Reply to Question 3-6**
>
> > **Q3: Moreover, the $\mathcal{D'}$ used later in the relaxed alignment objective (Eq 8) is different. What is the rationale to have different $\mathcal{D'}$ in the strict and relaxed objectives?**
>
> **Reply**: Thanks for your insightful question. The different choices of $\mathcal{D}’$ essentially result from their different expressions of acceptance rate and different alignment objectives. Specifically,
>
> - For the *strict verification*, since we hope every drafted sequence can achieve high probability in target LLM distribution, the alignment objective is maximizing $\frac{p(\mathbf{y})}{p(\mathbf{y}_K)}$ over the output distribution of draft model. Based on the consideration for higher training efficiency and mitigation of the low-quality training data issue as mentioned in the previous response, we adopt the mixed sampling for AtSpeed-S.
>
> - For the *relaxed verification*, we are inspired by Eq.(1) in [1] to maximize the sequence acceptance rate over the output distribution of target LLM. This is because, with each approximated with-replacement sequence sampling (Eq.(6)), the expectation over the drafted sequence is essentially the expectation over the model vocabulary given the prefix. While the vocabulary is the same for both the draft model and the target LLM, the prefix of $\mathbf{y}$ in Eq.(6) is accepted by the target LLM in inference and thus follows the with-replacement sampling distribution of target LLM. Therefore, the $\mathbf{y}$ in Eq.(7) of our paper is considered to be sampled over the target LLM distribution for AtSpeed-R.
>
>
> [1] Yongchao Zhou, et al. DistillSpec: Improving Speculative Decoding via Knowledge Distillation. ICLR 2024.
>
>
>
> > **Q4:What loss is used in practice for LRec? It is only mentioned to be a recommendation loss but additional details would be appreciated (at least in the appendix)**
>
> **Reply**: Thanks for your valuable suggestions. The recommendation loss used in our work is defined as
> $$
> \mathcal{L}\_\text{Rec} = - \frac{1}{N} \sum\_{(\mathbf{x}, \mathbf{y}) \sim \mathcal{D}} \sum\_{t=1}^{|\mathbf{y}|} \log \mathcal{M}\_p({y}\_t|\mathbf{y}\_{<t}, \mathbf{x}),
> $$
>
> where $\mathbf{x}$ is user’s historical interactions, $\mathbf{y}$ is the user’s next interacted item identifier,  and $D=\\{(\mathbf{x}, \mathbf{y})\\}$ denotes the original recommendation dataset. We have also added this to our manuscript in Appendix.
>
>
> > **Q5: The relaxation sampling verification strategy misses some intuition: why is $p(\mathbf{y}) \geq q(\mathbf{y})$ the right criterion for accepting a sequence $\mathbf{y}$? Why are such $\mathbf{y}$'s good candidates? Moreover, in the case of rejection, $\mathbf{y}$ is drawn from $p' = norm(max(0, p(\mathbf{y}) - q(\mathbf{y})))$ but it is unclear whether this is an actual distribution and what the norm operator exactly consists of. Is $p'$ denoting the uniform distribution over the $\mathbf{y}$'s such that $p(\mathbf{y}) \geq q(\mathbf{y})$? If so, how does one sample from this in practice? The definition of this distribution would deserve more clarifications and details.**
>
> **Reply**: Thanks for your valuable questions and we appreciate your careful review.
>
> - The intuition behind accepting sequence $\mathbf{y}$ if $p(\mathbf{y}) \ge p(\mathbf{y})$ is that, given a candidate sequence $\mathbf{y}$ with a high probability according to the draft model, if the target LLM is even more confident of generating the sequence (i.e., $p(\mathbf{y}) \ge p(\mathbf{y})$), the candidate sequence is likely to be generated by the LLMs and should be accepted. We have added the intuition explanation in our updated manuscript.
>
> - In the case of rejection, the normalized probability distribution is adjusted as follows:
> $$
> p’ = \text{norm}(\max (0, p(\mathbf{y})- q(\mathbf{y})))
> = \frac{max(0, p(\mathbf{y})-q(\mathbf{y}))}{\sum_{\mathbf{y}}{max(0, p(\mathbf{y})-q(\mathbf{y}))}}
> $$
> We then sample a sequence from this normalized distribution to replace the rejected sequence. The clarification of normalized distribution is added in our updated manuscript.
>
>
> > **Q6: The proof of Lemma 1 in App A.2, which corresponds to Lemma 3.3 from Leviathan et al (2023), is missing the step with the term $1 - \sum_{\mathbf{y}} \frac{p(\mathbf{y})+q(\mathbf{y})-|p(\mathbf{y})-q(\mathbf{y})|}{2}$**.
>
> **Reply:** Thank you for pointing out this typo. Your careful review is greatly appreciated. We have revised it in our updated manuscript.

---

> ### Author Response · Authors · 2024-11-19
> **Reply to Question 7-9**
>
> > **Q7: In Eq 7, what does $\sum_K$ denote? It seems like there is an index variable missing there.**
>
> **Reply**: Thanks for your question. The $\sum_{K}$ in Eq.(7) is written in the form of summation to denote the derivation from the joint distribution of $K$ sequences, i.e., $K$ in $\log \beta = - \sum_{K} \log \text{TVD}(q, p)$ in the “Alignment Objective” paragraph of Section 3.2. To clarify this, we explained the meaning of $K$ in the context of Eq.(7) in line 281-284 in our submitted manuscript, i.e., line 284-286 in our updated manuscript (“Eq.(7) minimizes TVD with the strength of K for the same sequence. Nevertheless, considering the beam search with sampling would obtain $K$ different sequences, we alternatively leverage the top-$K$ sequences from the target LLM”). This also partially explains our choice of $\mathcal{D}’$ to sample Top-$K$ sequences from the target LLM for the alignment training under relaxed sampling verification.
>
> > **Q8: The strategy relying on tree-based attention to speed up inference would benefit from being described in more details (at least in Appendix). Section 3.3 only refers to Figure 2(c) to describe the strategy, but this figure is not self-explanatory.**
>
> **Reply**: Thanks for your insightful comments. We provide more details and a concrete example of how tree-based attention is implemented in practice. We have also added them to our updated manuscript.
>
> The main idea of tree-based attention is to eliminate the repeated self-attention calculation for the same prefix of the beam sequences during beam search.
>
> - **Detailed explanation of tree-based attention.** In tree-based attention strategy, we first compress the N beam sequences into a single flattened sequence, and then construct the sparse tree-based attention mask for efficient target LLM verification. More precisely, given $\gamma N $ drafted beam sequences with different lengths, where $N$ is the draft beam size and $\gamma$ is the number of drafted beam steps,
>
> 	- we first flattened the beam sequences by sequentially adding the newly generated tokens from the beam sequences at each step. We denote the length of the flattened sequence as $L_f$.
>
> 	- Then, based on the flattened sequence, we construct an attention mask with the shape of $L_f \times L_f$. Specifically, each row in attention mask represents a specific beam sequence, and for each row, we set the corresponding column of the last token and the preceding tokens in the beam sequence as 1, otherwise 0.
> - **Example**. Here, we give a concrete example to illustrate how tree-based attention is implemented to save repeated calculation.
> We set the beam size of 2 and collect the beam sequences of 3 steps. The collected beam sequences are as follows:
>
> 	> step 1 beam 1: ``a1``
>
> 	> step 1 beam 2: ``a2``
>
> 	> step 2 beam 1: ``a1 b1``
>
> 	> step 2 beam 2: ``a2 b2``
>
> 	> step 3 beam 1: ``a1 b1 c1``
>
> 	> step 3 beam 2: ``a1 b1 c2``
>
> 	The flattened sequence will be ``a1 a2 b1 b2 c1 c2``.
> 	The constructed tree-based attention is shown in Figure 2(c) of our manuscript, where each row represents a specific beam sequence. For each row, the tickled cell represents the preceding tokens and the last tokens of each beam. And the different colors represent the different steps of beam search.
> 	This flattened sequence and sparse tree-based attention enable efficient verification since it saves the repeated calculation of the same prefix across different beam sequences, e.g., ``a1b1``.
>
>
> > **Q9: The WS metric represents the walltime speedup, but with respect to which baseline? I assume this is in comparison to directly running the target model without speculative decoding, but it would be helpful for the reader to mention this when defining the metric.**
>
> **Reply**: Thanks for your valuable question and suggestion. Your understanding is correct. The walltime speedup is compared to the execution of original target LLM without speculative decoding. And the walltime speedup is defined as $WS=\frac{T}{T'}$, where T is the time for running target LLM without speculative decoding and T’ is the time for running LLM with speculative decoding with a specific draft model. We have also added the clarification to our updated manuscript.

---

> ### Author Response · Authors · 2024-11-19
> **Reply to Question 10-11**
>
> > **Q10: Table 1 reports the results for AtSpeed-S and AtSpeed-R on both the strict and the relaxed settings. How can AtSpeed-S be applied to the relaxed setting and AtSpeed-R to the strict setting?**
>
> **Reply**: Thanks for your valuable questions. The AtSpeed-S and AtSpeed-R are essentially two alignment methods to train a draft model. After the alignment training, the well-trained draft model can be applied to both strict top-$K$ and relaxed sampling verification. The difference between AtSpeed-S and AtSpeed-R is that the training loss is specifically designed to improve the acceptance rate for strict top-$K$ and relaxed sampling verification, respectively. Therefore, the alignment effectiveness might be different, where AtSpeed-S has a better alignment under strict verification while AtSpeed-R achieves a better alignment under relaxed sampling verification (empirical results also validate this as shown in Table 1 in the manuscript).
>
> > **Q11: In Figure 5 of the Appendix, it seems that a larger value for α is always beneficial for AtSpeed-R, whereas Figure 3 (c) showed that α should neither be too small nor too large for AtSpeed-S. Are there any intuitions on these different behaviors between AtSpeed-R and AtSpeed-S?**
>
> **Reply**: Thanks for your insightful questions.
> We suspect the different behaviors are due to the different scales between $L_\text{Align-S}$ and $L_\text{Align-R}$. Specifically, given the sequence distribution from draft model $q$ and that from target LLM $p$, we have RKLD $q\log \frac{q}{p}$ in AtSpeed-S and TVD $\frac{|q-p|}{2}$ in AtSpeed-R to align the two models. When sequence probability $q$ becomes small as the sequence length increases, $\frac{q}{p}$ becomes very large. It is possible for AtSpeed-S to give a larger loss value compared to AtSpeed-R (i.e., $\frac{|p-q|}{2}$). Therefore, for the same $\alpha=0.7$, the strength is still not too large to hurt the alignment for AtSpeed-R.
>
>
> - *Empirical results.* To validate this, we continue increasing $\alpha$ to 1 for AtSpeed-R and present the results in the following table. We can find that AtSpeed-R yields lower WS@20 and AS@20 when we increase $\alpha$ to 0.9. Besides, we have the worst performance when $\alpha=1$, i.e., removing the recommendation loss. This overall behavior is consistent with AtSpeed-S, where the extremely large $\alpha$ will hurt the alignment. We have also updated results and observations into our latest manuscript.
>
>
> |           | $\alpha$ | WS@10  | WS@20  | AS@10  | AS@20  |
> |-----------|----------|--------|--------|--------|--------|
> | AtSpeed-R |        0 | 2.0421 | 2.3801 | 1.9893 | 1.4866 |
> |           |      0.1 | 2.0282 | 2.3284 | 1.9916 | 1.5336 |
> |           |      0.3 | 2.0340 | 2.3909 | 1.9975 | 1.6085 |
> |           |      0.5 | 2.0270 | 2.3665 | 1.9932 | 1.5840 |
> |           |      0.7 | 2.0394 | 2.4521 | 2.0144 | 1.6116 |
> |           |      0.9 | 2.1055 | 2.3100 | 2.0419 | 1.5398 |
> |           |        1 | 1.5002 | 1.9245 | 1.0639 | 0.9491 |

---

> ### Author Response · Authors · 2024-11-19
> **Detailed Derivation of Alignment Objective of AtSpeed-S (Step 1)**
>
> > **Q12: Derivation of Eq 15 in App A.2 (which leads to the definition of the alignment objective in Eq 3 seems incorrect or at least misses important steps to be understandable for the reader. More steps or explanations are needed.**
>
> **Reply**: Your thorough review is greatly appreciated. We provide detailed step-by-step derivations of Eq.(3) to facilitate the reader’s understanding. And we have also updated the Appendix with the detailed derivation in our revised manuscript.
>
> ***Step1. Decompose $p(\mathbf{y})$ and $q(\mathbf{y})$.***
>
> We start from Eq.(2) in our paper, i.e., the alignment objective under strict top-$K$ verification, as
>
> $$
> -\sum\_{\mathbf{y}}q(\mathbf{y}) \log \frac{q(\mathbf{y})}{p(\mathbf{y})} + \sum\_{\mathbf{y}} q(\mathbf{y}) \log \frac{q(\mathbf{y})}{p(\mathbf{y}\_K)}
> $$
>
> where $p(y\_K)=p(\mathbf{y}\_{K,t}|\mathbf{y}\_{K,<t})$. Denote $c\_{<t}=(\mathbf{x},\mathbf{y}\_{<t})$, referred to as context, we have  $q(\mathbf{y})=\prod\_{t}q(y\_t|\mathbf{x},\mathbf{y}\_{<t})=\prod\_{t}q(y\_t|c\_{<t})$ and $p(\mathbf{y})=\prod\_{t}p(y\_t|\mathbf{x},\mathbf{y}\_{<t})=\prod\_{t}p(y\_t|c\_{<t})$. We then can substitute $q(\mathbf{y})$ with $\prod\_{t}q(y\_t|c\_{<t})$ and $p(\mathbf{y})$ with $\prod\_{t}p(y\_t|c\_{<t})$ and rewrite the objective as:
>
> $$
> \begin{aligned}
> &-\sum\_{\mathbf{y}}q(\mathbf{y}) \log \frac{q(\mathbf{y})}{p(\mathbf{y})} + \sum\_{\mathbf{y}} q(\mathbf{y}) \log \frac{q(\mathbf{y})}{p(\mathbf{y}\_K)} \\\\
>   =& -\sum\_{\mathbf{y}} q(\mathbf{y}) \log \frac{\prod\_t q(y\_t|c\_{<t})}{\prod\_t p(y\_t|c\_{<t})} + \sum\_{\mathbf{y}} q(\mathbf{y}) \log \frac{\prod\_t q(y\_t|c\_{<t})}{\prod\_t p(\mathbf{y}\_{K,t}|\mathbf{y}\_{K,<t})} \\\\
> =&  - \sum\_{\mathbf{y}} q(\mathbf{y}) [\log \prod\_t q(y\_t|c\_{<t}) - \log \prod\_t p(y\_t|c\_{<t})] + \sum\_{\mathbf{y}} q(\mathbf{y}) [\log \prod\_t q(y\_t|c\_{<t}) - \log \prod\_t p (\mathbf{y}\_{K,t}|\mathbf{y}\_{K,<t})] \\\\
>     =&  - \sum\_{\mathbf{y}} q(\mathbf{y}) [\sum\_t \log  q(y\_t|c\_{<t}) - \sum\_t \log p(y\_t|c\_{<t})] + \sum\_{\mathbf{y}} q(\mathbf{y}) [\sum\_t \log q(y\_t|c\_{<t}) - \sum\_t \log p (\mathbf{y}\_{K,t}|\mathbf{y}\_{K,<t})] \\\\
>     =&  - \sum\_{\mathbf{y}} q(\mathbf{y}) \sum\_t \log  \frac{q(y\_t|c\_{<t})}{p(y\_t|c\_{<t})} + \sum\_{\mathbf{y}} q(\mathbf{y}) \sum\_t \log \frac{q(y\_t|c\_{<t})}{p (\mathbf{y}\_{K,t}|\mathbf{y}\_{K,<t})}  \\\\
>     =& - \sum\_t \sum\_{\mathbf{y}} q(\mathbf{y})  \log  \frac{q(y\_t|c\_{<t})}{p(y\_t|c\_{<t})} + \sum\_t \sum\_{\mathbf{y}} q(\mathbf{y})  \log \frac{q(y\_t|c\_{<t})}{p (\mathbf{y}\_{K,t}|\mathbf{y}\_{K,<t})}.
> \end{aligned}
> $$

---

> ### Author Response · Authors · 2024-11-19
> **Detailed Derivation of Alignment Objective of AtSpeed-S (Step 2)**
>
> ***Step2. Decompose $\sum\_{\mathbf{y}}q(\mathbf{y})$ and obtain step-wise alignment.***
>
> Since $\mathbf{y}=(y\_1, y\_2, \dots, y\_n) = (\mathbf{y}\_{<t}, y\_t, \mathbf{y}\_{>t})$ is a sequence, we can decompose $\sum\_{\mathbf{y}} q(\mathbf{y})$ into nested sum
> $\sum\_{\mathbf{y}\_{<t}}
>     \sum\_{y\_t}
> \sum\_{\mathbf{y}\_{>t}}
> q(\mathbf{y}) = \sum\_{\mathbf{y}\_{<t}}
>     \sum\_{y\_t}
> \sum\_{\mathbf{y}\_{>t}}
> q(\mathbf{y}\_{<t}, y\_t, \mathbf{y}\_{>t})$, where the nested sum over three parts of $\mathbf{y}$, i.e., $\mathbf{y}\_{<t}, \mathbf{y}\_t$, and $\mathbf{y}\_{>t}$ can cover all possible sequence $\mathbf{y}$.
>
> Since $q(\mathbf{y}\_{<t}, y\_t, \mathbf{y}\_{>t})=q(\mathbf{y}\_{<t})q(y\_t|c\_{<t})q(\mathbf{y}\_{>t}|c\_{\le t})$, we can rewrite the nested sum of $q(\mathbf{y}\_{<t}, y\_t, \mathbf{y}\_{>t})$ over $\mathbf{y}$:
>
> $$
> \begin{aligned}
> & \sum\_{\mathbf{y}} q(\mathbf{y}) \\\\
>     =&\sum\_{\mathbf{y}\_{<t}}
>     \sum\_{y\_t}
> \sum\_{\mathbf{y}\_{>t}} q(\mathbf{y}\_{<t})q(y\_t|c\_{<t})q(\mathbf{y}\_{>t}|c\_{\le t}) \\\\
> = &\sum\_{\mathbf{y}\_{<t}}
> \sum\_{y\_t} q(\mathbf{y}\_{<t})q(y\_t|c\_{<t})
> \sum\_{\mathbf{y}\_{>t}} q(\mathbf{y}\_{>t}|c\_{\le t}) \\\\
> = &\sum\_{\mathbf{y}\_{<t}} q(\mathbf{y}\_{<t})
>     \sum\_{y\_t} q(y\_t|c\_{<t})
> \sum\_{\mathbf{y}\_{>t}} q(\mathbf{y}\_{>t}|c\_{\le t}).
> \end{aligned}
> $$
>
> We then can substitute $\sum\_{\mathbf{y}} q(\mathbf{y})$ with $\sum\_{\mathbf{y}\_{<t}} q(\mathbf{y}\_{<t})
>     \sum\_{y\_t} q(y\_t|\mathbf{y}\_{<t})
> \sum\_{\mathbf{y}\_{>t}} q(\mathbf{y}\_{>t}|c\_{\le t})$ in previous expansion and obtain:
>
> $$
> \begin{aligned}
>     & \quad - \sum\_t \sum\_{\mathbf{y}} q(\mathbf{y})  \log  \frac{q(y\_t|c\_{<t})}{p(y\_t|c\_{<t})}
>     + \sum\_t \sum\_{\mathbf{y}} q(\mathbf{y})  \log \frac{q(y\_t|c\_{<t})}{p (\mathbf{y}\_{K,t}|\mathbf{y}\_{K,<t})} \\\\
>     & = - \sum\_t
>     [
>     \sum\_{\mathbf{y}\_{<t}} q(\mathbf{y}\_{<t})
>     \sum\_{y\_t} q(y\_t|c\_{<t})
>     \sum\_{\mathbf{y}\_{>t}} q(\mathbf{y}\_{>t}|c\_{\le t})
>     ]
>     \log \frac{q(y\_t|c\_{<t})}{p(y\_t|c\_{<t})}
>     + \sum\_t
>     [
>     \sum\_{\mathbf{y}\_{<t}} q(c\_{<t})
>     \sum\_{y\_t} q(y\_t|c\_{<t})
>     \sum\_{\mathbf{y}\_{>t}} q(\mathbf{y}\_{>t}|c\_{\le t})
>     ]
>     \log \frac{q(y\_t|c\_{<t})}{p (\mathbf{y}\_{K,t}|\mathbf{y}\_{K,<t})} \\\\
>     & = - \sum\_t
>     \sum\_{\mathbf{y}\_{<t}} q(\mathbf{y}\_{<t})
>     \sum\_{y\_t} q(y\_t|c\_{<t})
>     \log \frac{q(y\_t|c\_{<t})}{p(y\_t|c\_{<t})}
>     \sum\_{\mathbf{y}\_{>t}} q(\mathbf{y}\_{>t}|c\_{\le t})
>     + \sum\_t
>     \sum\_{\mathbf{y}\_{<t}} q(\mathbf{y}\_{<t})
>     \sum\_{y\_t} q(y\_t|c\_{<t})
>     \log \frac{q(y\_t|c\_{<t})}{p (\mathbf{y}\_{K,t}|\mathbf{y}\_{K,<t})}
>     \sum\_{\mathbf{y}\_{>t}} q(\mathbf{y}\_{>t}|c\_{\le t}).
> \end{aligned}
> $$
>
> Since $\sum\_{\mathbf{y}\_{>t}} q(\mathbf{y}\_{>t}|c\_{\le t})=1$, we can remove $\sum\_{\mathbf{y}\_{>t}} q(\mathbf{y}\_{>t}|c\_{\le t})$:
>
> $$
> \begin{aligned}
>     & - \sum\_t
>     \sum\_{\mathbf{y}\_{<t}} q(\mathbf{y}\_{<t})
>     \sum\_{y\_t} q(y\_t|c\_{<t})
>     \log \frac{q(y\_t|c\_{<t})}{p(y\_t|c\_{<t})}
>     \sum\_{\mathbf{y}\_{>t}} q(\mathbf{y}\_{>t}|c\_{\le t})
>     + \sum\_t
>     \sum\_{\mathbf{y}\_{<t}} q(\mathbf{y}\_{<t})
>     \sum\_{y\_t} q(y\_t|c\_{<t})
>     \log \frac{q(y\_t|c\_{<t})}{p (\mathbf{y}\_{K,t}|\mathbf{y}\_{K,<t})}
>     \sum\_{\mathbf{y}\_{>t}} q(\mathbf{y}\_{>t}|c\_{\le t}) \\\\
>     =&  - \sum\_t
>     \sum\_{\mathbf{y}\_{<t}} q(\mathbf{y}\_{<t})
>     \sum\_{y\_t} q(y\_t|c\_{<t})
>     \log \frac{q(y\_t|c\_{<t})}{p(y\_t|c\_{<t})}
>     + \sum\_t
>     \sum\_{\mathbf{y}\_{<t}} q(\mathbf{y}\_{<t})
>     \sum\_{y\_t} q(y\_t|c\_{<t})
>     \log \frac{q(y\_t|c\_{<t})}{p (\mathbf{y}\_{K,t}|\mathbf{y}\_{K,<t})} \\\\
>     =&  - \sum\_t
>     \sum\_{\mathbf{y}\_{<t}} q(\mathbf{y}\_{<t})
>     [
>     \sum\_{y\_t} q(y\_t|c\_{<t})
>     \log \frac{q(y\_t|c\_{<t})}{p(y\_t|c\_{<t})}
>     -
>     \sum\_{y\_t} q(y\_t|c\_{<t})
>     \log \frac{q(y\_t|c\_{<t})}{p (\mathbf{y}\_{K,t}|\mathbf{y}\_{K,<t})}
>     ] \\\\
>     =& - \sum\_t \mathbb{E}\_{\mathbf{y}\_{<t}\sim q}
>     [
>     \sum\_{y\_t} q(y\_t|c\_{<t})
>     \log \frac{q(y\_t|c\_{<t})}{p(y\_t|c\_{<t})}
>     -
>     \sum\_{y\_t} q(y\_t|c\_{<t})
>     \log \frac{q(y\_t|c\_{<t})}{p (\mathbf{y}\_{K,t}|\mathbf{y}\_{K,<t})}
>     ].
> \end{aligned}
> $$
>
> Now we have a step-wise alignment over all sequence from $q$, i.e., $\mathbb{E}\_{\mathbf{y}\_{<t}\sim q}$.

---

> ### Author Response · Authors · 2024-11-19
> **Detailed Derivation of Alignment Objective of AtSpeed-S (Step 3-4)**
>
> ***Step3. Move expectation over $\mathbf{y}\_{<t}$.***
>
> We aim to align the draft model with the target LLM at every beam search step.
> Notably, at each beam search step $T$, the sequence lengths are fixed and are independent with $t$. Therefore, we can rewrite the objective into:
>
> $$
> \begin{aligned}
>     & - \sum\_t \mathbb{E}\_{\mathbf{y}\_{<t}\sim q}
>     [
>     \sum\_{y\_t} q(y\_t|c\_{<t})
>     \log \frac{q(y\_t|c\_{<t})}{p(y\_t|c\_{<t})}
>     -
>     \sum\_{y\_t} q(y\_t|c\_{<t})
>     \log \frac{q(y\_t|c\_{<t})}{p (\mathbf{y}\_{K,t}|\mathbf{y}\_{K,<t})}
>     ] \\\\
>      = &- \mathbb{E}\_{\mathbf{y}\_{\le T}\sim q} \sum\_{t=1}^{T}
>     [
>     \sum\_{y\_t} q(y\_t|c\_{<t})
>     \log \frac{q(y\_t|c\_{<t})}{p(y\_t|c\_{<t})}
>     -
>     \sum\_{y\_t} q(y\_t|c\_{<t})
>     \log \frac{q(y\_t|c\_{<t})}{p (\mathbf{y}\_{K,t}|\mathbf{y}\_{K,<t})}
>     ].
> \end{aligned}
> $$
>
> ***Step4. Normalize by length and rewrite the objective.***
>
> Since we aim to align at every step $T\in\{1,\dots,L\}$, where $L$ is the length of item identifier in LLM-based recommendation,
> we further normalize the expression by sequence length to prevent different scales on alignment loss across different steps.
> As such, for every step $T\in\{1,\dots,L\}$, the objective can be rewritten as
>
> $$
> \begin{aligned}
>     & - \mathbb{E}\_{\mathbf{y}\_{\le T}\sim q}
>     \frac{1}{|y\_T|}
>     \sum\_{t=1}^{|y\_T|}
>     [
>     \sum\_{y\_t} q(y\_t)
>     \log \frac{q(y\_t|c\_{<t})}{p(y\_t|c\_{<t})}
>     -
>     \sum\_{y\_t} q(y\_t)
>     \log \frac{q(y\_t|c\_{<t})}{p (\mathbf{y}\_{K,t}|\mathbf{y}\_{K,<t})}
>     ].
> \end{aligned}
> $$
>
> However, the expectation over the sequence space, (i.e., $\mathbb{E}\_{\mathbf{y}\_{\le T}\sim q}$) is intractable, so we follow previous work (Wen et al., 2023; Kim & Rush, 2016) to approximate it by sampling top-$K$ sequences generated by draft model $\mathcal{M}\_q$.
> Now we can rewrite our alignment objective for strict top-$K$ verification and obtain Eq.(3) in our paper:
>
> $$
> \begin{aligned}
>     & \mathop{\arg\max}\_{\theta\in\Theta}
>     - \mathbb{E}\_{(\mathbf{x},\mathcal{Y})\sim D'}
>     \sum\_{\mathbf{y}\in \mathcal{Y}}
>     \frac{1}{|\mathbf{y}|}
>     \sum\_{t=1}^{\mathbf{|y|}}
>     [
>     \sum\_{y\_t} q(y\_t|c\_{<t})
>     \log \frac{q(y\_t|c\_{<t})}{p(y\_t|c\_{<t})}
>     -
>     \sum\_{y\_t} q(y\_t|c\_{<t})
>     \log \frac{q(y\_t|c\_{<t})}{p (\mathbf{y}\_{K,t}|\mathbf{y}\_{K,<t})}
>     ] \\\\
>     =&  \mathop{\arg\min}\_{\theta\in\Theta}
>     \mathbb{E}\_{(\mathbf{x},\mathcal{Y})\sim D'}
>     \sum\_{\mathbf{y}\in \mathcal{Y}}
>     \frac{1}{|\mathbf{y}|}
>     \sum\_{t=1}^{|\mathbf{y}|}
>     [
>     \sum\_{y\_t} q(y\_t|c\_{<t})
>     \log \frac{q(y\_t|c\_{<t})}{p(y\_t|c\_{<t})}
>     -
>     \sum\_{y\_t} q(y\_t|c\_{<t})
>     \log \frac{q(y\_t|c\_{<t})}{p (\mathbf{y}\_{K,t}|\mathbf{y}\_{K,<t})}
>     ],
> \end{aligned}
> $$
>
> where $\mathcal{Y}$ denotes the top-$K$ beam sequences generated from the mixture distribution of draft model and target LLM (as stated in the "Alignment Objective" paragraph in Section 3.1, line 216-218).

---

> ### Author Response · Authors · 2024-11-19
> **Detailed Proof of With-replacement Sampling Approximation (Notation Clarification and Introduction of Multivariate Hypergeometric Distribution, Multinomial Distribution, and Stirling's Approximation)**
>
> > **Q13: The "with-replacement sampling approximation" paragraph in App A.2 is difficult to follow so it would benefit from being reworked.**
>
> **Reply**: Thanks for your valuable comments. The with-replacement sampling approximation is mainly based on Stirling’s approximation, and we provide the detailed step-by-step proof to facilitate better understanding. The detailed derivation has also been updated in the Appendix of our updated manuscript.
>
>
> We aim to prove that the distribution of sampling without replacement is approximately equivalent to that of sampling with replacement. Our proof is mainly based on Stirling's approximation.
> In the following, we will first clarify notations, and introduce multivariate hypergeometric distribution and the multinomial distribution, which are used to model the sampling without replacement and sampling with replacement, respectively.
> We then present Stirling's approximation, and show the step-by-step proof.
>
> **Notations.** To model the sampling, we have the total population size $N$, sample size $n$, the category size $r$, the number of items in category $i$ in the population $K\_i$, and the number of items in category $i$ in the samples $k\_i$.
> In the case of sequence sampling in LLM decoding, the population includes every possible sequence. Every possible sequence is a unique category, and the sample size $n$ is the beam size, the population size is the total number of all possible sequences at each beam search step.
> Now we assume that the population sizes go to infinity in such a way that
> $p\_i=\frac{K\_i}{N}$.
> And we have $\sum\_{i=1}^{r}k\_i=n$ and $\sum\_{i=1}^{r} K\_i = N$.
>
> **Multivariate hypergeometric distribution.** Formally, when sampling without replacement, the probability of drawing $k\_1, k\_2, \dots, k\_r$ items from each category is given by the multivariate hypergeometric distribution
>
> $$
> \begin{aligned}
> P\_\text{hyper}(k\_1,k\_2,\dots,k\_r) &= \frac{\prod\_{i=1}^{r}\binom{K\_i}{k\_i}}{\binom{N}{n}} \\\\
> &=\frac{\prod\_{i=1}^{r}\frac{K\_i!}{k\_i!(K\_i-k\_i)!}}{\frac{N!}{n!(N-n)!}}.
> \end{aligned}
> $$
>
> **Multinomial distribution.**
> Formally, when sampling with replacement, the probability follows the multinomial distribution as
>
> $$
> \begin{aligned}
>     P\_\text{multi}(k\_1, k\_2, \dots, k\_r)
>     = \frac{n!}{\prod\_{i=1}^{r}k\_i!}\prod\_{i=1}^{r}p\_i^{k\_i}.
> \end{aligned}
> $$
>
> **Stirling's approximation.** Stirling's approximation gives us the approximation of the logarithm of factorials as:
>
> $$
>     \ln n! \approx n\ln n -n + \frac{1}{2}\ln(2\pi n).
> $$

---

> ### Author Response · Authors · 2024-11-19
> **Detailed Proof of With-replacement Sampling Approximation (Step 1-2)**
>
> **Theorem 1.**
>      When population size $N$ is large and sample size $n$ is small compared to $N$ (i.e., $n\ll N$), the multivariate hypergeometric distribution approximates the multinomial distribution:
>
>  $$
>         P\_\text{hyper}(k\_1, k\_2, \dots, k\_r) \approx P\_\text{multi}(k\_1,k\_2, \dots, k\_r).
> $$
>
> *Proof.*
>
> ***Step1. Apply Stirling's approximation on logarithm of multivariate hypergeometric distribution.***
>
> We can expand the logarithm of hypergeometric probability in factorials as follows:
>
> $$
> \begin{aligned}
>     &\ln P\_\text{hyper}(k\_1,k\_2,\dots,k\_r) =\ln \frac{\prod\_{i=1}^{r}\frac{K\_i!}{k\_i!(K\_i-k\_i)!}}{\frac{N!}{n!(N-n)!}} \\\\
>     =& \ln \prod\_{i=1}^{r}\frac{K\_i !}{k\_i!(K\_i-k\_i)!}
>     - \ln \frac{N!}{n!(N-n)!} \\\\
>     =& \sum\_{i=1}^{r}\ln \frac{K\_i!}{k\_i!(K\_i-k\_i)!}
>     - \ln \frac{N!}{n!(N-n)!} \\\\
>     =& \sum\_{i=1}^{r} [\ln K\_i! - \ln k\_i! - \ln (K\_i-k\_i)!]
>     - [\ln N! - \ln n! - \ln (N-n)!].
> \end{aligned}
> $$
>
> Using the Stirling's approximation, we have:
>
> $$
> \left\\{
> \begin{aligned}
>     \ln K\_i ! &\approx K\_i \ln K\_i - K\_i + \frac{1}{2} \ln 2\pi K\_i ,\\\\
>     \ln k\_i ! &\approx k\_i \ln k\_i - k\_i + \frac{1}{2} \ln 2\pi k\_i , \\\\
>     \ln (K\_i-k\_i) ! &\approx (K\_i-k\_i) \ln (K\_i-k\_i) - (K\_i-k\_i) + \frac{1}{2} \ln 2\pi (K\_i-k\_i) ,\\\\
>     \ln N! &\approx N\ln N -N + \frac{1}{2}\ln2\pi N ,\\\\
>     \ln n! &\approx n\ln n -n + \frac{1}{2}\ln2\pi n ,\\\\
>     \ln (N-n)! &\approx (N-n)\ln (N-n) -(N-n) + \frac{1}{2}\ln2\pi (N-n).
> \end{aligned}
> \\right.
> $$
>
> Then, we can substitute the logarithm of factorials with the approximation as:
>
> $$
> \begin{aligned}
>      &\ln P\_\text{hyper}(k\_1,k\_2,\dots,k\_r) \\\\
>      = & \sum\_{i=1}^{r} [\ln K\_i! - \ln k\_i! - \ln (K\_i-k\_i)!]
>     - [\ln N! - \ln n! - \ln (N-n)!] \\\\
>     = &\sum\_{i=1}^{r}
>     [
>     K\_i \ln K\_i - K\_i + \frac{1}{2}\ln 2\pi K\_i
>     - (k\_i \ln k\_i - k\_i + \frac{1}{2}\ln 2\pi k\_i) \\\\
>     & \quad\quad - ((K\_i-k\_i)\ln (K\_i-k\_i) - (K\_i-k\_i) + \frac{1}{2}\ln2\pi (K\_i-k\_i) )] \\\\
>     & \quad\quad - [N\ln N - N + \frac{1}{2}\ln 2\pi N
>     - (n\ln n - n + \frac{1}{2} \ln 2\pi n ) \\\\
>     & \quad\quad - ((N-n)\ln (N-n) - (N-n) + \frac{1}{2} \ln 2\pi (N-n) )] \\\\
>     = &\sum\_{i=1}^{r}
>     [
>      K\_i \ln K\_i - k\_i \ln k\_i - (K\_i-k\_i) \ln (K\_i-k\_i)
>      + \frac{1}{2} \ln 2\pi K\_i
>       - \frac{1}{2} \ln 2\pi k\_i
>       - \frac{1}{2}\ln 2\pi (K\_i-k\_i)
>     ] \\\\
>     &  \quad\quad - [N\ln N -n\ln n - (N-n) \ln (N-n) + \frac{1}{2} \ln 2\pi N - \frac{1}{2} \ln 2\pi n - \frac{1}{2} \ln 2\pi (N-n) ].
> \end{aligned}
> $$
>
> Since $N$ is a very large number and $n \ll N$, $k\_i \ll K\_i$, we have $\frac{1}{2} \ln 2\pi K\_i - \frac{1}{2} \ln 2\pi k\_i - \frac{1}{2}\ln 2\pi (K\_i-k\_i) \approx 0$ and $\frac{1}{2} \ln 2\pi N - \frac{1}{2} \ln 2\pi n - \frac{1}{2}\ln 2\pi (N-n) \approx 0$.
> Then, the logarithm of multivariate hypergeometric distribution is approximated as:
>
> $$
>     \ln P\_\text{hyper} \approx
>     \sum\_{i=1}^{r} [K\_i \ln K\_i - k\_i \ln k\_i - (K\_i-k\_i) \ln (K\_i-k\_i)]
>     - [N\ln N -n\ln n - (N-n) \ln (N-n)].
> $$
>
> ***Step2. Approximate $\ln (K\_i-k\_i)$ and $\ln (N-n)$ using Taylor expansion.***
>
> Since $k\_i$ is small compared to $K\_i$, we can expand $\ln (K\_i-k\_i)$ using Taylor expansion
>
> $$
> \ln (K\_i-k\_i) = \ln K\_i - \frac{k\_i}{K\_i} - \frac{1}{2}(\frac{k\_i}{K\_i})^{2} + \dots,
> $$
>
> where we can neglect the high-order terms and obtain
>
> $$
>     \ln (K\_i-k\_i) \approx \ln K\_i - \frac{k\_i}{K\_i}.
> $$
>
>
> Similarly, for $\ln (N-n)$, we have $\ln (N-n) \approx \ln N - \frac{n}{N}.$
>
> We can then substitute $\ln (K\_i-k\_i)$ and $\ln (N-n)$ in logarithm of multivariate hypergeometric distribution and obtain
>
> $$
> \begin{aligned}
>     \ln P\_\text{hyper}
>     &\approx
>     \sum\_{i=1}^{r} [K\_i \ln K\_i - k\_i \ln k\_i - (K\_i-k\_i) \ln (K\_i-k\_i)]
>     - [N\ln N -n\ln n - (N-n) \ln (N-n)] \\\\
>     & = \sum\_{i=1}^{r} [K\_i\ln K\_i - k\_i \ln k\_i - (K\_i-k\_i) (\ln K\_i + \frac{k\_i}{K\_i})]
>     - [N\ln N - n\ln n - (N-n)(\ln N + \frac{n}{N})] \\\\
>     & = \sum\_{i=1}^{r}
>     [K\_i\ln K\_i - k\_i\ln k\_i - (K\_i\ln K\_i - k\_i\ln K\_i + k\_i - \frac{k\_i^2}{K\_i}) ]
>     - [N\ln N - n\ln n - (N\ln N - n\ln N + n - \frac{n^2}{N})] \\\\
>     & = \sum\_{i=1}^{r}
>     [k\_i \ln \frac{K\_i}{k\_i} + \frac{k\_i^2}{K\_i} - k\_i]
>     - [n\ln \frac{N}{n} + \frac{n^2}{N} -n] \\\\
>     & = \sum\_{i=1}^{r}
>     [k\_i \ln \frac{K\_i}{k\_i} + \frac{k\_i^{2}}{K\_i}]
>     - [n \ln \frac{N}{n}+ \frac{n^2}{N}]. \quad\quad (\text{we have} -\sum\_{i=1}^{r}k\_i+n=0 \text{ in last expression})
> \end{aligned}
> $$

---

> ### Author Response · Authors · 2024-11-19
> **Detailed Proof of With-replacement Sampling Approximation (Step 3-4)**
>
> ***Step3. Relate $K\_i$ to $N$ and $p\_i$.***
>
> We then relate $K\_i$ to $N$ and $p\_i$. Since $K\_i=Np\_i$, we have $\ln \frac{K\_i}{k\_i}=\ln \frac{Np\_i}{k\_i}$.
> We also have $n=\sum\_{i=1}^{r}k\_i$.
> Then, we can express the logarithm of hypergeometric distribution in terms of $p\_i$ and $k\_i$ as
>
> $$
> \begin{aligned}
> \ln P\_\text{hyper}
>     % & \approx \sum\_{i=1}^{r} [k\_i(\ln Np\_i - \ln k\_i) + \frac{k\_i^2}{Np\_i}]
>     % - [n(\ln N - \ln n) + \frac{n^2}{N}] \\\\
>     % & \approx \sum\_{i=1}^{r} [k\_i(\ln N + \ln p\_i - \ln k\_i) + \frac{k\_i^2}{Np\_i}]
>     % - [n(\ln N - \ln n) + \frac{n^2}{N}] \\\\
>     & \approx \sum\_{i=1}^{r} [k\_i\ln \frac{K\_i}{k\_i}+ \frac{k\_i^2}{K\_i}] - [n\ln \frac{N}{n} + \frac{n^2}{N}] \\\\
>     & = \sum\_{i=1}^{r} [k\_i\ln \frac{Np\_i}{k\_i} + \frac{k\_i^2}{Np\_i}] - [n \ln \frac{N}{n}+\frac{n^2}{N}] \\\\
>     & =\sum\_{i=1}^{r} [k\_i (\ln N + \ln p\_i - \ln k\_i)+ \frac{k\_i^2}{Np\_i}] -
>     [n\ln \frac{N}{n} + \frac{n^2}{N}] \\\\
>     & = \sum\_{i=1}^{r}
>     k\_i\ln N + \sum\_{i=1}^{r}[k\_i(\ln p\_i - \ln k\_i)+ \frac{k\_i^2}{Np\_i}] - n\ln \frac{N}{n} - \frac{n^2}{N} \\\\
>     & = n\ln N - n\ln N + n\ln n - \frac{n^2}{N} + \sum\_{i=1}^{r} [k\_i(\ln p\_i - \ln k\_i) + \frac{k\_i^2}{Np\_i}] \quad (\text{we have } \sum\_{i=1}^{r}k\_i=n \text{ in last expression})\\\\
>     & = \sum\_{i=1}^{r} [k\_i(\ln p\_i - \ln k\_i)+ \frac{k\_i^2}{Np\_i}] + n\ln n - \frac{n^2}{N} \\\\
>     &= n\ln n - \sum\_{i=1}^{r}k\_i \ln k\_i + \sum\_{i=1}^{r} k\_i \ln p\_i - \frac{n^2}{N} + \sum\_{i=1}^{r}\frac{k\_i^2}{Np\_i}.
> \end{aligned}
> $$
>
> Now the approximation of the logarithm of multivariate hypergeometric distribution has been finished.
>
> ***Step4. Compare with multinomial distribution.***
>
> Similarly, we approximate the multinomial distribution with Stirling's approximation.
> We expand the logarithm of multinomial distribution as
>
> $$
> \begin{aligned}
>     \ln P\_\text{multi}(k\_1, k\_2, \dots, k\_r)
>     & = \ln \frac{n!}{\prod\_{i=1}^{r}k\_i!}\prod\_{i=1}^{r}p\_i^{k\_i} \\\\
>     & = \ln n! - \sum\_{i=1}^{r} \ln k\_i! + \sum\_{i=1}^{r}k\_i \ln p\_i.
> \end{aligned}
> $$
>
> Using Stirling's approximation, we have $\ln n ! \approx n\ln n - n$ and $\ln k\_i! \approx k\_i \ln k\_i -k\_i$. We then substitute $\ln n!$ and $\ln k\_i!$ with the approximation and obtain
>
> $$
> \begin{aligned}
>     \ln P\_\text{multi} &\approx
>     n\ln n - n -\sum\_{i=1}^{r} (k\_i \ln k\_i - k\_i) + \sum\_{i=1}^{r} k\_i \ln p\_i \\\\
>     & = n\ln n - n - \sum\_{i=1}^{r}k\_i\ln k\_i + \sum\_{i=1}^{r}k\_i + \sum\_{i=1}^{r}k\_i \ln p\_i.
> \end{aligned}
> $$
>
>
> Since we have $\sum\_{i=1}^{r}k\_i = n$, we have
>
> $$
> \begin{aligned}
>     \ln P\_\text{multi} &\approx
>     n\ln n - n - \sum\_{i=1}^{r}k\_i\ln k\_i + n + \sum\_{i=1}^{r}k\_i \ln p\_i \\\\
>     & = n\ln n - \sum\_{i=1}^{r}k\_i \ln k\_i + \sum\_{i=1}^{r}k\_i\ln p\_i.
> \end{aligned}
> $$
>
> Now, comparing the approximated logarithm of multinomial distribution with the approximated logarithm of multivariate hypergeometric distribution, we have
>
> $$
> \begin{aligned}
>     \ln P\_\text{multi} \approx \ln P\_\text{hyper} + \frac{n^2}{N} - \sum\_{i=1}^{r}\frac{k\_i^2}{Np\_i}.
> \end{aligned}
> $$
>
> Note that the term $\frac{n^2}{N} - \sum\_{i=1}^{r}\frac{k\_i^2}{Np\_i}$ is negalectable when $N$ is large and $k\_i$ and $n$ are small compared to N. Therefore, we show that when the population size $N$ is large (e.g., all possible sequences for sampling) and $k\_i, n$ are small (e.g., $k\_i=1$ or $0$ since each sequence represents a category and $n$ is usually less than 20 in LLM-based recommendation), the multivariate hypergeometric distribution can be approximated to the multinomial distribution.
> That is, sampling without replacement is approximately equivalent to sampling with replacement.

---

> > ### Comment · Reviewer_Up12 · 2024-11-22
> > **Response to authors**
> >
> > Thank you very much for taking the time to provide such a detailed answer, I greatly appreciate the amount of work you have put into answering my concerns.
> >
> > In particular, I have carefully checked the derivation of Eq 15 that you have detailed in your response and in the updated version of the paper -- which was my primary concern. While some steps remain slightly unclear to me (specially the expectation/sum swapping in step 3 and the length normalization in step 4), it looks overall reasonable and convincing to me. Given the numerous, non-straightforward steps of this derivation, I do think that it was indeed crucial to include it in the paper.
> >
> > Some additional minor comments I have:
> > - Line 202 in the updated paper, I think $\mathcal{Y}'_p \in \mathcal{Y}_p$ should be replaced with $\mathcal{Y}'_p \subset \mathcal{Y}_p$.
> > - The answer to Q7 on $\sum_K$ did not help clarify my understanding. The explanation that "Eq. (7) minimizes TVD with the strength of K for the same sequence." from the paper is hard to understand and should be reformulated. Additionally, using the notation for sum in this context is not standard as a sum should be over an index and, unless I am mistaken, $K$ here does not seem to play the role of an index. If it is, please specify the range of this index in the sum.
> > - The notation of $q(\mathbf{y})$ for $\prod_t q(y_t | x, y_{<t})$ seems slightly confusing to me, shouldn't this be noted as $q(\mathbf{y} | \mathbf{x})$ instead?
> >
> > Nonetheless, I am overall happy with the authors' comprehensive response and I will update my score to 6.

---

> > > ### Author Response · Authors · 2024-11-22
> > >
> > > Dear Reviewer Up12,
> > >
> > > Thanks for your positive reply. Your hard work is greatly appreciated! Your valuable suggestions really help us to improve our paper.
> > >
> > > For the additional comments:
> > >
> > > **Comment 1: Line 202 in the updated paper, I think $\mathcal{Y}'_p \in \mathcal{Y}_p$ should be replaced with $\mathcal{Y}'_p \subset \mathcal{Y}_p$**
> > >
> > > > **Reply**: Thank you for your comments. We agree with you and we will revise our manuscript accordingly.
> > >
> > > **Comment 2: The answer to Q7 on $\sum_K$ did not help clarify my understanding. The explanation that "Eq. (7) minimizes TVD with the strength of K for the same sequence." from the paper is hard to understand and should be reformulated. Additionally, using the notation for sum in this context is not standard as a sum should be over an index and, unless I am mistaken, $K$ here does not seem to play the role of an index. If it is, please specify the range of this index in the sum.**
> > >
> > > > **Reply**: Thank you for pointing out the notation usage issue. Indeed, we agree with you that the sum notation should be over an index. To facilitate standard notation usage and better understanding, we will revise the sum notation into a multiplier $K$, that represents "the strength of $K$ for the same sequence".
> > >
> > > **Comment 3: The notation of $q(\mathbf{y})$ for $\prod_t q(y_t | x, y_{<t})$ seems slightly confusing to me, shouldn't this be noted as $q(\mathbf{y} | \mathbf{x})$ instead?**
> > >
> > > > **Reply**: Thanks for your valuable comments. Your understanding is correct. The notation of $q(\mathbf{y})$ should be $q(\mathbf{y} | \mathbf{x})$. We omitted the condition $x$ for $q(\mathbf{y}|\mathbf{x})$ in our previous version, which could lead to confusion. We will keep the condition in $q(\mathbf{y}|\mathbf{x})$ and  $\prod_t q(y_t | x, y_{<t})$ consistent and revise our manuscript accordingly.
> > >
> > > Thanks again for taking the time to give us such a detailed and insightful review.

---

> > > > ### Comment · Reviewer_Up12 · 2024-11-22
> > > > **Response to authors**
> > > >
> > > > Thank you for your fast reply and additional clarifications! I again appreciate your engagement in this discussion.

---

### Official Review · Reviewer_WeyY · 2024-11-02

**Soundness:** 2
**Presentation:** 2
**Contribution:** 2
**Rating:** 6
**Confidence:** 4

**Summary:**

This paper proposes AtSpeed, a framework to accelerate LLM-based generative recommendation systems through speculative decoding. The main challenge addressed is the inherent inefficiency of autoregressive beam search in generating top-K recommendations. The authors identify that traditional SD methods, which focus on N-to-1 verification, are insufficient for recommendation tasks that require N-to-K verification to output a ranked list of K items. To tackle this, the paper introduces two methods: AtSpeed-S for strict top-K alignment and AtSpeed-R for relaxed sampling verification, which aims to reduce the number of LLM calls while preserving accuracy. Experimental results demonstrate that AtSpeed achieves significant speedups (up to 2.5×) with minimal degradation in recommendation accuracy.

**Strengths:**

- The paper extends the N-to-1 verification to the N-to-K verification in the context of generative recommendation, which fits the setting of real-world recommender systems.
- The authors provide a solid theoretical foundation for AtSpeed to align the draft and target models. The optimization objectives are clearly motivated and mathematically sound.
- Empirical results are compelling, showing that AtSpeed can achieve up to a 2.5× speedup while maintaining competitive recommendation accuracy. The results are well presented and highlight the practical utility of the proposed method.

**Weaknesses:**

- The paper may overclaim their contribution to be the first to propose the speculative decoding task for LLM-based recommender acceleration. There already exists prior work on speculative decoding for LLM-based recommendation: *A Decoding Acceleration Framework for Industrial Deployable LLM-based Recommender Systems (Xi, Yunjia, et al)*, which needs to be compared and discussed.
- The experiments are somewhat limited to two datasets (Amazon Beauty and Games). While these datasets are commonly used, the paper would benefit from broader validation across additional domains or larger-scale datasets.

**Questions:**

- [Q1] The paper primarily compares AtSpeed with KD-based baselines. Existing SD-based baselines should also be compared including the paper I mentioned before (A Decoding Acceleration Framework for Industrial Deployable LLM-based Recommender Systems).
- [Q2] Have the authors tested their framework on any larger and more diverse datasets, such as MovieLens or more complex datasets like Goodreads? If not, can the authors comment on the expected performance and generalization ability of AtSpeed in these contexts?
- [Q3] The paper mentions the use of LLaMA-7B, but it would be interesting to know how AtSpeed scales with even larger models (e.g., LLaMA-13B or GPT-3). Do the authors anticipate any bottlenecks or limitations when scaling to larger models, especially concerning memory usage and GPU efficiency?

---

> ### Author Response · Authors · 2024-11-22
> **Reply to Weakness 1**
>
> Dear Reviewer WeyY,
>
> Thanks for your valuable comments. We greatly appreciate your effort to review. We have provided detailed explanations and additional experimental results to address your concerns. We have also included the additional results in Appendix of the updated manuscript accordingly (marked in orange). Please feel free to leave further comments or questions if there’s any misunderstanding. And we will reply quickly.
>
> > **Weakness 1. The paper may overclaim their contribution to be the first to propose the speculative decoding task for LLM-based recommender acceleration. There already exists prior work on speculative decoding for LLM-based recommendation: A Decoding Acceleration Framework for Industrial Deployable LLM-based Recommender Systems (Xi, Yunjia, et al), which needs to be compared and discussed.**
>
> **Reply**: Thanks for your valuable comments. We also recognized this work and discussed it in our paper. It is a great work, which focuses on the speculative decoding for feature generation. Although this paper also adopts speculative decoding, it aims to accelerate the feature generation (**LLM as feature generator**) for the downstream non-LLM recommender, while we aim to accelerate the item generation (**LLM as item recommender**). Specifically,
>
> - This related work (named DARE) aims to apply SD to the inference of LLMs for user/item feature generation. The generated user/item feature is then utilized in the downstream conventional recommender models for Click-Through-Rate (CTR) tasks. However, the user/item feature generation process only requires a single sequence as output, which simply follows the traditional N-to-1 verification as the same as traditional SD to verify each drafted token at each step.
>
>
> - In contrast, our work aims to apply SD to the inference of LLMs for top$K$ item recommendation, which emphasizes addressing the challenge of difficult $N$-to-$K$ verification at each step to accept $K$ sequences out of $N$ drafted sequences instead of accepting only a token.
>
> Considering the significant difference in the tasks between DARE and our work, we classify DARE as **SD for LLM-enhanced recommendation (LLM as feature generator)** and ours as **SD for LLM-based recommendation (LLM as item recommender)**. The distinction between LLMs for feature generation and LLMs for item generation has also been widely recognized in current literature [1][2]. Therefore, we claim that we are the first work to apply SD for LLM-based recommendation and discuss DARE in the related work of our manuscript.
>
>
> [1] Wu Likang, et al. A Survey on Large Language Models for Recommendation. In World Wide Web 2024.
>
> [2] Lin Jianghao, et al. How Can Recommender Systems Benefit from Large Language Models: A Survey. In TOIS 2024.

---

> ### Author Response · Authors · 2024-11-22
> **Reply to Weakness 2 (Additional Results on MovieLens-1M and Goodreads)**
>
> > **Weakness 2. The experiments are somewhat limited to two datasets (Amazon Beauty and Games). While these datasets are commonly used, the paper would benefit from broader validation across additional domains or larger-scale datasets.**
>
> **Reply**: Thanks for your valuable comments. Following your suggestions in Question 2, we added new experiments on MovieLens-1M dataset and the Goodreads dataset. The results are as follows.
>
>
> Table1. Performance comparison of AtSpeed and baselines on MovieLens-1M dataset under strict topK verification and relaxed sampling verification.
>
> |   MovieLens-1M   |               |          |          |          |          |          |          |          |          |
> |:----------------:|:-------------:|:--------:|:--------:|:--------:|:--------:|:--------:|:--------:|:--------:|:--------:|
> | Verification     | Method        |   WS@3   |   WS@5   |   WS@10  |   WS@20  |   AS@3   |   AS@5   |   AS@10  |   AS@20  |
> | Strict TopK      | DARE          |   1.26   |   1.28   |   1.39   |   1.72   |   1.00   |   1.00   |   1.00   |   1.00   |
> |                  | SFT           |   1.65   |   1.35   |   1.48   |   1.76   |   1.99   |   1.26   |   1.14   |   1.03   |
> |                  | WordKD        |   1.29   |   1.29   |   1.39   |   1.69   |   1.07   |   1.02   |   1.00   |   1.00   |
> |                  | TVDKD         |   1.73   |   1.72   |   1.25   |   1.24   |   1.98   |   1.90   |   1.04   |   1.00   |
> |                  | SeqKD         |   1.77   |   1.78   |   1.42   |   1.50   |   2.03   |   2.00   |   1.24   |   1.05   |
> |                  | **AtSpeed-S** | **1.86** | **1.79** | **1.80** | **1.75** | **2.08** | **2.03** | **1.98** | **1.09** |
> |                  | AtSpeed-R     |   1.76   |   1.78   |   1.54   |   1.74   |   2.01   |   1.99   |   1.29   |   1.08   |
> | Relaxed Sampling | DARE          |   2.01   |   1.84   |   1.35   |   1.44   |   2.16   |   2.02   |   1.00   |   0.35   |
> |                  | SFT           |   2.08   |   2.03   |   2.02   |   1.61   |   2.28   |   2.20   |   2.06   |   0.93   |
> |                  | WordKD        |   1.97   |   1.87   |   1.48   |   1.28   |   2.15   |   2.08   |   1.23   |   0.00   |
> |                  | TVDKD         |   2.00   |   1.98   |   1.68   |   1.29   |   2.30   |   2.21   |   1.59   |   0.00   |
> |                  | SeqKD         |   2.13   |   2.08   |   1.93   |   1.56   |   2.19   |   2.14   |   2.01   |   0.78   |
> |                  | AtSpeed-S     |   2.23   |   2.16   |   2.11   | **1.65** |   2.41   |   2.33   | **2.16** | **1.02** |
> |                  | **AtSpeed-R** | **2.24** | **2.22** | **2.14** |   1.64   | **2.44** | **2.38** |   2.15   |   0.95   |
>
>
> Table 2. Performance comparison of AtSpeed and baselines on Goodreads dataset under strict topK verification and relaxed sampling verification.
>
> |     Goodreads    |               |          |          |          |          |          |          |          |          |
> |:----------------:|:-------------:|:--------:|:--------:|:--------:|:--------:|:--------:|:--------:|:--------:|:--------:|
> | Verification     | Method        |   WS@3   |   WS@5   |   WS@10  |   WS@20  |   AS@3   |   AS@5   |   AS@10  |   AS@20  |
> | Strict TopK      | DARE          |   1.30   |   1.32   |   1.44   |   1.75   |   1.00   |   1.00   |   1.00   |   1.00   |
> |                  | SFT           |   1.83   |   1.81   |   2.17   |   2.46   |   2.04   |   1.98   |   1.72   |   1.07   |
> |                  | WordKD        |   1.83   |   1.92   |   2.07   |   2.38   |   2.00   |   1.96   |   1.58   |   1.00   |
> |                  | TVDKD         |   1.89   |   1.93   |   2.17   |   2.46   |   2.07   |   1.97   |   1.70   |   1.07   |
> |                  | SeqKD         |   1.82   |   1.89   |   2.19   |   2.48   |   2.00   |   1.96   |   1.73   |   1.08   |
> |                  | **AtSpeed-S** | **2.25** | **2.26** | **2.20** |   2.48   | **2.32** | **2.18** | **1.81** |   1.08   |
> |                  | AtSpeed-R     |   2.11   |   2.07   |   2.20   | **2.49** |   2.24   |   2.09   |   1.80   | **1.12** |
> | Relaxed Sampling | DARE          |   1.84   |   1.83   |   1.35   |   1.43   |   2.06   |   2.02   |   1.00   |   0.35   |
> |                  | SFT           |   2.15   |   2.09   |   1.70   |   1.91   |   2.27   |   2.08   |   1.01   |   0.10   |
> |                  | WordKD        |   2.01   |   2.04   |   1.68   |   1.92   |   2.15   |   2.05   |   1.00   |   0.15   |
> |                  | TVDKD         |   2.27   |   2.22   |   1.71   |   2.02   |   2.36   |   2.18   |   1.03   |   0.25   |
> |                  | SeqKD         |   1.90   |   1.96   |   1.66   |   1.85   |   2.08   |   2.01   |   1.00   |   0.02   |
> |                  | AtSpeed-S     |   2.18   |   2.13   |   1.71   |   1.93   |   2.28   |   2.12   |   1.02   |   0.17   |
> |                  | **AtSpeed-R** | **2.45** | **2.39** | **1.77** | **2.36** | **2.50** | **2.32** | **1.10** | **0.87** |

---

> > ### Author Response · Authors · 2024-11-22
> > **Observations of Additional Results on MovieLens-1M and Goodreads**
> >
> > From the results, we can observe that
> >
> > - 1) AtSpeed-S and AtSpeed-R outperforms baselines in most cases under strict top$K$ verification and relaxed sampling verification, respectively. This validate the effectiveness and generalization ability of our proposed method on diverse datasets and is consistent with the observations on Amazon Beauty and Games (Table 1 in our manuscript).
> > - 2) The relaxed sampling verification generally shows superior speedup compared to strict top$K$ verification when $K=3,5$, while yields inferior speedup when $K$ is large (e.g., $K=20$ on MovieLens-1M and Goodreads). One possible reason is that the item size is relatively small on the two datasets ($3,017$ movies and $4,667$ books) compared to Beauty ($12,035$ products) and Games ($17,332$ products), which might results in long-tailed draft distribution, where top$K$ valid sequences have overwhelmingly high probabilities (i.e., $q\ge p$), thus leading to a high rejection probabilities.
> > We have also included the results on the two additional datasets in the Appendix of our updated manuscripts on page 23 (marked in orange).

---

> ### Author Response · Authors · 2024-11-22
> **Reply to Question 1 (Additional Performance Comparison with SD-based Baseline DARE on Four Datasets)**
>
> > **Q1: The paper primarily compares AtSpeed with KD-based baselines. Existing SD-based baselines should also be compared including the paper I mentioned before (A Decoding Acceleration Framework for Industrial Deployable LLM-based Recommender Systems).**
>
> **Reply**: Thanks for your valuable comments.
> Following your suggestions, we extend the related work you mentioned to our setting and report the performance comparison as follows. We have also updated the additional results in our latest manuscript (Table 1 on page 8 and Table 6 on page 23).
>
> From the results, we can observe that 1) our proposed method consistently outperforms extended DARE. This is reasonable since the candidate items are uniformly sampled from the valid items, which might not be well-aligned with the top$K$ sequence distribution from the target LLM, thus leading to a low acceptance rate and less satisfying speedup. Notably, 2) DARE has constant zero accept step on Games, which is mainly due to the large valid item size during retrieval. Uniform sampling from a large population is less likely to get accepted by the target LLM. In contrast, DARE achieves constant one accept step on MovieLens-1M and Goodreads. The possible reason is that these two datasets have relatively small item size, thus the number of first valid tokens might be smaller than $K$. As such, all retrieved valid items will be verified to be accepted for the first step.
>
>
> | Beauty           |               |          |          |          |          |          |          |          |          |
> |------------------|:-------------:|:--------:|:--------:|:--------:|:--------:|:--------:|:--------:|:--------:|:--------:|
> | Verification     | Method        |   WS@3   |   WS@5   |   WS@10  |   WS@20  |   AS@3   |   AS@5   |   AS@10  |   AS@20  |
> | Strict topK      | DARE          |   1.07   |   1.06   |   1.15   |   1.48   |   0.44   |   0.26   |   0.05   |   0.00   |
> |                  | **AtSpeed-S** | **1.97** | **1.84** | **1.87** | **1.84** | **2.20** | **2.00** | **1.64** | **0.57** |
> | Relaxed Sampling | DARE          |   1.65   |   1.70   |   1.53   |   1.95   |   2.00   |   1.97   |   1.14   |     1    |
> |                  | **AtSpeed-R** | **1.94** | **1.94** | **2.16** | **2.47** | **2.19** | **2.13** | **2.01** | **1.77** |
>
>
>
> | Games            |               |          |          |          |          |          |          |          |          |
> |------------------|:-------------:|:--------:|:--------:|:--------:|:--------:|:--------:|:--------:|:--------:|:--------:|
> | Verification     | Method        |   WS@3   |   WS@5   |   WS@10  |   WS@20  |   AS@3   |   AS@5   |   AS@10  |   AS@20  |
> | Strict topK      | DARE          |   0.95   |   0.99   |   1.13   |   1.44   |   0.00   |   0.00   |   0.00   |   0.00   |
> |                  | **AtSpeed-S** | **1.77** | **1.78** | **1.85** | **1.76** | **2.02** | **1.96** | **1.69** | **0.68** |
> | Relaxed Sampling | DARE          |   1.64   |   1.68   |   1.19   |   1.42   |   2.00   |   1.96   |   0.37   |     0    |
> |                  | **AtSpeed-R** | **1.92** | **2.00** | **2.05** | **2.20** | **2.18** | **2.17** | **1.98** | **1.35** |
>
>
> | MovieLens-1M     |               |          |          |          |          |          |          |          |          |
> |------------------|:-------------:|:--------:|:--------:|:--------:|:--------:|:--------:|:--------:|:--------:|:--------:|
> | Verification     | Method        |   WS@3   |   WS@5   |   WS@10  |   WS@20  |   AS@3   |   AS@5   |   AS@10  |   AS@20  |
> | Strict topK      | DARE          |   1.26   |   1.28   |   1.39   |   1.72   |   1.00   |   1.00   |   1.00   |   1.00   |
> |                  | **AtSpeed-S** | **1.86** | **1.79** | **1.80** | **1.75** | **2.08** | **2.03** | **1.98** | **1.09** |
> | Relaxed Sampling | DARE          |   2.01   |   1.84   |   1.35   |   1.44   |   2.16   |   2.02   |   1.00   |   0.35   |
> |                  | **AtSpeed-R** | **2.24** | **2.22** | **2.14** | **1.64** | **2.44** | **2.38** | **2.15** | **0.95** |
>
>
> | Goodreads        |               |          |          |          |          |          |          |          |          |
> |------------------|:-------------:|:--------:|:--------:|:--------:|:--------:|:--------:|:--------:|:--------:|:--------:|
> | Verification     | Method        |   WS@3   |   WS@5   |   WS@10  |   WS@20  |   AS@3   |   AS@5   |   AS@10  |   AS@20  |
> | Strict topK      | DARE          |   1.30   |   1.32   |   1.44   |   1.75   |   1.00   |   1.00   |   1.00   |   1.00   |
> |                  | **AtSpeed-S** | **2.25** | **2.26** | **2.20** | **2.48** | **2.32** | **2.18** | **1.81** | **1.08** |
> | Relaxed Sampling | DARE          |   1.84   |   1.83   |   1.35   |   1.43   |   2.06   |   2.02   |   1.00   |   0.35   |
> |                  | **AtSpeed-R** | **2.45** | **2.39** | **1.77** | **2.36** | **2.50** | **2.32** | **1.10** | **0.87** |

---

> > ### Author Response · Authors · 2024-11-22
> > **Reply to Question 1 (Discussion on Other Existing SD-based Methods)**
> >
> > For the other existing SD-based methods:
> >
> > **From the perspective of verification strategy**, they are typically designed for SD with $N$-to-$1$ verification, including the mentioned related work DARE. However, the top-$K$ item generation requires an $N$-to-$K$ sequence verification. Therefore, from the perspective of verification strategy, prior SD-based methods cannot be directly adopted for performance comparison.
> >
> > **On the other hand, from the perspective of drafting strategy**, current $N$-to-$1$ SD-based methods can be broadly categorized into self-drafting, external language model drafting, and external retrieval-based drafting [1].  While the self-drafting and external retrieval-based drafting approaches fail to be directly adopted in SD for LLM-based recommendation, we mainly compared our method with the baselines from external language model drafting. Specifically,
> >
> > 1. **Self-drafting methods** typically leverage target LLM to efficiently generate multiple tokens at each future step (e.g., via multi-head prediction [2][3]). However, it is non-trivial to adopt the self-drafted multiple tokens for top-$K$ item generation via beam search. In particular, the SD under beam search requires each candidate a token sequence for every future step rather than a specific token.
> >
> > 	For example, we have $\gamma=3$ and $N=5$ for each drafted future step, the self-drafting approach gives N drafted tokens as:
> > 	> step 1: ``a1``, ``a2``, ``a3``, ``a4``, ``a5``
> > 	>
> > 	> step 2: ``b1``, ``b2``, ``b3``, ``b4``, ``b5``
> >         >
> >         > step 3: ``c1``, ``c2``, ``c3``, ``c4``, ``c5``
> >
> > 	Since we need sequence for each step under the $N$-to-$K$ verification, we need to construct sequences at each step. An intuitive way is to obtain all possible combinations, e.g., 25 possible sequences ``a1b1``, ``a1b2``, …, ``a1b5``, ``a2b1``, …, ``a2b5``, …, ``a5b1``, …, ``a5b5`` at step 2. Similarly, we have $5^3$ sequences at step 3. However, the $N$ is usually set to a large number, e.g., 40, which makes it infeasible to verify all these possible sequences.
> > Therefore, it requires extensive additional work to design an effective combination strategy to combine the tokens at different steps into token sequences that align well with the target LLM.
> >
> > 2. **External language model drafting** aligns with our work, which utilizes a small-sized language model as a draft model and mainly leverages KD to achieve better alignment. Therefore, we compare the representative KD-based methods for performance comparison.
> >
> > 3. **External retrieval-based drafting** retrieves tokens from the external corpus. The related work “A Decoding Acceleration Framework for Industrial Deployable LLM-based Recommender Systems” also lies in this line of work, which retrieves tokens from existing user features. Nevertheless, DARE focuses on user/item feature generation, where the proposed retrieval method is based on previously generated user/item features, and thereby cannot be directly used in our setting. To compare with DARE, we borrow the concept and devise a retrieval-based drafting method, which retrieves all valid sequences from item identifiers.
> >
> > [1] Heming Xia, et al., Unlocking Efficiency in Large Language Model Inference: A Comprehensive Survey of Speculative Decoding. In ACL 2024.
> >
> > [2] Tianle Cai, et al., Medusa: Simple LLM Inference Acceleration Framework with Multiple Decoding Heads. In ICML 2024.
> >
> > [3] Yuhui Li, et al., EAGLE: Speculative Sampling Requires Rethinking Feature Uncertainty. In ICML 2024.

---

> ### Author Response · Authors · 2024-11-22
> **Reply to Question 2-3**
>
> > **Q2: Have the authors tested their framework on any larger and more diverse datasets, such as MovieLens or more complex datasets like Goodreads? If not, can the authors comment on the expected performance and generalization ability of AtSpeed in these contexts?**
>
>
> **Reply**: Thanks for your valuable questions. Following your suggestions, we run additional experiments on the MovieLens-1M dataset and Goodreads dataset to validate the generalization ability of our proposed methods. Please refer to the ``Reply to Weakness 2`` for detailed results. We have also included the additional results into the Appendix of our updated manuscripts (marked in orange on page 23).
>
>
> > **Q3: The paper mentions the use of LLaMA-7B, but it would be interesting to know how AtSpeed scales with even larger models (e.g., LLaMA-13B or GPT-3). Do the authors anticipate any bottlenecks or limitations when scaling to larger models, especially concerning memory usage and GPU efficiency?**
>
>
> **Reply**: Thanks for your insightful questions.
>
> Currently, research studies on LLM-based recommender typically run experiments on 7B models, such as LC-Rec [1],  PALR [2], TALLRec [3], CoLLM [4], LLaRA [5], and BIGRec [6]. As such, we follow the current literature to validate the effectiveness of our method in inference acceleration on the 7B LLM recommender. The bottleneck of using larger LLMs (e.g., LLaMA-13B or GPT-3) mainly lies in the large resource costs of fine-tuning LLMs on recommendation data. If we continue scaling up the model size, the computational costs (e.g., memory, GPU) and time costs for fine-tuning LLMs on recommendation data will be significantly high.
>
> Nevertheless, if there are sufficient resources for fine-tuning larger LLM recommender (e.g., LLaMA-13B), **our method is expected to be practically feasible and effective to accelerate the larger LLM inference for recommendation**. This is because our work aims to train a draft model to generate drafts that align well with the target model’s output. Therefore, given a fine-tuned target LLM and the training data, the draft model’s training cost remains constant, thus facilitating accelerations regardless of the target LLM’s size.
>
> [1] Bowen Zheng et al., Adapting Large Language Models by Integrating Collaborative Semantics for Recommendation. In ICDE 2024.
>
> [2] Fan Yang et al., PALR: Personalization Aware LLMs for
> Recommendation. Arxiv 2023.
>
> [3] Keqin Bao et al., TALLRec: An Effective and Efficient Tuning Framework to Align Large Language Model with Recommendation. In RecSys 2023.
>
> [4] Yang Zhang et al., CoLLM: Integrating Collaborative Embeddings into Large Language Models for Recommendation. In TKDE.
>
> [5] Jiayi Liao et al., LLaRA: Large Language-Recommendation Assistant. In SIGIR 2024.
>
> [6] Keqin Bao et al., A Bi-Step Grounding Paradigm for Large Language Models in Recommendation Systems. Arxiv 2023.

---

> ### Author Response · Authors · 2024-11-24
>
> Dear Reviewer WeyY,
>
> We would like to kindly follow up to see if our response addresses your concerns. We are happy to take any further questions and we eagerly anticipate our discussion with you! Please feel free to let us know if there's any misunderstanding. Thanks for your time and review.

---

> > ### Comment · Reviewer_WeyY · 2024-11-24
> > **Thank you for your response**
> >
> > Thanks for addressing most of my concerns, and I decided to raise my score.

---

> > > ### Author Response · Authors · 2024-11-24
> > >
> > > Thanks for your positive feedback and your hard work on the review. We sincerely appreciate it! Your constructive and insightful feedback has been very helpful in improving our paper.

---

### Official Review · Reviewer_y4np · 2024-11-04

**Soundness:** 3
**Presentation:** 4
**Contribution:** 3
**Rating:** 8
**Confidence:** 4

**Summary:**

This paper focuses on accelerating inference in LLM-based generative recommendation using speculative decoding. It highlights the challenges of applying speculative decoding directly to the generative recommendation, due to the N-to-K issue. The authors introduce two improvements in both the drafting and verification stages. Experiments on two public datasets demonstrate the efficiency of the proposed method.

**Strengths:**

1. A timely study addressing inference efficiency in generative recommendation.
2. The ideas of (1) fine-tuning the draft model to align with top-K items of the target model and (2) adding a probability to accept rejected items in a more flexible verification setting are both novel and interesting.
3. The paper is well-written and easy to follow.
4. Experiments on two public datasets demonstrate the efficiency of the proposed method.
5. Code is available during the review phase, enhancing reproducibility.

**Weaknesses:**

1. Performance metrics (e.g., NDCG and Recall) are not well-presented. Only Table 2 includes some ranking metrics (also only Recall, without NDCG), which may raise doubts about the proposed method's ranking performance. Including these metrics in Table 1 and Figure 3 would strengthen the results. The limited metrics reported make it challenging to fully assess AtSpeed's ranking performance.
2. Presentation issues. Reporting WS@K and AS@K for all values of K in {1, 3, 5, 10, 20} in Table 1 seems unnecessary, especially since the discussion focuses on average WS and AS. I suggest presenting only representative Ks alongside the averages, freeing space for additional ranking metrics.

**Questions:**

Please refer to "Weaknesses" for more details.

Lastly, I want to share a related paper, "Inductive Generative Recommendation via Retrieval-based Speculation". This paper was released after the ICLR submission deadline and is currently available only as a preprint. While this is not a critique or question, it’s relevant as it discusses speculative decoding in generative recommendation. It proposes a dynamic N for the draft model and introduces a technique to perform limited beam search, using prefixes generated by the first few steps of the target model to guide the draft model. I believe this work shares some conceptual similarities with this paper, so I thought it worth mentioning.

---

> ### Author Response · Authors · 2024-11-22
> **Reply to Weakness 1 (Additional Experiment Results of Ranking Performance on Beauty and Games Datasets)**
>
> Dear Reviewer y4np,
>
>
> Thanks for your positive comments, we greatly appreciate your effort to review. We have provided additional experiments for support. If there’s any misunderstanding, please feel free to let us know and we will reply quickly.
>
>
> > **Weakness 1. Performance metrics (e.g., NDCG and Recall) are not well-presented. Only Table 2 includes some ranking metrics (also only Recall, without NDCG), which may raise doubts about the proposed method's ranking performance. Including these metrics in Table 1 and Figure 3 would strengthen the results. The limited metrics reported make it challenging to fully assess AtSpeed's ranking performance.**
>
> **Reply**: Thanks for your valuable comments. Following your suggestions, we added the recommendation performance of all methods in terms of Recall and NDCG, in comparison to target LLM with top$K$ beam search and sampling-based beam search. The results on Beauty and Games are as follows. We will also add the results to our manuscript.
>
> Table 1. Performance comparison under **strict top$K$ verification on Beauty**.
>
> | Beauty |  |  |  |  |  |  |  |
> |---|---|:---:|:---:|:---:|:---:|:---:|:---:|
> |  |  | Recall@5 | Recall@10 | NDCG@5 | NDCG@10 | WS@5 | WS@10 |
> | Without SD | Target LLM (topK) | 0.0056 | 0.0098 | 0.0051 | 0.0066 | 1 | 1 |
> | Strict TopK Verification | SFT | 0.0056 | 0.0098 | 0.0051 | 0.0066 | 1.43 | 1.37 |
> |  | WordKD | 0.0056 | 0.0098 | 0.0051 | 0.0066 | 1.58 | 1.52 |
> |  | TVDKD | 0.0056 | 0.0098 | 0.0051 | 0.0066 | 1.44 | 1.37 |
> |  | SeqKD | 0.0056 | 0.0098 | 0.0051 | 0.0066 | 1.75 | 1.67 |
> |  | **AtSpeed-S** | **0.0056** | **0.0098** | **0.0051** | **0.0066** | **1.84** | **1.87** |
> |  | AtSpeed-R | 0.0056 | 0.0098 | 0.0051 | 0.0066 | 1.70 | 1.71 |
>
> Table 2. Performance comparison under **relaxed sampling verification on Beauty**.
>
> | Beauty |  |  |  |  |  |  |  |
> |---|---|:---:|:---:|:---:|:---:|:---:|:---:|
> |  |  | Recall@5 | Recall@10 | NDCG@5 | NDCG@10 | WS@5 | WS@10 |
> | Without SD | Target LLM (topK) | 0.0056 | 0.0098 | 0.0051 | 0.0066 | 1 | 1 |
> |  | Target LLM (sampling) | 0.0056 | 0.0082 | 0.0043 | 0.0066 | 1 | 1 |
> | Relaxed Sampling Verification | SFT | 0.0057 | 0.0091 | 0.0041 | 0.0063 | 1.80 | 2.06 |
> |  | WordKD | 0.0066 | 0.0105 | 0.0043 | 0.0058 | 1.81 | 1.99 |
> |  | TVDKD | 0.0057 | 0.0083 | 0.0045 | 0.0054 | 1.81 | 2.06 |
> |  | SeqKD | 0.0055 | 0.0116 | 0.0045 | 0.0067 | 1.90 | 2.11 |
> |  | **AtSpeed-S** | **0.0060** | **0.0096** | **0.0046** | **0.0060** | **1.89** | **2.12** |
> |  | **AtSpeed-R** | **0.0058** | **0.0092** | **0.0049** | **0.0063** | **1.94** | **2.16** |
> |  | **_Average_** | **_0.0059_** | **_0.0097_** | **_0.0045_** | **_0.0061_** | / | / |
>
>
> Table 3. Performance comparison under **strict top$K$ verification on Games**.
>
> | Games |  |  |  |  |  |  |  |
> |---|---|:---:|:---:|:---:|:---:|:---:|:---:|
> |  |  | Recall@5 | Recall@10 | NDCG@5 | NDCG@10 | WS@5 | WS@10 |
> | Without SD | Target LLM (topK) | 0.0074 | 0.0125 | 0.0065 | 0.0083 | 1 | 1 |
> | Strict TopK Verification | SFT | 0.0074 | 0.0125 | 0.0065 | 0.0083 | 1.43 | 1.40 |
> |  | WordKD | 0.0074 | 0.0125 | 0.0065 | 0.0083 | 1.31 | 1.35 |
> |  | TVDKD | 0.0074 | 0.0125 | 0.0065 | 0.0083 | 1.24 | 1.32 |
> |  | SeqKD | 0.0074 | 0.0125 | 0.0065 | 0.0083 | 1.60 | 1.46 |
> |  | **AtSpeed-S** | **0.0074** | **0.0125** | **0.0065** | **0.0083** | **1.78** | **1.85** |
> |  | AtSpeed-R | 0.0074 | 0.0125 | 0.0065 | 0.0083 | 1.76 | 1.76 |
>
> Table 4. Performance comparison under **relaxed sampling verification on Games**.
>
> | Games |  |  |  |  |  |  |  |
> |---|---|:---:|:---:|:---:|:---:|:---:|:---:|
> |  |  | Recall@5 | Recall@10 | NDCG@5 | NDCG@10 | WS@5 | WS@10 |
> | Without SD | Target LLM (topK) | 0.0074 | 0.0125 | 0.0065 | 0.0083 | 1 | 1 |
> |  | Target LLM (sampling) | 0.0075 | 0.0115 | 0.0066 | 0.0079 | 1 | 1 |
> | Relaxed Sampling Verification | SFT | 0.0073 | 0.0112 | 0.0060 | 0.0074 | 1.84 | 1.97 |
> |  | WordKD | 0.0072 | 0.0113 | 0.0058 | 0.0073 | 1.78 | 1.84 |
> |  | TVDKD | 0.0069 | 0.0108 | 0.0061 | 0.0074 | 1.81 | 1.90 |
> |  | SeqKD | 0.0071 | 0.0110 | 0.0059 | 0.0073 | 1.90 | 2.03 |
> |  | **AtSpeed-S** | **0.0080** | **0.0131** | **0.0068** | **0.0085** | **1.91** | **2.04** |
> |  | **AtSpeed-R** | **0.0076** | **0.0123** | **0.0063** | **0.0080** | **2.00** | **2.05** |
> |  | **_Average_** | **_0.0073_** | **_0.0116_** | **_0.0062_** | **_0.0077_** | / | / |

---

> > ### Author Response · Authors · 2024-11-22
> > **Reply to Weakness 1 (Observations of the Additional Results of Ranking Performance)**
> >
> > From the tables, we have the following observations:
> >
> > - **The ranking performance under strict top$K$ verification is lossless (Table 1 and 3).** This is expected since strict verification only accepts the drafts that perfectly match the top$K$ sequence from the target LLM. Therefore, we obtain identical generation results with and without speculative decoding under strict verification. Based on lossless results, our proposed method AtSpeed-S achieves up to an average of 1.85X speedup.
> >
> > - **The ranking performance under relaxed sampling verification across different alignment methods only shows limited performance drops compared to the target LLM’s top$K$ results (comparable performance on AtSpeed-S, AtSpeed-R and “Average” line in Table 2 and 4)**, which is consistent with the results in Table 2 of our manuscript. This also meets our expectations since the sampling-based verification ensures the approximately equivalent distribution between the SD output and target LLM output under sampling-based generation. We calculate the average over all methods for comparison because we care about how relaxed sampling verification affects the recommendation accuracy. In other words, baseline draft models are also expected to show limited ranking performance drop even if they are less aligned with the target LLM and have a relatively low speedup (e.g., SFT in Table 2).
> >
> > - **Compared to NDCG, the Recall under relaxed sampling verification usually achieves comparable or even better values compared to that of the target LLM**. This is reasonable since this work aims to align the top$K$ sequence distribution between the draft model and the target LLM. We emphasize the top$K$ drafted sequence to be accepted with a higher acceptance rate (i.e., a high recall of top$K$ sequences), which does not explicitly require the draft model to distinguish the ranking between top$K$ sequences (potentially lead to relatively limited performance in terms of NDCG). Nonetheless, it is worth pursuing the non-trivial explicit probability ordering during alignment, which we consider leaving for further exploration in future work.

---

> > ### Comment · Reviewer_y4np · 2024-11-23
> >
> > Thank you to the author(s) for taking the time to address my concerns. The additional results and discussion have made the paper more self-contained, and I will raise my score.

---

> > > ### Author Response · Authors · 2024-11-24
> > >
> > > Dear Reviewer y4np,
> > >
> > > Thank you for your positive feedback. We greatly appreciate the time and effort you dedicated to the review! Your valuable and insightful comments are really helpful in guiding us to improve our work.

---

> ### Author Response · Authors · 2024-11-22
> **Reply to Weakness 1 (Additional Ranking Performance for Ablation Study)**
>
> In addition, following your suggestion, we run the ranking performance for the ablation study. The results are as follows. From the results, we can find that
>
> - Under strict top$K$ verification, our proposed method and ablation variants have identical ranking performance compared to the target LLM, which is expected.
>
> - Besides, under relaxed sampling verification, our methods and ablation variants achieve limited performance drop to the target LLM. This is also consistent with the observations of the ranking performance across different baselines, which also meets our expectation. We also theoretically show that the output distribution under relaxed sampling is approximately equivalent to the original output distribution of the target LLM. The empirical results further confirm the limited performance drop under our proposed relaxed sampling verification. We will also add the ranking performance to Figure 3 in our manuscript.
>
> Table 1. Ranking performance of our method and the ablation variants under **strict top$K$ verification**.
>
> | Beauty       |               |            |            |            |            |
> |--------------|---------------|:----------:|:----------:|:----------:|:----------:|
> | Verification |               |  Recall@5  |  Recall@10 |   NDCG@5   |   NDCG@10  |
> | Strict TopK  | w/o CA        |   0.0056   |   0.0098   |   0.0051   |   0.0066   |
> |              | w/o DR        |   0.0056   |   0.0098   |   0.0051   |   0.0066   |
> |              | w/o TA        |   0.0056   |   0.0098   |   0.0051   |   0.0066   |
> |              | **AtSpeed-S** | **0.0056** | **0.0098** | **0.0051** | **0.0066** |
>
>
> Table 2. Ranking performance of our method and the ablation variants under **relaxed sampling verification**.
>
> | Beauty |  |  |  |  |  |
> |---|---|:---:|:---:|:---:|:---:|
> | Verification |  | Recall@5 | Recall@10 | NDCG@5 | NDCG@10 |
> | Relaxed Sampling | Target LLM (topK) | 0.0056 | 0.0098 | 0.0051 | 0.0066 |
> |  | w/o CA | 0.0061 | 0.0094 | 0.0047 | 0.0058 |
> |  | w/o TopK | 0.0059 | 0.0094 | 0.0044 | 0.0057 |
> |  | w/o TA | 0.0060 | 0.0097 | 0.0049 | 0.0063 |
> |  | **AtSpeed-R** | **0.0058** | **0.0092** | **0.0049** | **0.0063** |
> |  | **_Average_** | **_0.0059_** | **_0.0094_** | **_0.0047_** | **_0.0060_** |

---

> ### Author Response · Authors · 2024-11-22
> **Reply to Weakness 2 (Presentation Adjustment of Table 1)**
>
> > **Weakness 2. Presentation issues. Reporting WS@K and AS@K for all values of K in {1, 3, 5, 10, 20} in Table 1 seems unnecessary, especially since the discussion focuses on average WS and AS. I suggest presenting only representative Ks alongside the averages, freeing space for additional ranking metrics.**
>
> **Reply**: Thanks for your great comments.
> Following your suggestion, we will add the ranking performance to Table 1 together with the acceleration performance. Since this work mainly focuses on inference acceleration, we consider reporting the acceleration metrics WS@K and AS@K with K=5,10,20 alongside the average of K=1,3,5,10,20, and present the Recall@5 and NDCG@5 in Table 1. The comprehensive results of ranking performance (as presented in the ``Reply to Weakness 1``) and acceleration performance will be presented in the Appendix.
>
> The adjusted Table 1 is shown below. We will also update the adjustments in our latest manuscript. We will promptly update the manuscript once we finish the revision.
>
> | Beauty |  |  |  |  |  |  |  |  |  |  |  |
> |---|---|:---:|:---:|:---:|:---:|:---:|:---:|:---:|:---:|:---:|:---:|
> | Verification | Method | WS@5 | WS@10 | WS@20 | Avg WS | AS@5 | AS@10 | AS@20 | Avg AS | Recall@5 | NDCG@5 |
> | Strict Top-K  | SFT | 1.43 | 1.37 | 1.55 | 1.56 | 1.32 | 0.66 | 0.09 | 1.18 | 0.0056 | 0.0051 |
> |  | WordKD | 1.58 | 1.52 | 1.58 | 1.68 | 1.60 | 1.03 | 0.16 | 1.40 | 0.0056 | 0.0051 |
> |  | TVDKD | 1.44 | 1.37 | 1.57 | 1.55 | 1.31 | 0.65 | 0.09 | 1.17 | 0.0056 | 0.0051 |
> |  | SeqKD | 1.75 | 1.67 | 1.68 | 1.83 | 1.85 | 1.27 | 0.30 | 1.60 | 0.0056 | 0.0051 |
> |  | **AtSpeed-S** | **1.84** | **1.87** | **1.84** | **1.97** | **2.00** | **1.64** | **0.57** | **1.80** | 0.0056 | 0.0051 |
> |  | AtSpeed-R | 1.70 | 1.71 | 1.74 | 1.76 | 1.82 | 1.33 | 0.43 | 1.56 | 0.0056 | 0.0051 |
> | Relaxed Sampling | SFT | 1.80 | 2.06 | 2.36 | 1.95 | 2.03 | 1.99 | 1.48 | 1.94 | 0.0057 (+0.0001) | 0.0041 (-0.0010) |
> |  | WordKD | 1.81 | 1.99 | 2.05 | 1.87 | 2.01 | 1.87 | 1.07 | 1.82 | 0.0066 (+0.0010) | 0.0043 (-0.0008) |
> |  | TVDKD | 1.81 | 2.06 | 2.35 | 1.96 | 2.03 | 1.99 | 1.45 | 1.94 | 0.0057 (+0.0001) | 0.0045 (-0.0006) |
> |  | SeqKD | 1.90 | 2.11 | 2.31 | 2.01 | 2.10 | 2.01 | 1.40 | 1.97 | 0.0055 (-0.0001) | 0.0045 (-0.0006) |
> |  | AtSpeed-S | 1.89 | 2.12 | **2.51** | 2.07 | 2.09 | **2.03** | 1.71 | 2.05 | 0.0060 (+0.0004) | 0.0046 (-0.0005) |
> |  | **AtSpeed-R** | **1.94** | **2.16** | 2.47 | **2.11** | **2.13** | 2.01 | **1.77** | **2.10** | 0.0058 (+0.0002) | 0.0049 (-0.0002) |
>
> | Games |  |  |  |  |  |  |  |  |  |  |  |
> |---|---|:---:|:---:|:---:|:---:|:---:|:---:|:---:|:---:|:---:|:---:|
> | Verification | Method | WS@5 | WS@10 | WS@20 | Avg WS | AS@5 | AS@10 | AS@20 | Avg AS | Recall@5 | NDCG@5 |
> | Strict Top-K  | SFT | 1.43 | 1.40 | 1.58 | 1.53 | 1.49 | 1.32 | 0.91 | 1.20 | 0.0074 | 0.0065 |
> |  | WordKD | 1.31 | 1.35 | 1.47 | 1.48 | 1.49 | 1.10 | 0.80 | 1.13 | 0.0074 | 0.0065 |
> |  | TVDKD | 1.24 | 1.32 | 1.50 | 1.42 | 1.26 | 0.95 | 0.66 | 0.99 | 0.0074 | 0.0065 |
> |  | SeqKD | 1.60 | 1.46 | **1.77** | 1.71 | 1.95 | 1.67 | 1.05 | 1.55 | 0.0074 | 0.0065 |
> |  | **AtSpeed-S** | **1.78** | **1.85** | 1.76 | **1.83** | **2.02** | **1.96** | **1.69** | **1.72** | 0.0074 | 0.0065 |
> |  | AtSpeed-R | 1.76 | 1.76 | 1.60 | 1.74 | 1.98 | 1.95 | 1.53 | 1.59 | 0.0074 | 0.0065 |
> | Relaxed Sampling | SFT | 1.84 | 1.97 | 1.69 | 1.86 | 2.12 | 2.05 | 1.89 | 1.78 | 0.0073 (-0.0001) | 0.0060 (-0.0005) |
> |  | WordKD | 1.78 | 1.84 | 1.56 | 1.76 | 2.05 | 1.99 | 1.68 | 1.63 | 0.0072 (-0.0002) | 0.0058 (-0.0007) |
> |  | TVDKD | 1.81 | 1.90 | 1.55 | 1.80 | 2.08 | 2.02 | 1.80 | 1.69 | 0.0069 (-0.0005) | 0.0061 (-0.0004) |
> |  | SeqKD | 1.90 | 2.03 | 2.05 | 1.95 | 2.13 | 2.10 | 1.98 | 1.93 | 0.0071 (-0.0003) | 0.0059 (-0.0006) |
> |  | AtSpeed-S | 1.91 | 2.04 | 2.13 | 2.04 | **2.19** | 2.10 | 1.98 | 2.00 | 0.0080 (+0.0006) | 0.0068 (+0.0003) |
> |  | **AtSpeed-R** | **2.00** | **2.05** | **2.20** | **2.05** | 2.18 | **2.17** | **1.98** | **2.02** | 0.0076 (+0.0002) | 0.0063 (-0.0002) |

---

> ### Author Response · Authors · 2024-11-22
> **Discussion on Great Related Paper Shared by the Reviewer**
>
> > **Related work: Lastly, I want to share a related paper, "Inductive Generative Recommendation via Retrieval-based Speculation". This paper was released after the ICLR submission deadline and is currently available only as a preprint. While this is not a critique or question, it’s relevant as it discusses speculative decoding in generative recommendation. It proposes a dynamic N for the draft model and introduces a technique to perform limited beam search, using prefixes generated by the first few steps of the target model to guide the draft model. I believe this work shares some conceptual similarities with this paper, so I thought it worth mentioning.**
>
> **Reply**: Thank you for sharing this interesting related work. We also recognize this work after its recent release and we will add this paper to our manuscript. It is a great work, which cleverly leverages the “draft-then-verify” paradigm to allow high-quality unseen items to be introduced into the system and thus address cold-start issue. Though conceptually similar to our work, we would like to discuss the difference between our work and this related work (named SpecGR).
>
> While SpecGR focuses on drafting unseen items for generative models to rerank, we aim at drafting beam sequences at every step that align well with the target LLM to reduce decoding steps. Specifically,
>
> - SpecGR aims to address the challenge for generative models to generate unseen items. It uses a draft model to allow unseen items to be reranked by generative models. As you mentioned, SpecGR introduces a guided re-drafting strategy, which essentially aligns the draft model with the target LLM on the unseen items (i.e., low-probability area of the target distribution).
> - In contrast, we focus on addressing the $N$-to-$K$ verification challenge when applying speculative decoding on inference acceleration. To tackle this problem, the key lies in the strong alignment over the top$K$ generated sequence between draft model and the target LLM. Therefore, our work proposes an alignment training framework to strengthen the alignment between the draft model and the target LLM on the top$K$ high-probability area of the distribution.
>
> We will also add the discussion to our latest manuscript.

---

### Author Response · Authors · 2024-12-03
**Brief Summary of Discussions**

We sincerely appreciate the thoughtful and constructive feedback provided by the three reviewers. We are delighted that all reviewers recognized our idea to be novel (Reviewers ``y4np``, ``Up12``), intuitive (Reviewer ``Up12``), and mathematically sound (Reviewer ``WeyY``). Furthermore, we are grateful for the reviewers' acknowledgment of the effectiveness of our experimental results and the reproducibility of our work (Reviewers ``y4np``, ``WeyY``, ``Up12``).

We also greatly value the detailed feedback regarding potential weaknesses in our study, which provided an opportunity to further improve our work. The major concerns raised by the reviewers included:

- **Insufficient results on ranking performance (Reviewer ``y4np``)**: In response, we have supplemented with comprehensive experimental results on recommendation performance across all methods. The results further validate the ranking capability of our proposed relaxed sampling verification strategy, as acknowledged by the reviewer.

- **Generalization ability across diverse datasets and comparison with more SD-based baselines (Reviewer ``WeyY``)**: To address this concern, we conducted experiments on two additional datasets (MovieLens-1M and Goodreads), comparing our method against all baselines and an additional SD-based method DARE. The consistent superior performance demonstrated by our approach substantiates its strong generalization ability, thereby addressing the reviewer’s concerns regarding datasets and baselines.

- **Mathematical rigor in proofs and notations (Reviewer ``Up12``)**: We have provided detailed clarifications, including step-by-step derivations of the alignment objective for AtSpeed-S and the proof of with-replacement sampling approximation. The reviewer carefully examined these derivations and the updated manuscript, ultimately finding the mathematics of our work to be reasonable and convincing.

**Following the discussion period, we are pleased that all reviewers expressed satisfaction with our responses, which effectively addressed most of their concerns**. Overall, we extend our heartfelt thanks to the reviewers for their hard work and active engagement during the discussion phase. Their valuable and constructive comments have been instrumental in refining and enhancing the quality of our paper.

---

### Meta-Review · Area_Chair_6q28 · 2024-12-23

**Metareview:**

This paper studied the problem of speeding up using LLM-based model for top K recommendations, based on the framework of speculative decoding.

Strength:
1. An potentially useful technique for speeding up with LLM-based generative top K recommendation with some promising results.


Weakness:
1. Presentation and experimental  needs improvement.
2. Experimental results are a bit limited to two Amazon datasets. (Authors did expand on Movie-Lens in rebuttal.)

**Additional Comments On Reviewer Discussion:**

All reviewers agreed that this is a good contribution to the conference. During the rebuttal, concerns were addressed and reviewers were satisfied with the rebuttal.

---

### Decision · Program_Chairs · 2025-01-22

Accept (Poster)